# Unsupervised Transfer Learning via Adversarial Contrastive Training

## Abstract

Learning transferable data representations from abundant unlabeled data remains a critical yet challenging task in machine learning. While numerous self-supervised contrastive learning methods have emerged to address this challenge, a notable class of these approaches focuses on aligning the covariance or correlation matrix with the identity matrix. Despite their impressive performance across various downstream tasks, these methods often suffer from biased sample risk. This bias not only leads to significant optimization offsets, especially in mini-batch scenarios, but also complicates the development of theoretical frameworks. In this paper, we introduce Adversarial Contrastive Training (ACT), a novel unbiased self-supervised transfer learning approach. This method allows us to develop a comprehensive end-to-end theoretical analysis for self-supervised contrastive learning. Our theoretical results reveal that minimaxing the loss function of ACT can lead to the downstream data distribution being clustered in the representation space, provided that the upstream unlabeled sample size is sufficient. As a result, even with a few downstream samples, ACT can achieve outstanding classification performance, offering valuable insights for few-shot learning. Furthermore, ACT demonstrates state-of-the-art classification performance across multiple benchmark datasets.

## 1. Introduction

Collecting unlabeled data is far more convenient and cost-effective than gathering labeled data in real-world applications. Consequently, learning representations from abundant unlabeled data presents a highly valuable yet challenging problem. The learned representations can be transferred to downstream tasks to enhance model performance or reduce the sample size required for those tasks.

Recently, self-supervised contrastive learning has emerged as a leading approach for learning representations from unlabeled data. This method aims to learn representations that are invariant to data augmentation. However, solely min-

imizing the distance between similar pairs leads to trivial solutions, known as model collapse. To address this issue, researchers have developed various strategies, broadly categorized into three types.

The first strategy treats augmented views of different images as negative pairs, ensuring their representations remain dissimilar (Ye et al., 2019; He et al., 2020; Chen et al., 2020a;b; HaoChen et al., 2021; Zhang et al., 2023). However, these methods require large batch sizes to ensure sufficient negative samples, leading to substantial computational and memory demands that may be prohibitive in many applications. Additionally, by treating augmented views of different images as negative pairs, these approaches fail to account for semantic similarities between distinct images, potentially forcing apart representations of conceptually related content. As pointed out by Chuang et al. (2020; 2022), this design can hurt the representation performance.

The second strategy prevents model collapse through asymmetric network architectures (Grill et al., 2020; Chen & He, 2021; Caron et al., 2020; 2021). Although eliminating the need for negative pairs, they exhibit significant sensitivity to architectural design choices, where minor modifications can lead to collapsed solutions (Grill et al., 2020; Chen & He, 2021). The specific architectural constraints may also limit the neural network's approximation capabilities. Besides, these methods simultaneously introduce significant challenges for explanation.

The third strategy prevents model collapse by imposing a regularization term to align the covariance or correlation matrix with the identity matrix (Zbontar et al., 2021; Ermolov et al., 2021; Bardes et al., 2022; HaoChen et al., 2022; HaoChen & Ma, 2023; Huang et al., 2023), encouraging the separation of category centers. These methods do not require negative samples and also facilitate a clear theoretical understanding. Among them, a typical regularization term takes the form (Zbontar et al., 2021; HaoChen & Ma, 2023; Huang et al., 2023) as:

$$\mathcal{R}(f) = \left\| \mathbb{E}_{\boldsymbol{x}} \mathbb{E}_{\boldsymbol{x}_1, \boldsymbol{x}_2 \in \mathcal{A}(\boldsymbol{x})} \left\{ f(\boldsymbol{x}_1) f(\boldsymbol{x}_2)^\top \right\} - I_{d^*} \right\|_F^2, \quad (1)$$

where $f : \mathbb{R}^d \to \mathbb{R}^{d^*}$ denotes the representation mapping from the original image space to the representation space, $\|\cdot\|_F$ denotes the Frobenius norm, $\boldsymbol{x}$ represents an original image, and $\mathcal{A}(\boldsymbol{x})$ denotes the collection of all augmented

views of $\boldsymbol{x}$. The terms $\boldsymbol{x}_1, \boldsymbol{x}_2 \in \mathcal{A}(\boldsymbol{x})$ indicate two augmented views independently and uniformly sampled from $\mathcal{A}(\boldsymbol{x})$, while $I_{d^*}$ is the identity matrix with the same dimension $d^*$ as the representation space.

The population risk defined in equation (1) is typically intractable. The following sample-level risk is used to estimate it (HaoChen et al., 2022; HaoChen & Ma, 2023; Zbontar et al., 2021):

$$\widehat{\mathcal{R}}(f) = \left\| \frac{1}{n} \sum_{i=1}^{n} f(\boldsymbol{x}_1^{(i)}) f(\boldsymbol{x}_2^{(i)})^\top - I_{d^*} \right\|_F^2, \quad (2)$$

where $\{\boldsymbol{x}^{(i)}\}_{i \in [n]}$ denotes the original dataset, and $\widetilde{D}_s = \{(\boldsymbol{x}_1^{(i)}, \boldsymbol{x}_2^{(i)}) : \boldsymbol{x}_1^{(i)}, \boldsymbol{x}_2^{(i)} \in \mathcal{A}(\boldsymbol{x}^{(i)})\}_{i \in [n]}$ represents the augmented dataset for learning representations. Unfortunately, it is evident that $\widehat{\mathcal{R}}(f)$ is a biased estimator of $\mathcal{R}(f)$, i.e., $\mathbb{E}_{\widetilde{D}_S}\{\widehat{\mathcal{R}}(f)\} \neq \mathcal{R}(f)$ due to the non-commutativity between the expectation and the Frobenius norm. Two significant challenges emerge due to this bias nature.

Firstly, the biased estimator (2) used in HaoChen et al. (2022); HaoChen & Ma (2023); Zbontar et al. (2021) introduces significant optimization deviations during the training procedure. Although theoretically $\mathbb{E}_{\widetilde{D}_s}\{\widehat{\mathcal{R}}(f)\}$ converges to $\mathcal{R}(f)$ as $n$ approaches infinity, practical constraints necessitate the use of mini-batch samples to estimate $\mathbb{E}_{\boldsymbol{x}} \mathbb{E}_{\boldsymbol{x}_1, \boldsymbol{x}_2 \in \mathcal{A}(\boldsymbol{x})}\{f(\boldsymbol{x}_1) f(\boldsymbol{x}_2)^\top\}$. In this regard, the bias leads to an offset in the optimization direction. Furthermore, this offset would compound across successive training iterations, as each gradient direction strongly depends on the previous one, ultimately resulting in a learned representation that may diverge significantly from the intended minimizer of the population risk in equation (1), as shown in Table 1.

Secondly, this inherent bias presents significant obstacles in establishing end-to-end theoretical guarantees. The development of such guarantees requires addressing three crucial aspects: *how does the downstream task error converge with respect to both the number of unlabeled samples in the source domain and labeled samples in the target domain, how does the abundance of unlabeled samples in self-supervised learning benefit downstream tasks, and why do self-supervised learning methods maintain their effectiveness even with limited downstream labeled data?*

Recent theoretical studies have significantly advanced our understanding of self-supervised learning. These studies can be categorized into two main lines of research. The first line (Garrido et al., 2022; HaoChen et al., 2022; Awasthi et al., 2022; Huang et al., 2023) focuses on analyzing the population risk of self-supervised learning methods. Consequently, these fundamental questions remain incompletely addressed due to the lack of discussion at the sample level. A comprehensive theoretical analysis requires bridging the gap between population-level and sample-level risks, which is a challenging task due to the biasedness of methods such as Zbontar et al. (2021); HaoChen et al. (2022); HaoChen & Ma (2023).

The second line of theoretical research (Saunshi et al., 2019; HaoChen et al., 2021; Ash et al., 2022; Lei et al., 2023; HaoChen & Ma, 2023) studies the generalization error through Rademacher complexity while overlooking the approximation error. Since the learning performance is decided by the overall error, which is the summation of generalization error (evaluated by the Rademacher complexity) and approximation error, the error analysis yielded from the second line may be invalid.

In this study, we introduce **A**dversarial **C**ontrastive **T**raining (ACT), a novel unbiased approach to self-supervised learning. ACT implements an innovative iteration format that eliminates the bias between the population risk (1) and its sample-level counterpart. This advancement effectively addresses two critical challenges: the training deviation and the theoretical obstacle introduced by bias. Through comprehensive end-to-end analysis of ACT, we demonstrate how the number of unlabeled data in the self-supervised pre-training phase enhances downstream task performance. Specifically, we demonstrate that through representation learning using ACT, the downstream data can be clustered in the representation space, provided that the upstream unlabeled sample size is sufficient. As a result, even with a few downstream samples, ACT can achieve outstanding classification performance, offering valuable insights for few-shot learning.

### 1.1. Related Work

**Self-Supervised Loss** The loss function proposed by HaoChen et al. (2022) can be regarded as a special version of ACT with the constraint $\boldsymbol{x}_1 = \boldsymbol{x}_2$. The main difference between ACT and the approach by HaoChen et al. (2022) lies in the iteration format. As stated in Section 1, optimization deviation can accumulate with each iteration, particularly in the mini-batch scenario, while ACT employs adversarial training to mitigate this issue. The same problem is encountered by Zbontar et al. (2021), which can be loosely regarded as a biased sample version of (1).

**Self-Supervised Theory** Recent theoretical studies can be categorized into two main lines of research. The first line (Garrido et al., 2022; HaoChen et al., 2022; Awasthi et al., 2022; Huang et al., 2023) focuses on analyzing the population risk of self-supervised learning methods, which can not characterize how the error in downstream tasks diminishes with increasing sample size. The second line of research (Saunshi et al., 2019; HaoChen et al., 2021; Ash et al., 2022; Lei et al., 2023; HaoChen & Ma, 2023) studies

the generalization error through Rademacher complexity without the consideration of approximation error. However, the scarcity of approximation error makes the resulting error analysis ineffective. Specifically, ignoring the approximation error by simply supposing $f$ belonging to a deep neural network class, the Rademacher complexity can be significantly reduced by controlling the scale of the network class, leading to impressive upper bounds. However, this controlled neural network class intuitively limits its approximation capacity. The increasing approximation error results in a larger overall error. Therefore, these studies cannot provide theoretical guidance for hypothesis class selection nor fully characterize the total error of self-supervised learning methods. In contrast, our work provides a comprehensive convergence analysis that characterizes how the downstream task error converges with respect to both the number of unlabeled samples in the source domain and labeled samples in the target domain.

### 1.2. Contributions

Our main contributions can be summarized as follows:

- We introduce Adversarial Contrastive Training (ACT), a novel self-supervised transfer learning method. This approach learns representations from unlabeled data by solving a min-max optimization problem that corrects the bias inherent in existing methods (HaoChen et al., 2022; Zbontar et al., 2021).

- Through extensive experiments, we demonstrate that ACT significantly outperforms traditional biased iterative methods (Table 1). Our empirical evaluation shows that ACT achieves state-of-the-art classification performance across multiple benchmark datasets using both fine-tuned linear probes and $k$-nearest neighbor ($k$-nn) protocols (Table 2).

- We establish comprehensive end-to-end theoretical guarantees for ACT in transfer learning scenarios under misspecified and overparameterized settings (Theorem 3.9). Our theoretical analysis demonstrates that ACT-learned representations that minimaxing the loss function of ACT can lead to the downstream data distribution being clustered by category in the representation space, provided that the upstream unlabeled sample size is sufficient. Hence, even with a few downstream samples, ACT can achieve outstanding classification performance, offering valuable insights for few-shot learning.

### 1.3. Preliminaries

Given an integer $n \in \mathbb{N}$, we use $[n]$ to represent the integer set $\{1, 2, \cdots, n\}$. For any vector $\boldsymbol{v}$, we denote

$\|\boldsymbol{v}\|_2$ and $\|\boldsymbol{v}\|_\infty$ as the 2-norm and $\infty$-norm of $\boldsymbol{v}$ respectively. Let $A, B \in \mathbb{R}^{d_1 \times d_2}$ be two matrices, we denote their Frobenius inner product by $\langle A, B \rangle_F = \text{tr}(A^\top B)$. Moreover, we denote $\|A\|_F$ as the Frobenius norm of $A$, which is the norm induced by Frobenius inner product, and $\|A\|_\infty = \sup_{\|\boldsymbol{x}\|_\infty \leq 1} \|A\boldsymbol{x}\|_\infty$ as the $\infty$-norm of $a$, which is the maximum 1-norm of the rows of $A$. For a given map $f$ and $0 \leq a_1 \leq a_2$, we use $a_1 \leq \|f\|_2 \leq a_2$ to denote $b_1 \leq \inf_{\boldsymbol{v}} \|f(\boldsymbol{v})\|_2 \leq \sup_{\boldsymbol{v}} \|f(\boldsymbol{v})\|_2 \leq b_2$. Besides that, the Lipschitz norm of $f$ is given by $\|f\|_{\text{Lip}} = \sup_{\boldsymbol{u} \neq \boldsymbol{v}} \frac{\|f(\boldsymbol{u}) - f(\boldsymbol{v})\|_2}{\|\boldsymbol{u} - \boldsymbol{v}\|_2}$. Furthermore, for a given function $f : \mathbb{R}^{d_1} \to \mathbb{R}^{d_2}$, we use $f \in \text{Lip}(L)$ to represent $\|f\|_{\text{Lip}} \leq L$. For ease of presentation, throughout this paper, we use $X \lesssim Y$ or $Y \gtrsim X$ to denote the statement that $X \leq CY$ for two quantities $X$ and $Y$, where $C > 0$ can be arbitrary constant.

We will adopt the following ReLU neural network class as the hypothesis space in the subsequent content.

**Definition 1.1** (ReLU neural network class). Given $0 < d_1, d_2; L, N_1, \ldots, N_L \in \mathbb{N}; 0 < \mathcal{K}$ and $0 < B_1 \leq B_2$, define $W = \max\{N_1, \ldots, N_L\}$, a deep ReLU network class with parameter $(W, L, \mathcal{K}, B_1, B_2)$, $\mathcal{NN}_{d_1,d_2}(W, L, \mathcal{K}, B_1, B_2)$, is defined as the collection of all maps of the form

$$f_{\boldsymbol{\theta}}(\boldsymbol{x}) = A_L \sigma(A_{L-1} \sigma(\cdots \sigma(A_0 \boldsymbol{x} + b_0)) + b_{L-1})$$

such that $B_1 \leq \|f_{\boldsymbol{\theta}}\|_2 \leq B_2$ and $\kappa(\boldsymbol{\theta}) \leq \mathcal{K}$, where $\sigma(x) = \max\{0, x\}$ is the ReLU activate function, $N_0 = d_1$, $N_{L+1} = d_2$, $A_i \in \mathbb{R}^{N_{i+1} \times N_i}$ and $\boldsymbol{b}_i \in \mathbb{R}^{N_{i+1}}$. The integers $W$ and $L$ are called the width and depth of the neural network respectively. The parameters set of the neural network is defined as $\boldsymbol{\theta} := ((A_0, \boldsymbol{b}_0), \ldots, (A_{L-1}, \boldsymbol{b}_{L-1}), A_L)$. Further, $\kappa(\boldsymbol{\theta})$ is defined as $\kappa(\boldsymbol{\theta}) = \|A_L\|_\infty \prod_{l=0}^{L-1} \max\{\|(A_l, \boldsymbol{b}_l)\|_\infty, 1\}$.

For any $f_{\boldsymbol{\theta}} \in \mathcal{NN}_{d_1,d_2}(W, L, \mathcal{K}, B_1, B_2)$, we can justify $\|f_{\boldsymbol{\theta}}\|_{\text{Lip}} \leq \mathcal{K}$. The proof details are deferred to Appendix A.1.

Besides that, for any two measures $\mu$ and $\nu$, we define the 1-Wasserstein distance as $\mathcal{W}(\mu, \nu) = \max_{g \in \text{Lip}(1)} \mathbb{E}_{X \sim \mu}\{g(X)\} - \mathbb{E}_{Y \sim \nu}\{g(Y)\}$.

### 1.4. Organization

This paper is structured as follows: Section 2 introduces the core concept of ACT and presents our alternating optimization algorithm. In Section 3, we develop a comprehensive end-to-end theoretical guarantee for ACT. Section 4 demonstrates ACT's effectiveness through extensive experimental evaluations across diverse datasets and metrics. Section 5 concludes with a summary of our findings. All detailed proofs are provided in Section A.

## 2. Adversarial Contrastive Training

### 2.1. Notations for Unsupervised Transfer Learning

Throughout this paper, we use $d$ and $d^*$ to represent the dimensions of the original image and the representation dimension, respectively. We denote image instances from the source domain $\mathcal{X}_s \subseteq [0,1]^d$ with distribution $\mathbb{P}_s$ using the letter $\boldsymbol{x}$ and its subscripted or superscripted variants. In contrast, we use the letter $\boldsymbol{z}$ and its subscripted or superscripted variants for image instances from the target domain $\mathcal{X}_t \subseteq [0,1]^d$ with distribution $\mathbb{P}_t$. In this context, we can independently and identically sample a total of $n_s$ source image instances from $\mathbb{P}_s$ and $n_t$ downstream samples from $\mathbb{P}_t$. Notably, the label for each $\boldsymbol{z}^{(i)} \sim \mathbb{P}_t$ is observable. We refer to these two datasets as $D_s = \{\boldsymbol{x}^{(i)}\}_{i\in[n_s]}$ and $D_t = \{(\boldsymbol{z}^{(i)}, y_i)\}_{i\in[n_t]}$, respectively.

Since the primary objective of contrastive learning is to learn a representation that is invariant to augmentations, data augmentation plays a crucial role in this area. A data augmentation $A : \mathbb{R}^d \to \mathbb{R}^d$ is essentially a predefined transformation applied to original images. Common augmentations include a composition of random transformations, such as Random-Crop, HorizontalFlip, and Color Distortion (Chen et al., 2020a). We refer to the collection of used data augmentations as $\mathcal{A} = \{A_i(\cdot)\}_{i\in[m]}$ as the collection of used data augmentations, where $m$ is the total number of data augmentation under consideration. Theoretically, $m$ could be infinite. But we might consider only a finite but sufficiently large $m$ for convenient theoretical treatment. In fact, as long as $m$ is sufficiently large, essentially any type of data augmentation might be well approximated by some $A \in \mathcal{A}$. Base on $\mathcal{A}$, we can construct an augmented dataset $\widetilde{D}_s = \{\tilde{\boldsymbol{x}}^{(i)}\}_{i\in[n_s]}$, where $\tilde{\boldsymbol{x}}^{(i)} = (\boldsymbol{x}_1^{(i)}, \boldsymbol{x}_2^{(i)}) = (A_{i,1}(\boldsymbol{x}^{(i)}), A_{i,2}(\boldsymbol{x}^{(i)}))$, and $A_{i,1}$ and $A_{i,1}$ are independently drawn from the uniform distribution on $\mathcal{A}$.

### 2.2. Adversarial Contrastive Training

We begin by recalling $\mathcal{R}(f)$ defined in (1), which is the regularization term adopted by various studies (HaoChen et al., 2022; HaoChen & Ma, 2023; Huang et al., 2023) to prevent model collapse. Its empirical version at the sample level is given by $\widehat{\mathcal{R}}(f)$. However, as stated in Section 1, $\widehat{\mathcal{R}}(f)$ is a biased counterpart of $\mathcal{R}(f)$, i.e., $\mathbb{E}_{\widetilde{D}_s}\{\widehat{\mathcal{R}}(f)\} \neq \mathcal{R}(f)$, which hinders establishing a theoretical foundation at the sample level and introduces optimization deviation.

To address these two issues, we then propose a novel sample-level estimator for the population risk (1). A key observation to motivate ACT is that we can rewrite $\mathcal{R}(f)$ as

$$\mathcal{R}(f) = \sup_{G \in \mathcal{G}(f)} \mathcal{R}(f, G), \qquad (3)$$

where $G \in \mathbb{R}^{d^* \times d^*}$ is a matrix variable, and

$$\mathcal{R}(f, G) = \langle \mathbb{E}_{\boldsymbol{x}}\mathbb{E}_{\boldsymbol{x}_1, \boldsymbol{x}_2 \in \mathcal{A}(\boldsymbol{x})}\{f(\boldsymbol{x}_1)f(\boldsymbol{x}_2)^\top\} - I_{d^*}, G \rangle_F,$$
$$\mathcal{G}(f) = \{G \in \mathbb{R}^{d^* \times d^*} : \|G\|_F \leq \sqrt{\mathcal{R}(f)}\}.$$

The equation (3) holds because of the fact that $\langle A, B \rangle_F \leq \|A\|_F\|B\|_F$ for any matrices $A, B$ of same dimension, with equality holding if and only if $A = B$. Correspondingly, the sample-level counterpart associated with (3) is given by

$$\widehat{\mathcal{R}}(f) = \sup_{G \in \widehat{\mathcal{G}}(f)} \widehat{\mathcal{R}}(f, G),$$

where

$$\widehat{\mathcal{R}}(f, G) = \langle \frac{1}{n_s}\sum_{i=1}^{n_s} f(\boldsymbol{x}_1^{(i)})f(\boldsymbol{x}_2^{(i)})^\top - I_{d^*}, G \rangle_F,$$
$$\widehat{\mathcal{G}}(f) = \{G \in \mathbb{R}^{d^* \times d^*} : \|G\|_F \leq \sqrt{\widehat{\mathcal{R}}(f)}\}.$$

It can be shown,

$$\mathcal{R}(f, G) = \mathbb{E}_{\widetilde{D}_s}\{\widehat{\mathcal{R}}(f, G)\}.$$

Hence, the equivalent transformation (3) help us avoid the issue of biasedness. Specifically, with the equivalent transformation (3) and its empirical version, we learn the contrastive representation through the Adversarial Contrastive Training (ACT) at the sample level, which can be formulated as a mini-max problem as follows:

$$\hat{f}_{n_s} \in \arg\min_{f \in \mathcal{F}} \max_{G \in \widehat{\mathcal{G}}(f)} \widehat{\mathcal{L}}(f, G), \qquad (4)$$
$$\widehat{\mathcal{L}}(f, G) = \widehat{\mathcal{L}}_{\text{align}}(f) + \lambda\widehat{\mathcal{R}}(f, G),$$
$$\widehat{\mathcal{L}}_{\text{align}}(f) = \frac{1}{n_s}\sum_{i=1}^{n_s} \|f(\boldsymbol{x}_1^{(i)}) - f(\boldsymbol{x}_2^{(i)})\|_2^2,$$

where $\mathcal{F}$ is defined as $\mathcal{NN}(W, L, \mathcal{K}, B_1, B_2)$. We will specify the appropriate parameters $(W, L, \mathcal{K}, B_1, B_2)$ to satisfy the theoretical requirements in Section 3. The term $\widehat{\mathcal{L}}_{\text{align}}(f)$ embodies the core idea of contrastive learning: learning a representation that is invariant to augmentations. Additionally, $\lambda > 0$ serves as the regularization hyperparameter.

This mini-max problem naturally leads to an alternative optimization algorithm for solving it, where $G$ is fixed during the optimization of the encoder $f$ and $f$ is fixed when optimizing $G$. We present this algorithm in Algorithm 1. It is important to note that $G_t$ has been detached from the computational graph when updating the encoder parameters $\boldsymbol{\theta}$. This detachment implies that the gradient with respect to $\boldsymbol{\theta}$ is as given by the seventh line of Algorithm 1, rather than $\nabla_{\boldsymbol{\theta}} \|\frac{1}{N}\sum_{i=1}^N f_{\boldsymbol{\theta}}(\boldsymbol{x}_1^{(n_i^t)})f_{\boldsymbol{\theta}}(\boldsymbol{x}_2^{(n_i^t)})^\top - I_{d^*}\|_F^2$, which is the mini-batch gradient of $\widehat{\mathcal{R}}(f)$. In this regard, such a

**Algorithm 1** Alternative Optimization Algorithm

---

**Require:** Augmented dataset $D_s = \{\tilde{\boldsymbol{x}}^{(i)}\}_{i\in[n]}$, initial encoder parameter $\boldsymbol{\theta}_0$, iteration horizon $T$, mini-batch size $N$, learning rate $\eta$.

1: **for** $t \in \{0\} \cup [T-1]$ **do**
2:     Sample a mini-batch $\mathcal{B}_t = \{\boldsymbol{x}^{(n_i^t)}\}_{i\in N} \subseteq D_s$ of size $N$, where $n_i^t$ represents the index of the $i$-th sample in the mini-batch $\mathcal{B}_t$ within $D_s$.
3:     **if** $t = 0$ **then**
4:         $G_0 = \sum_{i=1}^{N} f_{\boldsymbol{\theta}_0}(\boldsymbol{x}_1^{(n_i^t)}) f_{\boldsymbol{\theta}_0}(\boldsymbol{x}_2^{(n_i^t)})^\top - I_{d^*}$.
5:         Detach: $G_0 \leftarrow G_0.\text{detach}()$.
6:     **end if**
7:     Update encoder $\boldsymbol{\theta}_{t+1} = \boldsymbol{\theta}_t - \eta\Delta_{\boldsymbol{\theta}}$, where $\Delta_{\boldsymbol{\theta}} = \nabla_{\boldsymbol{\theta}} \frac{1}{N} \sum_{i=1}^{N} \|f_{\boldsymbol{\theta}}(\boldsymbol{x}_1^{(n_i^t)}) - f_{\boldsymbol{\theta}}(\boldsymbol{x}_2^{(n_i^t)})\|_2^2 + \left\langle \nabla_{\boldsymbol{\theta}} \frac{1}{N} \sum_{i=1}^{N} f_{\boldsymbol{\theta}}(\boldsymbol{x}_1^{(n_i^t)}) f_{\boldsymbol{\theta}}(\boldsymbol{x}_2^{(n_i^t)})^\top - I_{d^*}, G_t \right\rangle_F$.
8:     $G_{t+1} = \sum_{i=1}^{N} f_{\boldsymbol{\theta}_{t+1}}(\boldsymbol{x}_1^{(n_i^t)}) f_{\boldsymbol{\theta}_{t+1}}(\boldsymbol{x}_2^{(n_i^t)})^\top - I_{d^*}$.
9:     Detach: $G_{t+1} \leftarrow G_{t+1}.\text{detach}()$.
10: **end for**
**output** The learned encoder $f_{\boldsymbol{\theta}_T}$.

---

mini-max iteration format will yield a distinctly different encoder in the mini-batch scenario compared to previous studies (Zbontar et al., 2021; HaoChen et al., 2022).

We compare ACT against two biased self-supervised learning methods: Barlow Twins (Zbontar et al., 2021) and the approach proposed by HaoChen et al. (2022), across multiple benchmark datasets. The experimental results, summarized in Table 1, demonstrate that ACT significantly improves downstream classification accuracy compared to both baseline methods, which are implemented using our repository, with a total training of 1000 epochs and a representation dimension of 512. While, ACT employs representation dimensions of 64, 64, and 128, which are significantly lower than those of Zbontar et al. (2021); HaoChen et al. (2022); yet, it achieved the most outstanding performance.

*Table 1.* Classification accuracy (top 1) of a linear classifier and a 5-nearest neighbors classifier for different methods and datasets. Here BT indicates Barlow Twins (Zbontar et al., 2021) while BS refers to the method proposed by HaoChen et al. (2022).

| Method | CIFAR-10 | | CIFAR-100 | | Tiny ImageNet | |
|---|---|---|---|---|---|---|
| | Linear | $k$-nn | Linear | $k$-nn | Linear | $k$-nn |
| BT | 83.96 | 81.18 | 56.75 | 47.91 | 34.08 | 19.40 |
| BS | 86.95 | 82.83 | 53.75 | 48.40 | 35.80 | 20.36 |
| ACT | **92.11** | **90.01** | **68.24** | **58.35** | **49.72** | **36.40** |

## 3. End-to-End Theoretical Guarantee

### 3.1. Problem Formulation

We first define the ideal $f^*$ for $f$ as the minimizer at the population level, which represents the ideal objective of ACT.

$$f^* \in \underset{f:B_1 \leq \|f\|_2 \leq B_2}{\arg\min} \sup_{G \in \mathcal{G}(f)} \mathcal{L}(f, G),$$
$$\mathcal{L}(f, G) = \mathcal{L}_{\text{align}}(f) + \lambda\mathcal{R}(f, G),$$
$$\mathcal{L}_{\text{align}}(f) = \mathbb{E}_{\boldsymbol{x}} \mathbb{E}_{\boldsymbol{x}_1, \boldsymbol{x}_2 \in \mathcal{A}(\boldsymbol{x})} \{\|f(\boldsymbol{x}_1) - f(\boldsymbol{x}_2)\|_2^2\}.$$

Apart from that, we further denote

$$\mathcal{L}(f) = \mathcal{L}_{\text{align}}(f) + \lambda\mathcal{R}(f).$$

Intuitively, in data representation, the most critical aspect is the differentiation between various features, rather than the specific ranges of their values. Therefore, the constraint $B_1 \leq \|f\|_2 \leq B_2$ will not diminish the performance of the encoder; instead, it facilitates the establishment of theoretical foundations for ACT.

Moreover, following a similar process to that used for obtaining $\widetilde{D}_s$, we can construct the downstream augmented dataset $\widetilde{D}_t = \{(\tilde{\boldsymbol{z}}^{(i)}, y_i)\}_{i\in[n_t]}$, where $\tilde{\boldsymbol{z}}^{(i)} = \{(\boldsymbol{z}_1^{(i)}, \boldsymbol{z}_2^{(i)})\}_{i\in[n_t]}$ with $\boldsymbol{z}_1^{(i)} = A_{i,1}(\boldsymbol{z}^{(i)})$, $\boldsymbol{z}_2^{(i)} = A_{i,2}(\boldsymbol{z}^{(i)})$. Therein, $A_{i,1}$, $A_{i,2}$ are independently and identically distributed samples drawn from the uniform distribution defined on $\mathcal{A}$. In this context, we construct the following linear probe as a classifier:

$$Q_{\hat{f}_{n_s}}(\boldsymbol{z}) = \underset{k\in[K]}{\arg\max} \left(\widehat{W}\hat{f}_{n_s}(\boldsymbol{z})\right)_k, \qquad (5)$$

where the $k$-th row of $\widehat{W}$ is given as $\hat{\mu}_t(k) = \frac{1}{2n_t(k)} \sum_{i=1}^{n_t} (\hat{f}_{n_s}(\boldsymbol{z}_1^{(i)}) + \hat{f}_{n_s}(\boldsymbol{z}_2^{(i)})) \mathbb{1}\{y_i = k\}$, therein, $n_t(k) = \sum_{i=1}^{n_t} \mathbb{1}\{y_i = k\}$. The classifier defined in (5) indicates that by calculating the average representations for each class, we build a template for each downstream class individually. Whenever a new sample needs to be classified, it is assigned to the category of the template that it most closely resembles. Furthermore, we use the following misclassification rate to evaluate the representation learned by ACT.

$$\text{Err}(Q_{\hat{f}_{n_s}}) = \sum_{k=1}^{K} \mathbb{P}_t\{Q_{\hat{f}_{n_s}}(\boldsymbol{z}) \neq k, \boldsymbol{z} \in C_t(k)\}, \quad (6)$$

where $C_t(k)$ is a set such that $\boldsymbol{z} \in C_t(k)$ if and only if $\boldsymbol{z}$ belongs to the $k$-th class. Correspondingly, similar to Huang et al. (2023), we assume that any upstream instance $\boldsymbol{x}$ can be categorized into one or more latent classes $\{C_s(k)\}_{k\in[K]}$. For ease of presentation, let $p_s(k) = \mathbb{P}_s\{\boldsymbol{x} \in C_s(k)\}$ and $\mathbb{P}_s(k)(\cdot) = \mathbb{P}_s\{\cdot | \boldsymbol{x} \in C_s(k)\}$. Similarly, let $p_t(k) =$

$\mathbb{P}_t\{\boldsymbol{z} \in C_t(k)\}$ and $\mathbb{P}_t(k)(\cdot) = \mathbb{P}_t(\cdot|\boldsymbol{z} \in C_t(k))$. In this context, we use the quantities

$$\epsilon_1 = \max_{k \in [K]} \mathcal{W}(\mathbb{P}_s(k), \mathbb{P}_t(k)),$$
$$\epsilon_2 = \max_{k \in [K]} |p_s(k) - p_t(k)|, \tag{7}$$

to measure the divergence between the source and the target domains, where $\mathcal{W}$ denotes 1-Wasserstein distance.

### 3.2. Theoretical Limitation of Bias

In this section, we aim to elucidate the limitations imposed by bias from a theoretical perspective. We first assert that $\mathbb{E}_{\widetilde{D}_s}\{\mathrm{Err}(Q_{\hat{f}_{n_s}})\} \lesssim \mathbb{E}_{\widetilde{D}_s}\{\mathcal{L}(\hat{f}_{n_s})\}$ under specific conditions, the details of which can be found in Section A.2. Consequently, analyzing the sample complexity of $\mathbb{E}_{\widetilde{D}_s}\{\mathcal{L}(\hat{f}_{n_s})\}$ is essential to establish an end-to-end theory for ACT. However, this analysis poses a significant challenge due to the presence of bias.

In fact, in the field of learning theory, the condition $\mathbb{E}_{\widetilde{D}_s}\{\widehat{\mathcal{L}}(f)\} = \mathcal{L}(f)$ is quite important to establish the upper bound of $\mathbb{E}_{\widetilde{D}_s}\{\mathcal{L}(\hat{f}_{n_s})\}$. Specifically, let $\bar{f}$ satisfy $\mathcal{L}(\bar{f}) - \mathcal{L}(f^*) = \inf_{f \in \mathcal{F}}\{\mathcal{L}(f) - \mathcal{L}(f^*)\}$, then

$$\mathcal{L}(\hat{f}_{n_s}) = \mathcal{L}(\hat{f}_{n_s}) - \widehat{\mathcal{L}}(\hat{f}_{n_s}) + \widehat{\mathcal{L}}(\hat{f}_{n_s}) - \widehat{\mathcal{L}}(f^*) + \widehat{\mathcal{L}}(f^*)$$
$$\quad - \mathcal{L}(f^*) + \mathcal{L}(f^*)$$
$$\leq \mathcal{L}(f^*) + 2\sup_{f \in \mathcal{F}}|\mathcal{L}(f) - \widehat{\mathcal{L}}(f)| + \{\widehat{\mathcal{L}}(\hat{f}_{n_s}) - \widehat{\mathcal{L}}(f^*)\}$$
$$\leq \mathcal{L}(f^*) + 2\sup_{f \in \mathcal{F}}|\mathcal{L}(f) - \widehat{\mathcal{L}}(f)| + \{\widehat{\mathcal{L}}(\bar{f}_{n_s}) - \widehat{\mathcal{L}}(f^*)\}.$$

Taking the expectation regarding to $\widetilde{D}_s$ on both sides yields $\mathbb{E}_{\widetilde{D}_s}\{\mathcal{L}(\hat{f}_{n_s})\} \leq \mathcal{L}(f^*) + 2\mathbb{E}_{\widetilde{D}_s}\{\sup_{f \in \mathcal{F}}|\mathcal{L}(f) - \widehat{\mathcal{L}}(f)|\} + \inf_{f \in \mathcal{F}}\{\mathcal{L}(f) - \mathcal{L}(f^*)\}$. As observed, the first term is a typical problem in the area of empirical process, where the sample bound also require unbiasedness; not to mention that such typical risk decomposition itself necessitates a guarantee of unbiasedness. In contrast, based on the modification of ACT, we develop an novel error decomposition as follows:

$$\mathbb{E}_{\widetilde{D}_s}[\mathcal{L}(\hat{f}_{n_s})] \lesssim \mathcal{L}(f^*) + \inf_{f \in \mathcal{F}}\{\mathcal{L}(f) - \mathcal{L}(f^*)\}$$
$$+ \mathbb{E}_{\widetilde{D}_s}\{\sup_{f \in \mathcal{F}, G \in \widehat{\mathcal{G}}(f)} |\mathcal{L}(f, G) - \widehat{\mathcal{L}}(f, G)|\}$$
$$+ \mathbb{E}_{\widetilde{D}_s}\big[\sup_{f \in \mathcal{F}}\{G^*(f) - \widehat{G}(f)\}\big] \tag{8}$$

where $G^*(f) = \mathbb{E}_{\boldsymbol{x}}\mathbb{E}_{\boldsymbol{x}_1, \boldsymbol{x}_2 \in \mathcal{A}(\boldsymbol{x})}\{f(\boldsymbol{x}_1)f(\boldsymbol{x}_2)^\top - I_{d^*}\}$ and $\widehat{G}(f) = \frac{1}{n_s}\sum_{i=1}^{n_s} f(\boldsymbol{x}_1^{(i)})f(\boldsymbol{x}_2^{(i)})^\top - I_{d^*}$. The proof details can be found in Section A.3.3. Through this decomposition in (8), we can systematically analysis $\mathbb{E}_{\widetilde{D}_s}[\mathcal{L}(\hat{f}_{n_s})]$. Particularly, the first term in (8) can be bounded under Assumption 3.3, as will be demonstrated subsequently. The

second term, known as approximation error, represents the error introduced by using $\mathcal{F}$ to approximate $f^*$. Utilizing the unbiasedness of $\widehat{\mathcal{L}}(f, G)$, the third term can be bounded using standard techniques from empirical process theory, while the last term can be reformulated as a common problem regarding the rate of convergence of the empirical mean to the population mean.

### 3.3. Assumptions

We begin with introducing the Hölder class, which plays a curial role in bounding the approximation error, i.e., the second term in (8).

**Definition 3.1** (Hölder class). Let $d \in \mathbb{N}$ and $\alpha = r + \beta > 0$, where $r \in \mathbb{N}_0$ and $\beta \in (0, 1]$. We assert $f : \mathbb{R}^d \to \mathbb{R}$ belongs to the Hölder class $\mathcal{H}^\alpha(\mathbb{R}^d)$ if and only if

$$|\partial^{\boldsymbol{s}} f(\boldsymbol{x})| \leq 1 \text{ and } \max_{\|\boldsymbol{s}\|_1 = r} \sup_{\boldsymbol{x} \neq \boldsymbol{y}} \frac{\partial^{\boldsymbol{s}} f(\boldsymbol{x}) - \partial^{\boldsymbol{s}} f(\boldsymbol{y})}{\|\boldsymbol{x} - \boldsymbol{y}\|_\infty^\beta} \leq 1,$$

where for a multi-index $\boldsymbol{s} = (s_1, \ldots, s_d) \in \mathbb{N}_0^d$ and $f : \mathbb{R}^d \to \mathbb{R}$, the symbol $\partial^{\boldsymbol{s}} f$ denotes the partial differential operator $\partial^{\boldsymbol{s}} = \frac{\partial^{s_1}}{\partial x_1^{s_1}} \frac{\partial^{s_2}}{\partial x_1^{s_2}} \cdots \frac{\partial^{s_d}}{\partial x_d^{s_d}}$. Furthermore, we define $\mathcal{H}^\alpha := \{f : [0, 1]^d \to \mathbb{R}, f \in \mathcal{H}^\alpha(\mathbb{R}^d)\}$ as the restriction of $\mathcal{H}^\alpha(\mathbb{R}^d)$ to $[0, 1]^d$.

The Hölder class is known to be a highly comprehensive functional class, providing a precise characterization of the low-order regularity of functions. In this regard, we make following assumption:

**Assumption 3.2.** There exists $\alpha = r + \beta$ with $r \in \mathbb{N}_0$ and $\beta \in (0, 1]$ s.t $f_i^* \in \mathcal{H}^\alpha$ for each $i \in [d^*]$.

Assumption 3.2 is standard and mild in the context of nonparametric statistics (Tsybakov, 2008; Schmidt-Hieber, 2020) due to the universality of the Hölder class.

As for the term $\mathcal{L}(f^*)$ in eq (8), we make following Assumption 3.3 to ensure $\mathcal{L}(f^*) = 0$.

**Assumption 3.3.** Assume there exists a measurable partition $\{\mathcal{P}_1, \ldots, \mathcal{P}_{d^*}\}$ of $\mathcal{X}_s$, such that $1/B_2^2 \leq \mathbb{P}_s(\mathcal{P}_i) \leq 1/B_1^2$ for each $i \in [d^*]$.

Assumption 3.3 suggests that the data distribution in the source domain should not be overly singular. All common continuous distributions defined on Borel algebra satisfy these requirements, as the measure of any single point is zero. More details are deferred to Section A.3.2.

*Remark* 3.4. (HaoChen & Ma, 2023, Assumption 4.2) assumes that the term $\mathcal{L}(f)$ can be sufficiently minimized by a specific network. Constructing a network $f_{\boldsymbol{\theta}} \in \mathcal{F}$ such that population statistic $\mathcal{L}(f_{\boldsymbol{\theta}})$ is sufficiently small is too complex. In contrast, we consider a more general setting where $f^*$ may not belong to $\mathcal{F}$. Based on the mild Assumption 3.3, we can theoretically illustrate that $\mathcal{L}(f^*)$ vanishes. This is crucial for subsequent theoretical analysis.

Additionally, we need to introduce two assumptions regarding to the data augmentation.

**Assumption 3.5.** Assume any data augmentation $A_i \in \mathcal{A}$ is $M$-Lipschitz map, i.e., $\|A_i(\boldsymbol{v}_1) - A_i(\boldsymbol{v}_2)\|_2 \leq M\|\boldsymbol{v}_1 - \boldsymbol{v}_2\|_2$ for any $\boldsymbol{v}_1, \boldsymbol{v}_2 \in [0, 1]^d$.

A typical example to illustrate Assumption 3.5 is that the augmented view yielded by cropping should not undergo drastic changes when minor perturbations are applied to the original image.

In addition to the Lipschitz property of data augmentation, we adopt Definition 3.6 to mathematically quantify the quality of data augmentations.

**Definition 3.6** (($\sigma_s, \sigma_t, \delta_s, \delta_t$)-Augmentation)**.** The augmentations in $\mathcal{A}$ is $(\sigma_s, \sigma_t, \delta_s, \delta_t)$-augmentations, that is, for each $k \in [K]$, there exists a subset $\widetilde{C}_s(k) \subseteq C_s(k)$ and $\widetilde{C}_t(k) \subseteq C_t(k)$, such that (i) $\mathbb{P}_s\{\boldsymbol{x} \in \widetilde{C}_s(k)\} \geq \sigma_s \mathbb{P}_s\{\boldsymbol{x} \in C_s(k)\}$, (ii) $\sup_{\boldsymbol{x}_1, \boldsymbol{x}_2 \in \widetilde{C}_s(k)} \min_{\boldsymbol{x}_1' \in \mathcal{A}(\boldsymbol{x}_1), \boldsymbol{x}_2' \in \mathcal{A}(\boldsymbol{x}_2)} \|\boldsymbol{x}_1' - \boldsymbol{x}_2'\|_2 \leq \delta_s$; (iii) $\mathbb{P}_t\{\boldsymbol{z} \in \widetilde{C}_t(k)\} \geq \sigma_t \mathbb{P}_t\{\boldsymbol{z} \in C_t(k)\}$, (iv) $\sup_{\boldsymbol{z}_1, \boldsymbol{z}_2 \in \widetilde{C}_t(k)} \min_{\boldsymbol{z}_1' \in \mathcal{A}(\boldsymbol{z}_1), \boldsymbol{z}_2' \in \mathcal{A}(\boldsymbol{z}_2)} \|\boldsymbol{z}_1' - \boldsymbol{z}_2'\|_2 \leq \delta_t$ and (v) $\mathbb{P}_t\{\cup_{k=1}^K \widetilde{C}_t(k)\} \geq \sigma_t$, where $\sigma_s, \sigma_t \in (0, 1]$ and $\delta_s, \delta_t \geq 0$.

The $(\sigma_s, \sigma_t, \delta_s, \delta_t)$-augmentation is an extensive version of the $(\sigma, \delta)$-augmentation proposed by Huang et al. (2023). This definition emphasizes that a robust data augmentation should consistently produce distance-closed augmented views for semantically similar original images. Therein, condition (v) replaces the assumption $\mathcal{A}(C_t(i)) \cap \mathcal{A}(C_t(j)) = \emptyset$ proposed by Huang et al. (2023). This implies that the augmentation methods used should be intelligent enough to recognize objects that align with the image labels in multi-objective images. A straightforward alternative to this requirement is to assume that different classes $C_t(k)$ are pairwise disjoint, meaning that for all $i \neq j$, $C_t(i) \cap C_t(j) = \emptyset$. This implies that $\mathbb{P}_t\{\cup_{k=1}^K \widetilde{C}_t(k)\} = \sum_{k=1}^K \mathbb{P}_t\{\widetilde{C}_t(k)\} \geq \sigma_t \sum_{k=1}^K \mathbb{P}_t\{C_t(k)\} = \sigma_t$.

In the context of Definition 3.6, we introduce the following assumption to delineate the data augmentation necessary for the end-to-end theory of ACT.

**Assumption 3.7** (Existence of augmentation sequence)**.** Assume there exists a sequence of $(\sigma_s^{(n)}, \sigma_t^{(n)}, \delta_s^{(n)}, \delta_t^{(n)})$-data augmentations $\mathcal{A}_n = \{A_i^{(n)}\}_{i \in [m]}$ and $\tau > 0$ such that (i) $\max\{\delta_s^{(n)}, \delta_t^{(n)}\} \leq n^{-\frac{\tau+d+1}{2(\alpha+d+1)}}$, (ii) $\min\{\sigma_s^{(n)}, \sigma_t^{(n)}\} \to 1$ as $n \to \infty$.

It is noteworthy that this assumption closely aligns with Assumption 3.5 in HaoChen et al. (2021) and Assumption 3.6 in HaoChen & Ma (2023), both of which stipulate that the augmentations must be sufficiently robust to ensure that the internal connections within latent classes remain strong enough to prevent the separation of instance clusters. Recently, various methods for developing more effective data augmentations, as discussed by Jahanian et al. (2022) and Trabucco et al. (2024), have been proposed, making it increasingly feasible to satisfy the theoretical requirements for data augmentation. Next, we will introduce the assumption related to distribution shift.

Prior to characterizing the transferability from the source domain to the target domain, we must first quantify the similarity between these domains.

**Assumption 3.8** (Domain shift)**.** Assume there exists $\nu > 0$ and $\varsigma > 0$ such that (i) $\epsilon_1 \lesssim n_s^{-\frac{\nu+d+1}{2(\alpha+d+1)}}$ and (ii) $\epsilon_2 \lesssim n_s^{-\frac{\varsigma}{2(\alpha+d+1)}}$, where $\epsilon_1$ and $\epsilon_2$ measure the divergence between the source and the target domains defined as (7).

As shown in (7), smaller values of $\epsilon_1$ and $\epsilon_2$ indicate less discrepancy between the source and target domains. Similar assumptions using alternative divergence measures have been proposed in Ben-David et al. (2010); Germain et al. (2013); Cortes et al. (2019).

### 3.4. End-to-end Theoretical Guarantee

Let $\mu_t(k) = \mathbb{E}_{\boldsymbol{x} \in C_t(k)} \mathbb{E}_{\boldsymbol{x}' \in \mathcal{A}(\boldsymbol{x})}\{\hat{f}_{n_s}(\boldsymbol{x}')\}$, which is the representation center of $k$-th class $C_t(k)$. We present the end-to-end theoretical guarantee of ACT as follows:

**Theorem 3.9.** *Suppose Assumptions 3.2, 3.3, 3.5, 3.7 and 3.8 all hold. Set the width, depth and the Lipschitz constraint of the deep neural network as*

$$W \geq \mathcal{O}\big(n_s^{\frac{2d+\alpha}{4(\alpha+d+1)}}\big), \quad L \geq \mathcal{O}(1), \quad \mathcal{K} = \mathcal{O}\big(n_s^{\frac{d+1}{2(\alpha+d+1)}}\big),$$

*then the following inequality holds*

$$\mathbb{E}_{\widetilde{D}_s}\{\max_{i \neq j} |\mu_t(i)^\top \mu_t(j)|\} \lesssim (1 - \sigma_s^{(n_s)}) + n_s^{-\frac{\min\{\alpha, 2\tau\}}{4(\alpha+d+1)}}.$$
$$(9)$$

*Furthermore, regarding to the misclassification rate of $Q_{\hat{f}_{n_s}}$, we have*

$$\mathbb{E}_{\widetilde{D}_s}\{\mathrm{Err}(Q_{\hat{f}_{n_s}})\} \leq (1 - \sigma_t^{(n_s)}) + \mathcal{O}\big(n_s^{-\frac{\min\{\alpha, \nu, \varsigma\}}{8(\alpha+d+1)}}\big),$$

*with probability at least $\sigma_s^{(n_s)} - \mathcal{O}(n_s^{-\frac{\min\{\alpha, \nu, \varsigma, \tau\}}{16(\alpha+d+1)}}) - \mathcal{O}(\frac{1}{\sqrt{\min_k n_t(k)}})$ for $n_s$ sufficiently large.*

**Provable Advantages of ACT** Theorem 3.9 demonstrates how the abundance of unlabeled data in the source domain leveraged by ACT benefits downstream tasks in the target domain. Specifically, the quantity $\max_{i \neq j} |\mu_t(i)^\top \mu_t(j)|$ in eq (9) reflects the angle between different representation

centers when $R_1 \approx R_2$. A smaller value indicates that the centers approach orthogonality, enhancing discriminability among categories and thus improving classification accuracy. eq (9) essentially indicates that minimaxing the loss function of ACT can lead to the downstream data distribution being clustered in the representation space, provided that the upstream unlabeled sample size is sufficient. On the other hand, the theorem shows that only the failure probability depends on the downstream sample size $n_t$ with fast convergence rate, while the misclassification rate converges with respect to the number of unlabeled samples in ACT. This finding indicates that the build classifier based on ACT can achieve excellent performance with a few labeled samples. In summary, this theorem not only demonstrates the provable advantages of ACT but also provides rigorous theoretical understandings for few-shot learning (Liu et al., 2021; Rizve et al., 2021; Yang et al., 2022; Lim et al., 2023).

**Over-parametrization** Theorem 3.9 does not impose an upper bound constraint on either the width $W$ or depth $L$ of the deep neural network, implying that the number of network parameters can grow arbitrarily large when only the weight norm is constrained. This aligns with the over-parameterization regime commonly concerned in deep learning.

## 4. Comparison with Existing Methods

As the experiments conducted in existing self-supervised learning methods, we pretrain the representation on CIFAR-10, CIFAR-100 and Tiny ImageNet, and subsequently conduct fine-tuning on each dataset with annotations. Table 2 shows the classification accuracy of representations learned by ACT, compared with the results reported in Ermolov et al. (2021). We can see that ACT consistently outperforms previous mainstream self-supervised methods across various datasets and evaluation metrics.

*Table 2.* Classification accuracy (top 1) of a linear classifier and a 5-nearest neighbors classifier for different loss functions and datasets.

| Method | CIFAR-10 | | CIFAR-100 | | Tiny ImageNet | |
|---|---|---|---|---|---|---|
| | Linear | $k$-nn | Linear | $k$-nn | Linear | $k$-nn |
| SimCLR | 91.80 | 88.42 | 66.83 | 56.56 | 48.84 | 32.86 |
| BYOL | 91.73 | 89.45 | 66.60 | 56.82 | **51.00** | 36.24 |
| WMSE2 | 91.55 | 89.69 | 66.10 | 56.69 | 48.20 | 34.16 |
| WMSE4 | 91.99 | 89.87 | 67.64 | 56.45 | 49.20 | 35.44 |
| ACT | **92.11** | **90.01** | **68.24** | **58.35** | 49.72 | **36.40** |

**Implementation details.** Except for tuning $\lambda$ for different datasets, all other hyperparameters used in our experiments align with Ermolov et al. (2021). Before each iteration, we first standardize the representations and then calculate the

loss of ACT. We train for 1,000 epochs with a learning rate of $3 \times 10^{-3}$ for CIFAR-10 and CIFAR-100, and $2 \times 10^{-3}$ for Tiny ImageNet. A learning rate warm-up is applied for the first 500 iterations of the optimizer, in addition to a 0.2 learning rate drop at 50 and 25 epochs before the training end. We use a mini-batch size of 256, and the dimension of the hidden layer in the projection head is set to 1024. The weight decay is set to $10^{-6}$. We adopt an embedding size ($d^*$) of 64 for CIFAR10, CIFAR100 and 128 for Tiny ImageNet during the pretraining process. The backbone network used in our implementation is ResNet-18.

**Image transformation details.** We randomly extract crops with sizes ranging from 0.08 to 1.0 of the original area and aspect ratios ranging from $3/4$ to $4/3$ of the original aspect ratio. Furthermore, we apply horizontal mirroring with a probability of 0.5. Additionally, color jittering is applied with a configuration of $(0.4; 0.4; 0.4; 0.1)$ and a probability of 0.8, while grayscaling is applied with a probability of 0.2. For CIFAR-10 and CIFAR-100, random Gaussian blurring is adopted with a probability of 0.5 and a kernel size of 0.1. During testing, only one crop is used for evaluation.

**Evaluation protocol.** During evaluation, we freeze the network encoder and remove the projection head after pretraining, then train a supervised linear classifier on top of it, which is a fully-connected layer followed by softmax. we train the linear classifier for 500 epochs using the Adam optimizer with corresponding labeled training set without data augmentation. The learning rate is exponentially decayed from $10^{-2}$ to $10^{-6}$. The weight decay is set as $10^{-6}$. we also include the accuracy of a k-nearest neighbors classifier with $k = 5$, which does not require fine tuning.

All experiments were conducted using a single Tesla V100 GPU unit. The PyTorch implementations can be found in https://anonymous.4open.science/r/ACT-1F45.

## 5. Conclusion

In this paper, we propose a novel adversarial contrastive learning method for unsupervised transfer learning. Our experimental results achieved state-of-the-art classification accuracy under both fine-tuned linear probe and $k$-nn protocol on various real datasets, comparing with the self-supervised learning methods. Meanwhile, we present end to end theoretical guarantee for the downstream classification task under misspecified and over-parameterized setting. Our theoretical results not only indicate that the misclassification rate of downstream task solely depends on the strength of data augmentation on the large amount of unlabeled data, but also bridge the gap in the theoretical understanding of the effectiveness of few-shot learning for downstream tasks with small sample size.

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

# A. Deferred Proof

**A.1. $\mathcal{K}$-Lipschitz property of $\mathcal{NN}_{d_1,d_2}(W, L, \mathcal{K}, B_1, B_2)$**

*Proof.* To demonstrate that any function $f_{\boldsymbol{\theta}} \in \mathcal{NN}_{d_1,d_2}(W, L, \mathcal{K}, B_1, B_2)$ is a $\mathcal{K}$-Lipschitz function, we first define two special classes. The first class is given by

$$\mathcal{NN}_{d_1,d_2}(W, L, \mathcal{K}) := \{f_{\boldsymbol{\theta}}(\boldsymbol{x}) = A_L \sigma(A_{L-1} \sigma(\cdots \sigma(A_0 \boldsymbol{x})) : \kappa(\boldsymbol{\theta}) \leq \mathcal{K}\}, \tag{10}$$

which is equivalent to $\mathcal{NN}_{d_1,d_2}(W, L, \mathcal{K}, B_1, B_2)$ when ignoring the condition $\|f_{\boldsymbol{\theta}}\|_2 \in [B_1, B_2]$. The second class is defined as

$$\mathcal{SNN}_{d_1,d_2}(W, L, \mathcal{K}) := \{\breve{f}(\boldsymbol{x}) = \breve{A}_L \sigma(\breve{A}_{L-1} \sigma(\cdots \sigma(\breve{A}_0 \breve{\boldsymbol{x}})) : \prod_{l=1}^{L} \|\breve{A}_l\|_\infty \leq \mathcal{K}\}, \quad \breve{\boldsymbol{x}} := \begin{pmatrix} \boldsymbol{x} \\ 1 \end{pmatrix},$$

where $\breve{A}_l \in \mathbb{R}^{N_{l+1} \times N_l}$ with $N_0 = d_1 + 1$.

It is clear that $\mathcal{NN}_{d_1,d_2}(W, L, \mathcal{K}, B_1, B_2) \subseteq \mathcal{NN}_{d_1,d_2}(W, L, \mathcal{K})$, and every element in $\mathcal{SNN}_{d_1,d_2}(W, L, \mathcal{K})$ is a $\mathcal{K}$-Lipschitz function due to the 1-Lipschitz property of the ReLU activation function. Thus, it suffices to show that

$$\mathcal{SNN}_{d_1,d_2}(W, L, \mathcal{K}) \subseteq \mathcal{NN}_{d_1,d_2}(W, L, \mathcal{K}) \subseteq \mathcal{SNN}_{d_1,d_2}(W + 1, L, \mathcal{K})$$

to establish our claim.

To begin, any function $f_{\boldsymbol{\theta}}(\boldsymbol{x}) = A_L \sigma(A_{L-1} \sigma(\cdots \sigma(A_0 \boldsymbol{x} + \boldsymbol{b}_0)) + \boldsymbol{b}_{L-1}) \in \mathcal{NN}_{d_1,d_2}(W, L, \mathcal{K})$ can be restructured as $\breve{f}(\boldsymbol{x}) = \breve{A}_L \sigma(\breve{A}_{L-1} \sigma(\cdots \sigma(\breve{A}_0 \breve{\boldsymbol{x}})))$, where

$$\breve{\boldsymbol{x}} := \begin{pmatrix} \boldsymbol{x} \\ 1 \end{pmatrix}, \quad \breve{A} = (A_L, \boldsymbol{0}), \quad \breve{A}_l = \begin{pmatrix} A_l & \boldsymbol{b}_l \\ \boldsymbol{0} & 1 \end{pmatrix}, \quad l = 0, \ldots, L - 1.$$

Notably, we have $\prod_{l=0}^{L} \|\breve{A}_l\|_\infty = \|A_L\|_\infty \prod_{l=0}^{L-1} \max\{\|(A_l, \boldsymbol{b}_l)\|_\infty, 1\} = \kappa(\boldsymbol{\theta}) \leq \mathcal{K}$, which implies that $f_{\boldsymbol{\theta}} \in \mathcal{SNN}_{d_1,d_2}(W + 1, L, \mathcal{K})$.

Conversely, since any $\breve{f} \in \mathcal{SNN}(W, L, \mathcal{K})$ can also be parameterized as $A_L \sigma(A_{L-1} \sigma(\cdots \sigma(A_0 \boldsymbol{x} + \boldsymbol{b}_0)) + \boldsymbol{b}_{L-1})$ with $\boldsymbol{\theta} = (\breve{A}_0, (\breve{A}_1, \boldsymbol{0}), \ldots, (\breve{A}_{L-1}, \boldsymbol{0}), \breve{A}_L)$, we can use the absolute homogeneity of the ReLU function to rescale $\breve{A}_l$ such that $\|\breve{A}_L\|_\infty \leq \mathcal{K}$ and $\|\breve{A}_l\|_\infty = 1$ for $l \neq L$. Consequently, we have $\kappa(\boldsymbol{\theta}) = \prod_{l=0}^{L} \|\breve{A}_l\|_\infty \leq \mathcal{K}$, which yields $\breve{f} \in \mathcal{NN}(W, L, \mathcal{K})$. This completes the proof. $\qquad\square$

## A.2. Proof of population theorem

In this section, we aim to present the population theorem of ACT and its proof. we begin by exploring the sufficient condition for achieving small $\text{Err}(Q_f)$ in A.2.1. Following that, we build the connection between the required condition and optimizing our adversarial self-supervised learning loss in Lemma A.4 of A.2.3, it reveals that small value of $\mathcal{L}(f)$ may induce significant class divergence and highly augmented concentration. Lastly, by combining Lemma A.1 and Lemma A.4, we present the population theorem as Theorem A.5.

### A.2.1. SUFFICIENT CONDITION OF SMALL MISCLASSIFICATION RATE

**Lemma A.1.** *Given a $(\sigma_s, \sigma_t, \delta_s, \delta_t)$-augmentation, if the encoder $f$ such that $B_1 \leq \|f\|_2 \leq B_2$ is $\mathcal{K}$-Lipschitz and*

$$\mu_t(i)^\top \mu_t(j) < B_2^2 \psi(\sigma_t, \delta_t, \varepsilon, f),$$

*holds for any pair of $(i, j)$ with $i \neq j$, then the downstream error rate of $Q_f$*

$$\text{Err}(Q_f) \leq (1 - \sigma_t) + R_t(\varepsilon, f),$$

*where $\Delta_{\hat{\mu}_t} = 1 - \frac{\min_{k \in [K]} \|\hat{\mu}_t(k)\|_2^2}{B_2^2}$. For any $\varepsilon > 0$, $R_t(\varepsilon, f) = \mathbb{P}_t\big(\boldsymbol{z} \in \cup_{k=1}^{K} C_t(k) : \sup_{\boldsymbol{z}_1, \boldsymbol{z}_2 \in \mathcal{A}(\boldsymbol{z})} \|f(\boldsymbol{z}_1) - f(\boldsymbol{z}_2)\|_2 > \varepsilon\big)$ and $\psi(\sigma_t, \delta_t, \varepsilon, f) = \Gamma_{\min}(\sigma_t, \delta_t, \varepsilon, f) - \sqrt{2 - 2\Gamma_{\min}(\sigma_t, \delta_t, \varepsilon, f)} - \frac{\Delta_{\hat{\mu}_t}}{2} - \frac{2\max_{k \in [K]} \|\hat{\mu}_t(k) - \mu_t(k)\|_2}{B_2}$, wherein $\Gamma_{\min}(\sigma_t, \delta_t, \varepsilon, f) = \Big(\sigma_t - \frac{R_t(\varepsilon, f)}{\min_i p_t(i)}\Big)\Big(1 + \big(\frac{B_1}{B_2}\big)^2 - \frac{\mathcal{K}\delta_t}{B_2} - \frac{2\varepsilon}{B_2}\Big) - 1$.*

*Proof.* For any encoder $f$, let $S_t(\varepsilon, f) := \{z \in \cup_{k=1}^{K} C_t(k) : \sup_{z_1, z_2 \in \mathcal{A}(z)} \|f(z_1) - f(z_2)\|_2 \leq \varepsilon\}$, if any $z \in \{\widetilde{C}_t(1) \cup \cdots \cup \widetilde{C}_t(K)\} \cap S_t(\varepsilon, f)$ can be correctly classified by $Q_f$, it turns out that $\mathrm{Err}(Q_f)$ can be bounded by $(1 - \sigma_t) + R_t(\varepsilon, f)$. In fact,

$$\mathrm{Err}(Q_f) = \sum_{k=1}^{K} \mathbb{P}_t\{Q_f(z) \neq k, \forall z \in C_t(k)\} \leq \mathbb{P}_t\Big[\{\widetilde{C}_t(1) \cup \cdots \cup \widetilde{C}_t(K) \cap S_t(\varepsilon, f)\}^c\Big]$$

$$= \mathbb{P}_t\big[\{\widetilde{C}_t(1) \cup \cdots \cup \widetilde{C}_t(K)\}^c \cup \{S_t(\varepsilon, f)\}^c\big] \leq (1 - \sigma_t) + \mathbb{P}_t\big[\{S_t(\varepsilon, f)\}^c\big]$$

$$= (1 - \sigma_t) + R_t(\varepsilon, f).$$

The first row is derived from the definition of $\mathrm{Err}(Q_f)$. Since any $z \in \{\widetilde{C}_t(1) \cup \cdots \cup \widetilde{C}_t(K)\} \cap S_t(\varepsilon, f)$ can be correctly classified by $Q_f$, we obtain the second row. De Morgan's laws imply the third row. The fourth row follows from Definition 3.6. Finally, noting that $R_t(\varepsilon, f) = \mathbb{P}_t[\{S_t(\varepsilon, f)\}^c]$ yields the last line.

Hence it suffices to show for given $i \in [K]$, $z \in \widetilde{C}_t(i) \cap S_t(\varepsilon, f)$ can be correctly classified by $Q_f$ if for any $j \neq i$,

$$\mu_t(i)^\top \mu_t(j) < B_2^2\Big(\Gamma_i(\sigma_t, \delta_t, \varepsilon, f) - \sqrt{2 - 2\Gamma_i(\sigma_t, \delta_t, \varepsilon, f)} - \frac{\Delta_{\hat{\mu}_t}}{2} - \frac{\|\hat{\mu}_t(i) - \mu_t(i)\|_2}{B_2} - \frac{\|\hat{\mu}_t(j) - \mu_t(j)\|_2}{B_2}\Big),$$

where $\Gamma_i(\sigma_t, \delta_t, \varepsilon, f) = \Big(\sigma_t - \frac{R_t(\varepsilon, f)}{p_t(i)}\Big)\Big(1 + \big(\frac{B_1}{B_2}\big)^2 - \frac{K\delta_t}{B_2} - \frac{2\varepsilon}{B_2}\Big) - 1$.

To this end, without losing generality, consider the case $i = 1$. To turn out $z_0 \in \widetilde{C}_t(1) \cap S_t(\varepsilon, f)$ can be correctly classified by $Q_f$, by the definition of $\widetilde{C}_t(1)$ and $S_t(\varepsilon, f)$, It just need to show $\forall k \neq 1, \|f(z_0) - \hat{\mu}_t(1)\|_2 < \|f(z_0) - \hat{\mu}_t(k)\|_2$, which is equivalent to

$$f(z_0)^\top \hat{\mu}_t(1) - f(z_0)^\top \hat{\mu}_t(k) - \Big(\frac{1}{2}\|\hat{\mu}_t(1)\|_2^2 - \frac{1}{2}\|\hat{\mu}_t(k)\|_2^2\Big) > 0.$$

We first deal with the term $f(z_0)^\top \hat{\mu}_t(1)$,

$$f(z_0)^\top \hat{\mu}_t(1) = f(z_0)^\top \mu_t(1) + f(z_0)^\top(\hat{\mu}_t(1) - \mu_t(1))$$

$$\geq f(z_0)^\top \mathbb{E}_{z \in C_t(1)} \mathbb{E}_{z' \in \mathcal{A}(z)}\{f(z')\} - \|f(z_0)\|_2 \|\hat{\mu}_t(1) - \mu_t(1)\|_2$$

$$\geq \frac{1}{p_t(1)} f(z_0)^\top \mathbb{E}_z \mathbb{E}_{z' \in \mathcal{A}(z)}[f(z')\mathbb{1}\{z \in C_t(1)\}] - B_2 \|\hat{\mu}_t(1) - \mu_t(1)\|_2$$

$$= \frac{1}{p_t(1)} f(z_0)^\top \mathbb{E}_z \mathbb{E}_{z' \in \mathcal{A}(z)}\big[f(z')\mathbb{1}\{z \in C_t(1) \cap \widetilde{C}_t(1) \cap S_t(\varepsilon, f)\}\big]$$

$$\quad + \frac{1}{p_t(1)} f(z_0)^\top \mathbb{E}_z \mathbb{E}_{z' \in \mathcal{A}(z)}\big[f(z')\mathbb{1}\{z \in C_t(1) \cap \{\widetilde{C}_t(1) \cap S_t(\varepsilon, f)\}^c\}\big] - B_2 \|\hat{\mu}_t(1) - \mu_t(1)\|_2$$

$$= \frac{\mathbb{P}_t\{\widetilde{C}_t(1) \cap S_t(\varepsilon, f)\}}{p_t(1)} f(z_0)^\top \mathbb{E}_{z \in \widetilde{C}_t(1) \cap S_t(\varepsilon, f)} \mathbb{E}_{z' \in \mathcal{A}(z)}\{f(z')\}$$

$$\quad + \frac{1}{p_t(1)} \mathbb{E}_z\big[\mathbb{E}_{z' \in \mathcal{A}(z)}\{f(z_0)^\top f(z')\}\mathbb{1}[z \in C_t(1)\backslash\{\widetilde{C}_t(1) \cap S_t(\varepsilon, f)\}]\big] - B_2 \|\hat{\mu}_t(1) - \mu_t(1)\|_2$$

$$\geq \frac{\mathbb{P}_t\{\widetilde{C}_t(1) \cap S_t(\varepsilon, f)\}}{p_t(1)} f(z_0)^\top \mathop{\mathbb{E}}_{z \in \widetilde{C}_t(1) \cap S_t(\varepsilon, f)} \mathop{\mathbb{E}}_{z' \in \mathcal{A}(z)} [f(z')] - \frac{B_2^2}{p_t(1)} \mathbb{P}_t\big[C_t(1)\backslash\{\widetilde{C}_t(1) \cap S_t(\varepsilon, f)\}\big]$$

$$\quad - B_2 \|\hat{\mu}_t(1) - \mu_t(1)\|_2. \tag{11}$$

The second row follows from the Cauchy–Schwarz inequality. The third and last rows are derived from the condition $\|f\|_2 \leq B_2$. Note that

$$\mathbb{P}_t\big[C_t(1)\backslash\{\widetilde{C}_t(1) \cap S_t(\varepsilon, f)\}\big] = \mathbb{P}_t\big[\{C_t(1)\backslash\widetilde{C}_t(1)\} \cup [\widetilde{C}_t(1) \cap \{S_t(\varepsilon, f)\}^c]\big] \leq (1 - \sigma_t)p_t(1) + R_t(\varepsilon, f), \tag{12}$$

and

$$\mathbb{P}_t\big(\widetilde{C}_t(1) \cap S_t(\varepsilon, f)\big) = \mathbb{P}_t(C_t(1)) - \mathbb{P}_t\big(C_t(1)\backslash(\widetilde{C}_t(1) \cap S_t(\varepsilon, f))\big) \geq p_t(1) - \{(1 - \sigma_t)p_t(1) + R_t(\varepsilon, f)\}$$

$$= \sigma_t p_t(1) - R_t(\varepsilon, f). \tag{13}$$

Plugging (12) and (13) into (11) yields

$$f(\boldsymbol{z}_0)^\top \hat{\mu}_t(1) \geq \Big(\sigma_t - \frac{R_t(\varepsilon, f)}{p_t(1)}\Big) f(\boldsymbol{z}_0)^\top \mathop{\mathbb{E}}_{\boldsymbol{z} \in \widetilde{C}_t(1) \cap S_t(\varepsilon, f)} \mathop{\mathbb{E}}_{\boldsymbol{z}' \in \mathcal{A}(\boldsymbol{z})} \{f(\boldsymbol{z}')\} - B_2^2\Big(1 - \sigma_t + \frac{R_t(\varepsilon, f)}{p_t(1)}\Big) - B_2\|\hat{\mu}_t(1) - \mu_t(1)\|_2.$$
$$\tag{14}$$

Notice that $\boldsymbol{z}_0 \in \widetilde{C}_t(1) \cap S_t(\varepsilon, f)$. Thus, for any $\boldsymbol{z} \in \widetilde{C}_t(1) \cap S_t(\varepsilon, f)$, by the definition of $\widetilde{C}_t(1)$, we have $\min_{\boldsymbol{z}_0' \in \mathcal{A}(\boldsymbol{z}_0), \boldsymbol{z}' \in \mathcal{A}(\boldsymbol{z})} \|\boldsymbol{z}_0' - \boldsymbol{z}'\|_2 \leq \delta_t$. Further, denote $(\boldsymbol{z}_0^*, \boldsymbol{z}^*) = \arg\min_{\boldsymbol{z}_0' \in \mathcal{A}(\boldsymbol{z}_0), \boldsymbol{z}' \in \mathcal{A}(\boldsymbol{z})} \|\boldsymbol{z}_0' - \boldsymbol{z}'\|_2$. Then, we have $\|\boldsymbol{z}_0^* - \boldsymbol{z}^*\|_2 \leq \delta_t$. Combining this with the $\mathcal{K}$-Lipschitz property of $f$, we obtain $\|f(\boldsymbol{z}_0^*) - f(\boldsymbol{z}^*)\|_2 \leq \mathcal{K}\|\boldsymbol{z}_0^* - \boldsymbol{z}^*\|_2 \leq \mathcal{K}\delta_t$. Moreover, since $\boldsymbol{z} \in S_t(\varepsilon, f)$, it follows that for all $\boldsymbol{z}' \in \mathcal{A}(\boldsymbol{z})$, $\|f(\boldsymbol{z}') - f(\boldsymbol{z}^*)\|_2 \leq \varepsilon$. Similarly, as $\boldsymbol{z}_0 \in S_t(\varepsilon, f)$ and both $\boldsymbol{z}_0$ and $\boldsymbol{z}_0^*$ belong to $\mathcal{A}(\boldsymbol{z}_0)$, we know $\|f(\boldsymbol{z}_0) - f(\boldsymbol{z}_0^*)\|_2 \leq \varepsilon$.

Therefore,

$$
\begin{aligned}
&f(\boldsymbol{z}_0)^\top \mathop{\mathbb{E}}_{\boldsymbol{z} \in \widetilde{C}_t(1) \cap S_t(\varepsilon, f)} \mathop{\mathbb{E}}_{\boldsymbol{z}' \in \mathcal{A}(\boldsymbol{z})} \{f(\boldsymbol{z}')\} = \mathop{\mathbb{E}}_{\boldsymbol{z} \in \widetilde{C}_t(1) \cap S_t(\varepsilon, f)} \mathop{\mathbb{E}}_{\boldsymbol{z}' \in \mathcal{A}(\boldsymbol{z})} \{f(\boldsymbol{z}_0)^\top f(\boldsymbol{z}')\} \\
&= \mathop{\mathbb{E}}_{\boldsymbol{z} \in \widetilde{C}_t(1) \cap S_t(\varepsilon, f)} \mathop{\mathbb{E}}_{\boldsymbol{z}' \in \mathcal{A}(\boldsymbol{z})} [f(\boldsymbol{z}_0)^\top \{f(\boldsymbol{z}') - f(\boldsymbol{z}_0) + f(\boldsymbol{z}_0)\}] \\
&\geq B_1^2 + \mathop{\mathbb{E}}_{\boldsymbol{z} \in \widetilde{C}_t(1) \cap S_t(\varepsilon, f)} \mathop{\mathbb{E}}_{\boldsymbol{z}' \in \mathcal{A}(\boldsymbol{z})} [f(\boldsymbol{z}_0)^\top \{f(\boldsymbol{z}') - f(\boldsymbol{z}_0)\}] \\
&= B_1^2 + \mathop{\mathbb{E}}_{\boldsymbol{z} \in \widetilde{C}_t(1) \cap S_t(\varepsilon, f)} \mathop{\mathbb{E}}_{\boldsymbol{z}' \in \mathcal{A}(\boldsymbol{z})} [f(\boldsymbol{z}_0)^\top \{\underbrace{f(\boldsymbol{z}') - f(\boldsymbol{z}^*)}_{\|\cdot\|_2 \leq \varepsilon} + \underbrace{f(\boldsymbol{z}^*) - f(\boldsymbol{z}_0^*)}_{\|\cdot\|_2 \leq \mathcal{K}\delta_t} + \underbrace{f(\boldsymbol{z}_0^*) - f(\boldsymbol{z}_0)}_{\|\cdot\|_2 \leq \varepsilon}\}] \\
&\geq B_1^2 - (B_2\varepsilon + B_2\mathcal{K}\delta_t + B_2\varepsilon) \\
&= B_1^2 - B_2(\mathcal{K}\delta_t + 2\varepsilon), \tag{15}
\end{aligned}
$$

where the fourth row is derived from $\|f\|_2 \geq B_1$.

Plugging (15) into the inequality (14) yields

$$
\begin{aligned}
f(\boldsymbol{z}_0)^\top \hat{\mu}_t(1) &\geq \Big(\sigma_t - \frac{R_t(\varepsilon, f)}{p_t(1)}\Big) f(\boldsymbol{z}_0)^\top \mathop{\mathbb{E}}_{\boldsymbol{z} \in \widetilde{C}_t(1) \cap S_t(\varepsilon, f)} \mathop{\mathbb{E}}_{\boldsymbol{z}' \in \mathcal{A}(\boldsymbol{z})} \{f(\boldsymbol{z}')\} - B_2^2\Big(1 - \sigma_t + \frac{R_t(\varepsilon, f)}{p_t(1)}\Big) - B_2\|\hat{\mu}_t(1) - \mu_t(1)\|_2 \\
&\geq \Big(\sigma_t - \frac{R_t(\varepsilon, f)}{p_t(1)}\Big)\big(B_1^2 - B_2(\mathcal{K}\delta_t + 2\varepsilon)\big) - B_2^2\Big\{1 - \sigma_t + \frac{R_t(\varepsilon, f)}{p_t(1)}\Big\} - B_2\|\hat{\mu}_t(1) - \mu_t(1)\|_2 \\
&= B_2^2\Big\{\Big(1 + \big(\frac{B_1}{B_2}\big)^2\Big)\Big(\sigma_t - \frac{R_t(\varepsilon, f)}{p_t(1)}\Big) - \Big(\sigma_t - \frac{R_t(\varepsilon, f)}{p_t(1)}\Big)\Big(\frac{\mathcal{K}\delta_t}{B_2} + \frac{2\varepsilon}{B_2}\Big) - 1\Big\} - B_2\|\hat{\mu}_t(1) - \mu_t(1)\|_2 \\
&= B_2^2\Big\{\Big(\sigma_t - \frac{R_t(\varepsilon, f)}{p_t(1)}\Big)\Big(1 + \big(\frac{B_1}{B_2}\big)^2 - \frac{\mathcal{K}\delta_t}{B_2} - \frac{2\varepsilon}{B_2}\Big) - 1\Big\} - B_2\|\hat{\mu}_t(1) - \mu_t(1)\|_2 \\
&= B_2^2\Gamma_1(\sigma_t, \delta_t, \varepsilon, f) - B_2\|\hat{\mu}_t(1) - \mu_t(1)\|_2.
\end{aligned}
$$

Similar process can also turn out

$$f(\boldsymbol{z}_0)^\top \mu_t(1) \geq B_2^2\Gamma_1(\sigma_t, \delta_t, \varepsilon, f). \tag{16}$$

Combining the fact that $\|\mu_t(k)\|_2 = \|\mathbb{E}_{\boldsymbol{z} \in \widetilde{C}_t(k)} \mathbb{E}_{\boldsymbol{z}' \in \mathcal{A}(\boldsymbol{z})}[f(\boldsymbol{z}')]\|_2 \leq \mathbb{E}_{\boldsymbol{x} \in \widetilde{C}_t(k)} \mathbb{E}_{\boldsymbol{z}' \in \mathcal{A}(\boldsymbol{z})}\|f(\boldsymbol{z}')\|_2 \leq B_2$ yields

$$
\begin{aligned}
f(\boldsymbol{z}_0)^\top \hat{\mu}_t(k) &\leq f(\boldsymbol{z}_0)^\top \mu_t(k) + f(\boldsymbol{z}_0)^\top (\hat{\mu}_t(k) - \mu_t(k)) \\
&\leq f(\boldsymbol{z}_0)^\top \mu_t(k) + \|f(\boldsymbol{z}_0)\|_2 \|\hat{\mu}_t(k) - \mu_t(k)\|_2 \\
&\leq f(\boldsymbol{z}_0)^\top \mu_t(k) + B_2\|\hat{\mu}_t(k) - \mu_t(k)\|_2 \\
&= \{f(\boldsymbol{z}_0) - \mu_t(1)\}^\top \mu_t(k) + \mu_t(1)^\top \mu_t(k) + B_2\|\hat{\mu}_t(k) - \mu_t(k)\|_2 \\
&\leq \|f(\boldsymbol{z}_0) - \mu_t(1)\|_2 \cdot \|\mu_t(k)\|_2 + \mu_t(1)^\top \mu_t(k) + B_2\|\hat{\mu}_t(k) - \mu_t(k)\|_2 \\
&\leq B_2\sqrt{\|f(\boldsymbol{z}_0)\|_2^2 - 2f(\boldsymbol{z}_0)^\top \mu_t(1) + \|\mu_t(1)\|_2^2} + \mu_t(1)^\top \mu_t(k) + B_2\|\hat{\mu}_t(k) - \mu_t(k)\|_2
\end{aligned}
$$

$$\leq B_2\sqrt{2B_2^2 - 2f(\boldsymbol{z}_0)^\top\mu_t(1) + \mu_t(1)^\top\mu_t(k)} + B_2\|\hat{\mu}_t(k) - \mu_t(k)\|_2$$

$$\leq B_2\sqrt{2B_2^2 - 2B_2^2\Gamma_1(\sigma_t, \delta_t, \varepsilon, f)} + \mu_t(1)^\top\mu_t(k) + B_2\|\hat{\mu}_t(k) - \mu_t(k)\|_2$$

$$= \sqrt{2}B_2^2\sqrt{1 - \Gamma_1(\sigma_t, \delta_t, \varepsilon, f)} + \mu_t(1)^\top\mu_t(k) + B_2\|\hat{\mu}_t(k) - \mu_t(k)\|_2,$$

where the inequality in eighth row stems from (16). Hence, by $\Delta_{\hat{\mu}_t} = 1 - \min_{k\in[K]}\|\hat{\mu}_t(k)\|_2^2/B_2^2$, we can conclude

$$f(\boldsymbol{z}_0)^\top\hat{\mu}_t(1) - f(\boldsymbol{z}_0)^\top\hat{\mu}_t(k) - \Big(\frac{1}{2}\|\hat{\mu}_t(1)\|_2^2 - \frac{1}{2}\|\hat{\mu}_t(k)\|^2\Big) = f(\boldsymbol{z}_0)^\top\hat{\mu}_t(1) - f(\boldsymbol{z}_0)^\top\hat{\mu}_t(k) - \frac{1}{2}\|\hat{\mu}_t(1)\|_2^2 + \frac{1}{2}\|\hat{\mu}_t(k)\|_2^2$$

$$\geq f(\boldsymbol{z}_0)^\top\hat{\mu}_t(1) - f(\boldsymbol{z}_0)^\top\hat{\mu}_t(k) - \frac{1}{2}B_2^2 + \frac{1}{2}\min_{k\in[K]}\|\hat{\mu}_t(k)\|_2^2$$

$$= f(\boldsymbol{z}_0)^\top\hat{\mu}_t(1) - f(\boldsymbol{z}_0)^\top\hat{\mu}_t(k) - \frac{1}{2}B_2^2\Delta_{\hat{\mu}_t}$$

$$\geq B_2^2\Gamma_1(\sigma_t, \delta_t, \varepsilon, f) - B_2\|\hat{\mu}_t(1) - \mu_t(1)\|_2 - \sqrt{2}B_2^2\sqrt{1 - \Gamma_1(\sigma_t, \delta_t, \varepsilon, f)} - \mu_t(1)^\top\mu_t(k) - B_2\|\hat{\mu}_t(k) - \mu_t(k)\|_2 -$$

$$- \frac{1}{2}B_2^2\Delta_{\hat{\mu}_t} > 0,$$

which is what we desire. Here the last inequality is derived from the given condition. $\qquad\square$

### A.2.2. PRELIMINARIES FOR LEMMA A.4

To obtain Theorem A.5, we need to bridge the gap between the condition in Lemma A.1 and the insights provided by ACT in Lemma A.4. To this end, we first introduce Lemma A.2 and Lemma A.3.

Following the notations in the target domain, we denote the center of the $k$-th latent class in the representation space as $\mu_s(k) := \mathbb{E}_{\boldsymbol{x}\in C_s(k)}\mathbb{E}_{\boldsymbol{x}'\in\mathcal{A}(\boldsymbol{x})}\{f(\boldsymbol{x}')\} = \frac{1}{p_s(k)}\mathbb{E}_{\boldsymbol{x}}\mathbb{E}_{\boldsymbol{x}'\in\mathcal{A}(\boldsymbol{x})}[f(\boldsymbol{x}')\mathbb{1}\{\boldsymbol{x}\in C_s(k)\}]$. Then Lemma A.2 can be presented as follows:

**Lemma A.2.** *If the encoder $f$ is $\mathcal{K}$-Lipschitz continuous, then for any $k\in[K]$,*

$$\|\mu_s(k) - \mu_t(k)\|_2 \leq \sqrt{d^*}M\mathcal{K}\epsilon_1.$$

*Proof.* For all $k\in[K]$,

$$\|\mu_s(k) - \mu_t(k)\|_2^2 = \sum_{l=1}^{d^*}\big[\{\mu_s(k)\}_l - \{\mu_t(k)\}_l\big]^2 = \sum_{l=1}^{d^*}[\mathbb{E}_{\boldsymbol{x}\in C_s(k)}\mathbb{E}_{\boldsymbol{x}'\in\mathcal{A}(\boldsymbol{x})}\{f_l(\boldsymbol{x}')\} - \mathbb{E}_{\boldsymbol{z}\in C_t(k)}\mathbb{E}_{\boldsymbol{z}'\in\mathcal{A}(\boldsymbol{z})}\{f_l(\boldsymbol{z}')\}]^2$$

$$= \sum_{l=1}^{d^*}\Big[\frac{1}{m}\sum_{\gamma=1}^{m}\Big(\mathbb{E}_{\boldsymbol{x}\in C_s(k)}\{f_l(A_i(\boldsymbol{x}))\} - \mathbb{E}_{\boldsymbol{z}\in C_t(k)}\{f_l(A_i(\boldsymbol{z}))\}\Big)\Big]^2 \leq d^*M^2\mathcal{K}^2\epsilon_1^2.$$

The final inequality is obtained from $\epsilon_1 = \max_{k\in[K]}\mathcal{W}(\mathbb{P}_s(k), \mathbb{P}_t(k))$ and the definition of Wasserstein distance, along with the fact that $f(A_i(\cdot))$ is $M\mathcal{K}$-Lipschitz continuous. In fact, since $f\in\mathrm{Lip}(\mathcal{K})$, it follows that for every $l\in[d^*]$, $f_l\in\mathrm{Lip}(\mathcal{K})$. Combining this with the property that $A_i(\cdot)\in\mathrm{Lip}(M)$ stated in Assumption 3.5, we conclude that $f(A_i(\cdot))$ is $M\mathcal{K}$-Lipschitz continuous. So that

$$\|\mu_s(k) - \mu_t(k)\|_2 \leq \sqrt{d^*}M\mathcal{K}\epsilon_1.$$

$\qquad\square$

Next we present Lemma A.3.

**Lemma A.3.** *Given a $(\sigma_s, \sigma_t, \delta_s, \delta_t)$-augmentation, if the encoder $f$ with $\|f\|_2 \leq B_2$ is $\mathcal{K}$-Lipschitz continuous, then*

$$\mathbb{E}_{\boldsymbol{x}\in C_s(k)}\mathbb{E}_{\boldsymbol{x}_1\in\mathcal{A}(\boldsymbol{x})}\|f(\boldsymbol{x}_1) - \mu_s(k)\|_2^2 \leq 4B_2^2\Big\{\Big(1 - \sigma_s + \frac{\mathcal{K}\delta_s + 2\varepsilon}{2B_2} + \frac{R_s(\varepsilon, f)}{p_s(k)}\Big)^2 + \Big(1 - \sigma_s + \frac{R_s(\varepsilon, f)}{p_s(k)}\Big)\Big\},$$

*where $R_s(\varepsilon, f) = \mathbb{P}_s\big\{\boldsymbol{x}\in\cup_{k=1}^K C_s(k) : \sup_{\boldsymbol{x}_1, \boldsymbol{x}_2\in\mathcal{A}(\boldsymbol{x})}\|f(\boldsymbol{x}_1) - f(\boldsymbol{x}_2)\|_2 > \varepsilon\big\}$.*

*Proof.* Let $S_s(\varepsilon, f) := \{\boldsymbol{x} \in \cup_{k=1}^K C_s(k) : \sup_{\boldsymbol{x}_1, \boldsymbol{x}_2 \in \mathcal{A}(\boldsymbol{x})} \|f(\boldsymbol{x}_1) - f(\boldsymbol{x}_2)\|_2 \le \varepsilon\}$, for each $k \in [K]$,

$$\mathbb{E}_{\boldsymbol{x} \in C_s(k)} \mathbb{E}_{\boldsymbol{x}_1 \in \mathcal{A}(\boldsymbol{x})} \|f(\boldsymbol{x}_1) - \mu_s(k)\|_2^2 = \frac{1}{p_s(k)} \mathbb{E}_{\boldsymbol{x}} \mathbb{E}_{\boldsymbol{x}_1 \in \mathcal{A}(\boldsymbol{x})} [\mathbb{1}\{\boldsymbol{x} \in C_s(k)\} \|f(\boldsymbol{x}_1) - \mu_s(k)\|_2^2]$$

$$= \frac{1}{p_s(k)} \mathbb{E}_{\boldsymbol{x}} \mathbb{E}_{\boldsymbol{x}_1 \in \mathcal{A}(\boldsymbol{x})} [\mathbb{1}\{\boldsymbol{x} \in \widetilde{C}_s(k) \cap S_s(\varepsilon, f)\} \|f(\boldsymbol{x}_1) - \mu_s(k)\|_2^2]$$

$$+ \frac{1}{p_s(k)} \mathbb{E}_{\boldsymbol{x}} \mathbb{E}_{\boldsymbol{x}_1 \in \mathcal{A}(\boldsymbol{x})} [\mathbb{1}\{\boldsymbol{x} \in C_s(k) \backslash (\widetilde{C}_s(k) \cap S_s(\varepsilon, f))\} \|f(\boldsymbol{x}_1) - \mu_s(k)\|_2^2]$$

$$\le \frac{1}{p_s(k)} \mathbb{E}_{\boldsymbol{x}} \mathbb{E}_{\boldsymbol{x}_1 \in \mathcal{A}(\boldsymbol{x})} \left[ \mathbb{1}\{\boldsymbol{x} \in \widetilde{C}_s(k) \cap S_s(\varepsilon, f)\} \|f(\boldsymbol{x}_1) - \mu_s(k)\|_2^2 + \frac{4B_2^2 \mathbb{P}_s [C_s(k) \backslash \{\widetilde{C}_s(k) \cap S_s(\varepsilon, f)\}]}{p_s(k)} \right.$$

$$\le \frac{1}{p_s(k)} \mathbb{E}_{\boldsymbol{x}} \mathbb{E}_{\boldsymbol{x}_1 \in \mathcal{A}(\boldsymbol{x})} \left[ \mathbb{1}\{\boldsymbol{x} \in \widetilde{C}_s(k) \cap S_s(\varepsilon, f)\} \|f(\boldsymbol{x}_1) - \mu_s(k)\|_2^2 + 4B_2^2 \left( 1 - \sigma_s + \frac{R_s(\varepsilon, f)}{p_s(k)} \right) \right.$$

$$\le \frac{\mathbb{P}_s(\widetilde{C}_s(k) \cap S_s(\varepsilon, f))}{p_s(k)} \mathbb{E}_{\boldsymbol{x} \in \widetilde{C}_s(k) \cap S_s(\varepsilon, f)} \mathbb{E}_{\boldsymbol{x}_1 \in \mathcal{A}(\boldsymbol{x})} \|f(\boldsymbol{x}_1) - \mu_s(k)\|_2^2 + 4B_2^2 \left( 1 - \sigma_s + \frac{R_s(\varepsilon, f)}{p_s(k)} \right)$$

$$\le \mathbb{E}_{\boldsymbol{x} \in \widetilde{C}_s(k) \cap S_s(\varepsilon, f)} \mathbb{E}_{\boldsymbol{x}_1 \in \mathcal{A}(\boldsymbol{x})} \|f(\boldsymbol{x}_1) - \mu_s(k)\|_2^2 + 4B_2^2 \left( 1 - \sigma_s + \frac{R_s(\varepsilon, f)}{p_s(k)} \right), \tag{17}$$

where the second inequality is due to

$$\mathbb{P}_s [C_s(k) \backslash \{\widetilde{C}_s(k) \cap S_s(\varepsilon, f)\}] = \mathbb{P}_s [\{C_s(k) \backslash \widetilde{C}_s(k)\} \cup \{C_s(k) \backslash S_s(\varepsilon, f)\}] \le (1 - \sigma_s) p_s(k) + R_s(\varepsilon, f).$$

Furthermore,

$$\mathbb{E}_{\boldsymbol{x} \in \widetilde{C}_s(k) \cap S_s(\varepsilon, f)} \mathbb{E}_{\boldsymbol{x}_1 \in \mathcal{A}(\boldsymbol{x})} \|f(\boldsymbol{x}_1) - \mu_s(k)\|_2^2 = \mathbb{E}_{\boldsymbol{x} \in \widetilde{C}_s(k) \cap S_s(\varepsilon, f)} \mathbb{E}_{\boldsymbol{x}_1 \in \mathcal{A}(\boldsymbol{x})} \|f(\boldsymbol{x}_1) - \mathbb{E}_{\boldsymbol{x}' \in C_s(k)} \mathbb{E}_{\boldsymbol{x}_2 \in \mathcal{A}(\boldsymbol{x}')} f(\boldsymbol{x}_2)\|_2^2$$

$$= \mathbb{E}_{\boldsymbol{x} \in \widetilde{C}_s(k) \cap S_s(\varepsilon, f)} \mathbb{E}_{\boldsymbol{x}_1 \in \mathcal{A}(\boldsymbol{x})} \left\| f(\boldsymbol{x}_1) - \frac{P\{\widetilde{C}_s(k) \cap S_s(\varepsilon, f)\}}{p_s(k)} \mathbb{E}_{\boldsymbol{x}' \in \widetilde{C}_s(k) \cap S_s(\varepsilon, f)} \mathbb{E}_{\boldsymbol{x}_2 \in \mathcal{A}(\boldsymbol{x}')} f(\boldsymbol{x}_2) \right.$$

$$\left. - \frac{\mathbb{P}_s [C_s(k) \backslash \{\widetilde{C}_s(k) \cap S_s(\varepsilon, f)\}]}{p_s(k)} \mathbb{E}_{\boldsymbol{x}' \in C_s(k) \backslash \{\widetilde{C}_s(k) \cap S_s(\varepsilon, f)\}} \mathbb{E}_{\boldsymbol{x}_2 \in \mathcal{A}(\boldsymbol{x}')} f(\boldsymbol{x}_2) \right\|_2^2$$

$$= \mathbb{E}_{\boldsymbol{x} \in \widetilde{C}_s(k) \cap S_s(\varepsilon, f)} \mathbb{E}_{\boldsymbol{x}_1 \in \mathcal{A}(\boldsymbol{x})} \left\| \frac{\mathbb{P}_s\{\widetilde{C}_s(k) \cap S_s(\varepsilon, f)\}}{p_s(k)} \left( f(\boldsymbol{x}_1) - \mathbb{E}_{\boldsymbol{x}' \in \widetilde{C}_s(k) \cap S_s(\varepsilon, f)} \mathbb{E}_{\boldsymbol{x}_2 \in \mathcal{A}(\boldsymbol{x}')} f(\boldsymbol{x}_2) \right) \right.$$

$$\left. - \frac{\mathbb{P}_s [C_s(k) \backslash \{\widetilde{C}_s(k) \cap S_s(\varepsilon, f)\}]}{p_s(k)} \left( f(\boldsymbol{x}_1) - \mathbb{E}_{\boldsymbol{x}' \in C_s(k) \backslash \{\widetilde{C}_s(k) \cap S_s(\varepsilon, f)\}} \mathbb{E}_{\boldsymbol{x}_2 \in \mathcal{A}(\boldsymbol{x}')} f(\boldsymbol{x}_2) \right) \right\|_2^2$$

$$\le \mathbb{E}_{\boldsymbol{x} \in \widetilde{C}_s(k) \cap S_s(\varepsilon, f)} \mathbb{E}_{\boldsymbol{x}_1 \in \mathcal{A}(\boldsymbol{x})} \left[ \left\| f(\boldsymbol{x}_1) - \mathbb{E}_{\boldsymbol{x}' \in \widetilde{C}_s(k) \cap S_s(\varepsilon, f)} \mathbb{E}_{\boldsymbol{x}_2 \in \mathcal{A}(\boldsymbol{x}')} f(\boldsymbol{x}_2) \right\|_2 + 2B_2 \left( 1 - \sigma_s + \frac{R_s(\varepsilon, f)}{p_s(k)} \right) \right]^2 \tag{18}$$

For any $\boldsymbol{x}, \boldsymbol{x}' \in \widetilde{C}_s(k) \cap S_s(\varepsilon, f)$, by the definition of $\widetilde{C}_s(k)$, we can yield that

$$\min_{\boldsymbol{x}_1 \in \mathcal{A}(\boldsymbol{x}), \boldsymbol{x}_2 \in \mathcal{A}(\boldsymbol{x}')} \|\boldsymbol{x}_1 - \boldsymbol{x}_2\|_2 \le \delta_s,$$

Thus, let $(\boldsymbol{x}_1^*, \boldsymbol{x}_2^*) = \arg \min_{\boldsymbol{x}_1 \in \mathcal{A}(\boldsymbol{x}), \boldsymbol{x}_2 \in \mathcal{A}(\boldsymbol{x}')} \|\boldsymbol{x}_1 - \boldsymbol{x}_2\|_2$, we have $\|\boldsymbol{x}_1^* - \boldsymbol{x}_2^*\|_2 \le \delta_s$. Furthermore, by the $\mathcal{K}$-Lipschitz continuity of $f$, we yield $\|f(\boldsymbol{x}_1^*) - f(\boldsymbol{x}_2^*)\|_2 \le \mathcal{K} \|\boldsymbol{x}_1^* - \boldsymbol{x}_2^*\|_2 \le \mathcal{K} \delta_s$. In addition, since $\boldsymbol{x} \in S_s(\varepsilon, f)$, we know for any $\boldsymbol{x}_1 \in \mathcal{A}(\boldsymbol{x}), \|f(\boldsymbol{x}_1) - f(\boldsymbol{x}_1^*)\|_2 \le \varepsilon$. Similarly, $\boldsymbol{x}' \in S_s(\varepsilon, f)$ implies $\|f(\boldsymbol{x}_2) - f(\boldsymbol{x}_2^*)\|_2 \le \varepsilon$ for any $\boldsymbol{x}_2 \in \mathcal{A}(\boldsymbol{x}')$. Therefore, for any $\boldsymbol{x}, \boldsymbol{x}' \in \widetilde{C}_s(1) \cap S_s(\varepsilon, f)$ and $\boldsymbol{x}_1 \in \mathcal{A}(\boldsymbol{x}), \boldsymbol{x}_2 \in \mathcal{A}(\boldsymbol{x}')$,

$$\|f(\boldsymbol{x}_1) - f(\boldsymbol{x}_2)\|_2 \le \|f(\boldsymbol{x}_1) - f(\boldsymbol{x}_1^*)\|_2 + \|f(\boldsymbol{x}_1^*) - f(\boldsymbol{x}_2^*)\|_2 + \|f(\boldsymbol{x}_2^*) - f(\boldsymbol{x}_2)\|_2 \le 2\varepsilon + \mathcal{K}\delta_s. \tag{19}$$

Combining inequalities (17), (18) and (19) concludes

$$\mathbb{E}_{\boldsymbol{x} \in C_s(k)} \mathbb{E}_{\boldsymbol{x}_1 \in \mathcal{A}(\boldsymbol{x})} \|f(\boldsymbol{x}_1) - \mu_s(k)\|_2^2 \le \left[ 2\varepsilon + \mathcal{K}\delta_s + 2B_2 \left( 1 - \sigma_s + \frac{R_s(\varepsilon, f)}{p_s(k)} \right) \right]^2 + 4B_2^2 \left( 1 - \sigma_s + \frac{R_s(\varepsilon, f)}{p_s(k)} \right)$$

$$= 4B_2^2 \left[ \left( 1 - \sigma_s + \frac{\mathcal{K}\delta_s}{2B_2} + \frac{\varepsilon}{B_2} + \frac{R_s(\varepsilon, f)}{p_s(k)} \right)^2 + \left( 1 - \sigma_s + \frac{R_s(\varepsilon, f)}{p_s(k)} \right) \right]$$

$\square$

Subsequently, we state Lemma A.4 to establish the connection between ACT and the requirements outlined in Lemma A.1.

### A.2.3. THE EFFECT OF MINIMAXING OUR LOSS

**Lemma A.4.** *Given a $(\sigma_s, \sigma_t, \delta_s, \delta_t)$-augmentation, if $d^* > K$ and the encoder $f$ with $B_1 \leq \|f\|_2 \leq B_2$ is $\mathcal{K}$-Lipschitz continuous, then for any $\varepsilon > 0$,*

$$R_s^2(\varepsilon, f) \leq \frac{m^4}{\varepsilon^2} \mathcal{L}_{\text{align}}(f),$$

$$R_t^2(\varepsilon, f) \leq \frac{m^4}{\varepsilon^2} \mathcal{L}_{\text{align}}(f) + \frac{8m^4}{\varepsilon^2} B_2 d^* M \mathcal{K} \epsilon_1 + \frac{4m^4}{\varepsilon^2} B_2^2 d^* K \epsilon_2,$$

*and*

$$\max_{i \neq j} |\mu_t(i)^\top \mu_t(j)| \leq \sqrt{\frac{2}{\min_{i \neq j} p_s(i) p_s(j)} \Big\{ \mathcal{R}(f) + \varphi(\sigma_s, \delta_s, \varepsilon, f) \Big\}} + 2\sqrt{d^*} B_2 M \mathcal{K} \epsilon_1.$$

*where $\varphi(\sigma_s, \delta_s, \varepsilon, f) := 4B_2^2 \Big[ \Big(1 - \sigma_s + \frac{\mathcal{K}\delta_s + 2\varepsilon}{2B_2}\Big)^2 + (1 - \sigma_s) + K R_s(\varepsilon, f)\Big(3 - 2\sigma_s + \frac{\mathcal{K}\delta_s + 2\varepsilon}{B_2}\Big) + R_s^2(\varepsilon, f)\Big(\sum_{k=1}^{K} \frac{1}{p_s(k)}\Big) \Big] + B_2(\varepsilon^2 + 4B_2^2 R_s(\varepsilon, f))^{\frac{1}{2}}.$*

*Proof.* Since the measure on $\mathcal{A}$ is uniform, we have

$$\mathbb{E}_{\boldsymbol{z}_1, \boldsymbol{z}_2 \in \mathcal{A}(\boldsymbol{z})} \|f(\boldsymbol{z}_1) - f(\boldsymbol{z}_2)\|_2 = \frac{1}{m^2} \sum_{\gamma=1}^{m} \sum_{\beta=1}^{m} \|f(A_\gamma(\boldsymbol{z})) - f(A_j(\boldsymbol{z}))\|_2,$$

hence,

$$\sup_{\boldsymbol{z}_1, \boldsymbol{z}_2 \in \mathcal{A}(\boldsymbol{z})} \|f(\boldsymbol{z}_1) - f(\boldsymbol{z}_2)\|_2 = \sup_{\gamma, \beta \in [m]} \|f(A_i(\boldsymbol{z})) - f(A_j(\boldsymbol{z}))\|_2 \leq \sum_{\gamma=1}^{m} \sum_{\beta=1}^{m} \|f(A_i(\boldsymbol{z})) - f(A_j(\boldsymbol{z}))\|_2$$

$$= m^2 \mathbb{E}_{\boldsymbol{z}_1, \boldsymbol{z}_2 \in \mathcal{A}(\boldsymbol{z})} \|f(\boldsymbol{z}_1) - f(\boldsymbol{z}_2)\|_2.$$

Denote $S := \{\boldsymbol{z} : \mathbb{E}_{\boldsymbol{z}_1, \boldsymbol{z}_2 \in \mathcal{A}(\boldsymbol{z})} \|f(\boldsymbol{z}_1) - f(\boldsymbol{z}_2)\|_2 > \frac{\varepsilon}{m^2}\}$, by the definition of $R_t(\varepsilon, f)$ along with Markov inequality, we have

$$R_t^2(\varepsilon, f) \leq \mathbb{P}_t^2(S) \leq \Big(\frac{\mathbb{E}_{\boldsymbol{z}} \mathbb{E}_{\boldsymbol{z}_1, \boldsymbol{z}_2 \in \mathcal{A}(\boldsymbol{z})} \|f(\boldsymbol{z}_1) - f(\boldsymbol{z}_2)\|_2}{\frac{\varepsilon}{m^2}}\Big)^2 \leq \frac{\mathbb{E}_{\boldsymbol{z}} \mathbb{E}_{\boldsymbol{z}_1, \boldsymbol{z}_2 \in \mathcal{A}(\boldsymbol{z})} \|f(\boldsymbol{z}_1) - f(\boldsymbol{z}_2)\|_2^2}{\frac{\varepsilon^2}{m^4}}$$

$$= \frac{m^4}{\varepsilon^2} \mathbb{E}_{\boldsymbol{z}} \mathbb{E}_{\boldsymbol{z}_1, \boldsymbol{z}_2 \in \mathcal{A}(\boldsymbol{z})} \|f(\boldsymbol{z}_1) - f(\boldsymbol{z}_2)\|_2^2. \tag{20}$$

Apart from that, similar process yields the first inequity to be justified in Lemma A.4:

$$R_s^2(\varepsilon, f) \leq \frac{m^4}{\varepsilon^2} \mathbb{E}_{\boldsymbol{x}} \mathbb{E}_{\boldsymbol{x}_1, \boldsymbol{x}_2 \in \mathcal{A}(\boldsymbol{x})} \|f(\boldsymbol{x}_1) - f(\boldsymbol{x}_2)\|_2^2 = \frac{m^4}{\varepsilon^2} \mathcal{L}_{\text{align}}(f).$$

Furthermore, we can turn out

$$\mathbb{E}_{\boldsymbol{z}} \mathbb{E}_{\boldsymbol{z}_1, \boldsymbol{z}_2 \in \mathcal{A}(\boldsymbol{z})} \|f(\boldsymbol{z}_1) - f(\boldsymbol{z}_2)\|_2^2$$

$$= \mathbb{E}_{\boldsymbol{x}} \mathbb{E}_{\boldsymbol{x}_1, \boldsymbol{x}_2 \in \mathcal{A}(\boldsymbol{x})} \|f(\boldsymbol{x}_1) - f(\boldsymbol{x}_2)\|_2^2 + \mathbb{E}_{\boldsymbol{z}} \mathbb{E}_{\boldsymbol{z}_1, \boldsymbol{z}_2 \in \mathcal{A}(\boldsymbol{z})} \|f(\boldsymbol{z}_1) - f(\boldsymbol{z}_2)\|_2^2 - \mathbb{E}_{\boldsymbol{x}} \mathbb{E}_{\boldsymbol{x}_1, \boldsymbol{x}_2 \in \mathcal{A}(\boldsymbol{x})} \|f(\boldsymbol{x}_1) - f(\boldsymbol{x}_2)\|_2^2$$

$$= \frac{1}{m^2} \sum_{\gamma=1}^{m} \sum_{\beta=1}^{m} \Big\{ \mathbb{E}_{\boldsymbol{z}} \|f(A_i(\boldsymbol{z})) - f(A_j(\boldsymbol{z}))\|_2^2 - \mathbb{E}_{\boldsymbol{x}} \|f(A_i(\boldsymbol{x})) - f(A_j(\boldsymbol{x}))\|_2^2 \Big\} + \mathbb{E}_{\boldsymbol{x}} \mathbb{E}_{\boldsymbol{x}_1, \boldsymbol{x}_2 \in \mathcal{A}(\boldsymbol{x})} \|f(\boldsymbol{x}_1) - f(\boldsymbol{x}_2)\|_2^2$$

$$= \frac{1}{m^2} \sum_{\gamma=1}^{m} \sum_{\beta=1}^{m} \sum_{l=1}^{d^*} \Big[ \mathbb{E}_{\boldsymbol{z}} \big\{ f_l(A_i(\boldsymbol{z})) - f_l(A_j(\boldsymbol{z})) \big\}^2 - \mathbb{E}_{\boldsymbol{x}} \big\{ f_l(A_i(\boldsymbol{x})) - f_l(A_j(\boldsymbol{x})) \big\}^2 \Big] + \mathbb{E}_{\boldsymbol{x}} \mathbb{E}_{\boldsymbol{x}_1, \boldsymbol{x}_2 \in \mathcal{A}(\boldsymbol{x})} \|f(\boldsymbol{x}_1) - f(\boldsymbol{x}_2)\|_2^2,$$

we subsequently focus on dealing with the first term. Since for all $\gamma \in [m], \beta \in [m]$ and $l \in [d^*]$,

$$\mathbb{E}_{\boldsymbol{z}}\big[f_l(A_i(\boldsymbol{z})) - f_l(A_j(\boldsymbol{z}))\big]^2 - \mathbb{E}_{\boldsymbol{x}}\big[f_l(A_i(\boldsymbol{x})) - f_l(A_j(\boldsymbol{x}))\big]^2$$

$$= \sum_{k=1}^{K} \Big[p_t(k)\mathbb{E}_{\boldsymbol{z}\in C_t(k)}\big\{f_l(A_i(\boldsymbol{z})) - f_l(A_j(\boldsymbol{z}))\big\}^2 - p_s(k)\mathbb{E}_{\boldsymbol{x}\in C_s(k)}\big\{f_l(A_i(\boldsymbol{x})) - f_l(A_j(\boldsymbol{x}))\big\}^2\Big]$$

$$= \sum_{k=1}^{K} \Big[p_t(k)\Big\{\mathbb{E}_{\boldsymbol{z}\in C_t(k)}\big\{f_l(A_i(\boldsymbol{z})) - f_l(A_j(\boldsymbol{z}))\big\}^2 - \mathbb{E}_{\boldsymbol{x}\in C_s(k)}\underbrace{\big\{f_l(A_i(\boldsymbol{x})) - f_l(A_j(\boldsymbol{x}))\big\}^2}_{g(\boldsymbol{x})}\Big\}$$

$$+ \big\{p_t(k) - p_s(k)\big\}\mathbb{E}_{\boldsymbol{x}\in C_s(k)}\big\{f_l(A_i(\boldsymbol{x})) - f_l(A_j(\boldsymbol{x}))\big\}^2\Big]$$

$$\leq 8B_2 M\mathcal{K}\epsilon_1 + 4B_2^2 K\epsilon_2.$$

To obtain the last inequality, it suffices to show $g(\boldsymbol{x}) \in \text{Lip}(8B_2M\mathcal{K})$. In fact, we know $\forall l \in [d^*], f_l \in \text{Lip}(\mathcal{K})$ as $f \in \text{Lip}(\mathcal{K})$, along with the fact that $A_i(\cdot)$ and $A_j(\cdot)$ are both $M$-Lipschitz continuous according to Assumption 3.5, we can conclude $f_l(A_i(\cdot)) - f_l(A_j(\cdot)) \in \text{Lip}(2M\mathcal{K})$. Additionally, note that $|f_l(A_i(\cdot)) - f_l(A_j(\cdot))| \leq 2B_2$ as $\|f\|_2 \leq B_2$, we can turn out outermost quadratic function remains locally $4B_2$-Lipschitz continuity in $[-2B_2, 2B_2]$, which implies that $g \in \text{Lip}(8B_2M\mathcal{K})$. Furthermore, by the definition of Wasserstein distance, we yield

$$\sum_{k=1}^{K} \Big[p_t(k)\Big(\mathbb{E}_{\boldsymbol{z}\in C_t(k)}\big\{f_l(A_i(\boldsymbol{z})) - f_l(A_j(\boldsymbol{z}))\big\}^2 - \mathbb{E}_{\boldsymbol{x}\in C_s(k)}\big\{f_l(A_i(\boldsymbol{x})) - f_l(A_j(\boldsymbol{x}))\big\}^2\Big)\Big] \leq 8B_2 M\mathcal{K}\epsilon_1 \sum_{k=1}^{K} p_t(k) = 8B_2 M\mathcal{K}\epsilon_1,$$

As for the second term in the last inequality, note that $f_l(A_i(\boldsymbol{x})) - f_l(A_j(\boldsymbol{x})) \leq 2B_2$ to yield

$$\sum_{k=1}^{K} \Big[\big\{p_t(k) - p_s(k)\big\}\mathbb{E}_{\boldsymbol{x}\in C_s(k)}\big\{f_l(A_i(\boldsymbol{x})) - f_l(A_j(\boldsymbol{x}))\big\}^2\Big] \leq 4B_2^2 K\epsilon_2.$$

Therefore,

$$\mathbb{E}_{\boldsymbol{z}}\mathbb{E}_{\boldsymbol{z}_1, \boldsymbol{z}_2 \in \mathcal{A}(\boldsymbol{z})}\|f(\boldsymbol{z}_1) - f(\boldsymbol{z}_2)\|_2^2 \leq \mathbb{E}_{\boldsymbol{x}}\mathbb{E}_{\boldsymbol{x}_1, \boldsymbol{x}_2 \in \mathcal{A}(\boldsymbol{x})}\|f(\boldsymbol{x}_1) - f(\boldsymbol{x}_2)\|_2^2 + 8B_2 d^* M\mathcal{K}\epsilon_1 + 4B_2^2 d^* K\epsilon_2. \tag{21}$$

Combining (20) and (21) turns out the second inequality of Lemma A.4.

$$R_t^2(\varepsilon, f) \leq \frac{m^4}{\varepsilon^2}\mathcal{L}_{\text{align}}(f) + \frac{8m^4}{\varepsilon^2}B_2 d^* M\mathcal{K}\epsilon_1 + \frac{4m^4}{\varepsilon^2}B_2^2 d^* K\epsilon_2.$$

To justify the third part of this Lemma, first recall Lemma A.2 that $\forall k \in [K], \|\mu_s(k) - \mu_t(k)\|_2 \leq \sqrt{d^*}M\mathcal{K}\epsilon_1$. Hence, for any $i \neq j$, we have

$$|\mu_t(i)^\top \mu_t(j) - \mu_s(i)^\top \mu_s(j)| = |\mu_t(i)^\top \mu_t(j) - \mu_t(i)^\top \mu_s(j) + \mu_t(i)^\top \mu_s(j) - \mu_s(i)^\top \mu_s(j)|$$

$$\leq \|\mu_t(i)\|_2\|\mu_t(j) - \mu_s(j)\|_2 + \|\mu_s(j)\|_2\|\mu_t(i) - \mu_s(i)\|_2 \leq 2\sqrt{d^*}B_2 M\mathcal{K}\epsilon_1,$$

so that we can further yield the relationship of class center divergence between the source domain and the target domain as follows:

$$\max_{i \neq j}|\mu_t(i)^\top \mu_t(j)| \leq \max_{i \neq j}|\mu_s(i)^\top \mu_s(j)| + 2\sqrt{d^*}B_2 M\mathcal{K}\epsilon_1. \tag{22}$$

Next, we will attempt to derive an upper bound for $\max_{i \neq j}|\mu_s(i)^\top \mu_s(j)|$. Let $U = \big(\sqrt{p_s(1)}\mu_s(1), \ldots, \sqrt{p_s(K)}\mu_s(K)\big) \in \mathbb{R}^{d^* \times K}$, then

$$\Big\|\sum_{k=1}^{K} p_s(k)\mu_s(k)\mu_s(k)^\top - I_{d^*}\Big\|_F^2 = \|UU^\top - I_{d^*}\|_F^2$$

$$= \text{Tr}(UU^\top UU^\top - 2UU^\top + I_{d^*}) \qquad (\|A\|_F^2 = \text{Tr}(A^\top A)))$$

$$= \mathrm{Tr}(U^\top U U^\top U - 2U^\top U) + \mathrm{Tr}(I_K) + d^* - K \qquad (\mathrm{Tr}(AB) = \mathrm{Tr}(BA))$$

$$\geq \|U^\top U - I_K\|_F^2 \qquad (d^* > K)$$

$$= \sum_{k=1}^{K}\sum_{l=1}^{K}(\sqrt{p_s(k)p_s(l)}\mu_s(k)^\top\mu_s(l) - \delta_{kl})^2$$

$$\geq p_s(i)p_s(j)(\mu_s(i)^\top\mu_s(j))^2.$$

Therefore,

$$(\mu_s(i)^\top\mu_s(j))^2 \leq \frac{\left\|\sum_{k=1}^{K}p_s(k)\mu_s(k)\mu_s(k)^\top - I_{d^*}\right\|_F^2}{p_s(i)p_s(j)}$$

$$= \frac{\left\|\mathbb{E}_{\boldsymbol{x}}\mathbb{E}_{\boldsymbol{x_1},\boldsymbol{x_2}\in\mathcal{A}(\boldsymbol{x})}\{f(\boldsymbol{x}_1)f(\boldsymbol{x}_2)^\top\} - I_{d^*} + \sum_{k=1}^{K}p_s(k)\mu_s(k)\mu_s(k)^\top - \mathbb{E}_{\boldsymbol{x}}\mathbb{E}_{\boldsymbol{x_1},\boldsymbol{x_2}\in\mathcal{A}(\boldsymbol{x})}\{f(\boldsymbol{x}_1)f(\boldsymbol{x}_2)^\top\}\right\|_F^2}{p_s(i)p_s(j)}$$

$$\leq \frac{2\left\|\mathbb{E}_{\boldsymbol{x}}\mathbb{E}_{\boldsymbol{x_1},\boldsymbol{x_2}\in\mathcal{A}(\boldsymbol{x})}\{f(\boldsymbol{x}_1)f(\boldsymbol{x}_2)^\top\} - I_{d^*}\right\|_F^2 + 2\left\|\sum_{k=1}^{K}p_s(k)\mu_s(k)\mu_s(k)^\top - \mathbb{E}_{\boldsymbol{x}}\mathbb{E}_{\boldsymbol{x_1},\boldsymbol{x_2}\in\mathcal{A}(\boldsymbol{x})}\{f(\boldsymbol{x}_1)f(\boldsymbol{x}_2)^\top\}\right\|_F^2}{p_s(i)p_s(j)} \qquad (23)$$

For the term $\left\|\sum_{k=1}^{K}p_s(k)\mu_s(k)\mu_s(k)^\top - \mathbb{E}_{\boldsymbol{x}}\mathbb{E}_{\boldsymbol{x}_1,\boldsymbol{x}_2\in\mathcal{A}(\boldsymbol{x})}[f(\boldsymbol{x}_1)f(\boldsymbol{x}_2)^\top]\right\|_F^2$, note that

$$= \sum_{k=1}^{K}p_s(k)\mathbb{E}_{\boldsymbol{x}\in C_s(k)}\mathbb{E}_{\boldsymbol{x}_1\in\mathcal{A}(\boldsymbol{x})}\{f(\boldsymbol{x}_1)f(\boldsymbol{x}_1)^\top\} - \sum_{k=1}^{K}p_s(k)\mu_s(k)\mu_s(k)^\top$$

$$+ \sum_{k=1}^{K}p_s(k)\mathbb{E}_{\boldsymbol{x}\in C_s(k)}\mathbb{E}_{\boldsymbol{x}_1,\boldsymbol{x}_2\in\mathcal{A}(\boldsymbol{x})}[f(\boldsymbol{x}_1)\{f(\boldsymbol{x}_2) - f(\boldsymbol{x}_1)\}^\top]$$

$$= \sum_{k=1}^{K}p_s(k)\mathbb{E}_{\boldsymbol{x}\in C_s(k)}\mathbb{E}_{\boldsymbol{x}_1\in\mathcal{A}(\boldsymbol{x})}[\{f(\boldsymbol{x}_1) - \mu_s(k)\}\{f(\boldsymbol{x}_1) - \mu_s(k)\}^\top] + \mathbb{E}_{\boldsymbol{x}}\mathbb{E}_{\boldsymbol{x}_1,\boldsymbol{x}_2\in\mathcal{A}(\boldsymbol{x})}[f(\boldsymbol{x}_1)\{f(\boldsymbol{x}_2) - f(\boldsymbol{x}_1)\}^\top],$$

$$(24)$$

where the last equation is derived from

$$\mathbb{E}_{\boldsymbol{x}\in C_s(k)}\mathbb{E}_{\boldsymbol{x}_1\in\mathcal{A}(\boldsymbol{x})}\{f(\boldsymbol{x}_1)f(\boldsymbol{x}_1)^\top\} - \mu_s(k)\mu_s(k)^\top = \mathbb{E}_{\boldsymbol{x}\in C_s(k)}\mathbb{E}_{\boldsymbol{x}_1\in\mathcal{A}(\boldsymbol{x})}\{f(\boldsymbol{x}_1)f(\boldsymbol{x}_1)^\top\} + \mu_s(k)\mu_s(k)^\top$$

$$- \left(\mathbb{E}_{\boldsymbol{x}\in C_s(k)}\mathbb{E}_{\boldsymbol{x}_1\in\mathcal{A}(\boldsymbol{x})}\{f(\boldsymbol{x}_1)\}\right)\mu_s(k)^\top - \mu_s(k)\left(\mathbb{E}_{\boldsymbol{x}\in C_s(k)}\mathbb{E}_{\boldsymbol{x}_1\in\mathcal{A}(\boldsymbol{x})}\{f(\boldsymbol{x}_1)\}\right)^\top$$

$$= \mathbb{E}_{\boldsymbol{x}\in C_s(k)}\mathbb{E}_{\boldsymbol{x}_1\in\mathcal{A}(\boldsymbol{x})}[\{f(\boldsymbol{x}_1) - \mu_s(k)\}\{f(\boldsymbol{x}_1) - \mu_s(k)\}^\top].$$

So its norm is

$$\left\|\sum_{k=1}^{K}p_s(k)\mu_s(k)\mu_s(k)^\top - \mathbb{E}_{\boldsymbol{x}}\mathbb{E}_{\boldsymbol{x_1},\boldsymbol{x_2}\in\mathcal{A}(\boldsymbol{x})}[f(\boldsymbol{x}_1)f(\boldsymbol{x}_2)^\top]\right\|_F$$

$$\leq \sum_{k=1}^{K}p_s(k)\mathbb{E}_{\boldsymbol{x}\in C_s(k)}\mathbb{E}_{\boldsymbol{x}_1\in\mathcal{A}(\boldsymbol{x})}[\|\{f(\boldsymbol{x}_1) - \mu_s(k)\}\{f(\boldsymbol{x}_1) - \mu_s(k)\}^\top\|_F] + \mathbb{E}_{\boldsymbol{x}}\mathbb{E}_{\boldsymbol{x_1},\boldsymbol{x_2}\in\mathcal{A}(\boldsymbol{x})}[\|f(\boldsymbol{x}_1)\{f(\boldsymbol{x}_2) - f(\boldsymbol{x}_1)\}^\top\|_F]$$

$$\leq \sum_{k=1}^{K}p_s(k)\mathbb{E}_{\boldsymbol{x}\in C_s(k)}\mathbb{E}_{\boldsymbol{x}_1\in\mathcal{A}(\boldsymbol{x})}\{\|f(\boldsymbol{x}_1) - \mu_s(k)\|_2^2\} + \mathbb{E}_{\boldsymbol{x}}\mathbb{E}_{\boldsymbol{x}_1,\boldsymbol{x}_2\in\mathcal{A}(\boldsymbol{x})}\{\|f(\boldsymbol{x}_1)\|_2\|f(\boldsymbol{x}_2) - f(\boldsymbol{x}_1)\|_2\}$$

$$\leq \sum_{k=1}^{K}p_s(k)\mathbb{E}_{\boldsymbol{x}\in C_s(k)}\mathbb{E}_{\boldsymbol{x}_1\in\mathcal{A}(\boldsymbol{x})}\{\|f(\boldsymbol{x}_1) - \mu_s(k)\|_2^2\} + \left\{\mathbb{E}_{\boldsymbol{x}}\mathbb{E}_{\boldsymbol{x}_1\in\mathcal{A}(\boldsymbol{x})}\|f(\boldsymbol{x}_1)\|_2^2\right\}^{\frac{1}{2}}\left\{\mathbb{E}_{\boldsymbol{x}}\mathbb{E}_{\boldsymbol{x}_1,\boldsymbol{x}_2\in\mathcal{A}(\boldsymbol{x})}\|f(\boldsymbol{x}_2) - f(\boldsymbol{x}_1)\|_2^2\right\}^{\frac{1}{2}}$$

$$\leq \sum_{k=1}^{K}p_s(k)\mathbb{E}_{\boldsymbol{x}\in C_s(k)}\mathbb{E}_{\boldsymbol{x}_1\in\mathcal{A}(\boldsymbol{x})}[\|f(\boldsymbol{x}_1) - \mu_s(k)\|_2^2] + B_2\left[\varepsilon^2 + \mathbb{E}_{\boldsymbol{x}}\mathbb{E}_{\boldsymbol{x}_1,\boldsymbol{x}_2\in\mathcal{A}(\boldsymbol{x})}[\|f(\boldsymbol{x}_2) - f(\boldsymbol{x}_1)\|_2^2\mathbb{1}\{\boldsymbol{x}\notin S_s(\varepsilon, f)\}]\right]^{\frac{1}{2}}$$

$\left( \text{Review that } S_s(\varepsilon, f) := \{ \boldsymbol{x} \in \cup_{k=1}^{K} C_s(k) : \sup_{\boldsymbol{x}_1, \boldsymbol{x}_2 \in \mathcal{A}(\boldsymbol{x})} \| f(\boldsymbol{x}_1) - f(\boldsymbol{x}_2) \|_2 \le \varepsilon \} \right)$

$$\le \sum_{k=1}^{K} p_s(k) \mathbb{E}_{\boldsymbol{x} \in C_s(k)} \mathbb{E}_{\boldsymbol{x}_1 \in \mathcal{A}(\boldsymbol{x})} \{ \| f(\boldsymbol{x}_1) - \mu_s(k) \|_2^2 \} + B_2 \left[ \varepsilon^2 + 4B_2^2 \mathbb{E}_{\boldsymbol{x}} \left[ \mathbb{1}\{ \boldsymbol{x} \notin S_s(\varepsilon, f) \} \right] \right]^{\frac{1}{2}}$$

$$= \sum_{k=1}^{K} p_s(k) \mathbb{E}_{\boldsymbol{x} \in C_s(k)} \mathbb{E}_{\boldsymbol{x}_1 \in \mathcal{A}(\boldsymbol{x})} \left[ \| f(\boldsymbol{x}_1) - \mu_s(k) \|_2^2 \right] + B_2 (\varepsilon^2 + 4B_2^2 R_s(\varepsilon, f))^{\frac{1}{2}}$$

$$\le 4B_2^2 \sum_{k=1}^{K} p_s(k) \left\{ \left( 1 - \sigma_s + \frac{\mathcal{K}\delta_s}{2B_2} + \frac{\varepsilon}{B_2} + \frac{R_s(\varepsilon, f)}{p_s(k)} \right)^2 + \left( 1 - \sigma_s + \frac{R_s(\varepsilon, f)}{p_s(k)} \right) \right\} + B_2 \{ \varepsilon^2 + 4B_2^2 R_s(\varepsilon, f) \}^{\frac{1}{2}}$$

$$\text{(Lemma A.3)}$$

$$= 4B_2^2 \left\{ \left( 1 - \sigma_s + \frac{\mathcal{K}\delta_s + 2\varepsilon}{2B_2} \right)^2 + (1 - \sigma_s) + K R_s(\varepsilon, f) \left( 3 - 2\sigma_s + \frac{\mathcal{K}\delta_s + 2\varepsilon}{B_2} \right) + R_s^2(\varepsilon, f) \left( \sum_{k=1}^{K} \frac{1}{p_s(k)} \right) \right\}$$

$$+ B_2 \{ \varepsilon^2 + 4B_2^2 R_s(\varepsilon, f) \}^{\frac{1}{2}}$$

If we define $\varphi(\sigma_s, \delta_s, \varepsilon, f) := 4B_2^2 \left\{ \left( 1 - \sigma_s + \frac{\mathcal{K}\delta_s + 2\varepsilon}{2B_2} \right)^2 + (1 - \sigma_s) + K R_s(\varepsilon, f) \left( 3 - 2\sigma_s + \frac{\mathcal{K}\delta_s + 2\varepsilon}{B_2} \right) + R_s^2(\varepsilon, f) \left( \sum_{k=1}^{K} \frac{1}{p_s(k)} \right) \right\} + B_2 (\varepsilon^2 + 4B_2^2 R_s(\varepsilon, f))^{\frac{1}{2}}$, above derivation implies

$$\left\| \sum_{k=1}^{K} p_s(k) \mu_s(k) \mu_s(k)^\top - \mathbb{E}_{\boldsymbol{x}} \mathbb{E}_{\boldsymbol{x}_1, \boldsymbol{x}_2 \in \mathcal{A}(\boldsymbol{x})} \{ f(\boldsymbol{x}_1) f(\boldsymbol{x}_2)^\top \} \right\|_F \le \varphi(\sigma_s, \delta_s, \varepsilon, f). \tag{25}$$

Besides that, Note that

$$\mathcal{R} = \left\| \mathbb{E}_{\boldsymbol{x}} \mathbb{E}_{\boldsymbol{x}_1, \boldsymbol{x}_2 \in \mathcal{A}(\boldsymbol{x})} [f(\boldsymbol{x}_1) f(\boldsymbol{x}_2)^\top] - I_{d^*} \right\|_F^2, \tag{26}$$

Combining (23), (24), (25) and (26) yields for any $i \ne j$

$$(\mu_s(i)^\top \mu_s(j))^2 \le \frac{2}{p_s(i) p_s(j)} \left\{ \mathcal{R}(f) + \varphi(\sigma_s, \delta_s, \varepsilon, f) \right\},$$

which implies that

$$\max_{i \ne j} |\mu_s(i)^\top \mu_s(j)| \le \sqrt{\frac{2}{\min_{i \ne j} p_s(i) p_s(j)} \left\{ \mathcal{R}(f) + \varphi(\sigma_s, \delta_s, \varepsilon, f) \right\}}.$$

So we can get what we desired according to (22)

$$\max_{i \ne j} |\mu_t(i)^\top \mu_t(j)| \le \sqrt{\frac{2}{\min_{i \ne j} p_s(i) p_s(j)} \left\{ \mathcal{R}(f) + \varphi(\sigma_s, \delta_s, \varepsilon, f) \right\}} + 2\sqrt{d^*} B_2 M \mathcal{K} \epsilon_1.$$

$$\square$$

Next we present the population theorem as follows, which is a direct corollary of Lemma A.4 because of the facts that $\mathcal{R}(f) \lesssim \mathcal{L}(f)$ and $\mathcal{L}_{\text{align}}(f) \lesssim \mathcal{L}(f)$.

**Theorem A.5.** *Given a $(\sigma_s, \sigma_t, \delta_s, \delta_t)$-augmentation, if $d^* > K$, Assumption 3.5 holds and the encoder $f$ with $B_1 \le \|f\|_2 \le B_2$ is $\mathcal{K}$-Lipschitz continuous, then for any $\varepsilon > 0$,*

$$\max_{i \ne j} |\mu_t(i)^\top \mu_t(j)| \lesssim \sqrt{\mathcal{L}(f) + \varphi(\sigma_s, \delta_s, \varepsilon, f)} + \mathcal{K}\epsilon_1.$$

*Furthermore, if $\max_{i \ne j} \mu_t(i)^\top \mu_t(j) < B_2^2 \psi(\sigma_t, \delta_t, \varepsilon, f)$, then the misclassification rate of $Q_f$*

$$\text{Err}(Q_f) \le (1 - \sigma_t) + \mathcal{O}\left( \{ \mathcal{L}_{\text{align}}(f) + \mathcal{K}\epsilon_1 + \epsilon_2 \} / \varepsilon^2 \right),$$

*where the specific formulations of $\varphi(\sigma_s, \delta_s, \varepsilon, f)$ and $\psi(\sigma_t, \delta_t, \varepsilon, f)$ can be found in Lemma A.4 and Lemma A.1, respectively.*

## A.3. Proof of Theorem 3.9

In this section, we focus on providing the proof of Theorem 3.9. Although Lemma A.4 elucidates some essential factors behind the success of our method, its analysis remains at the population level, leaving the impact of sample size on $\mathrm{Err}(Q_{\hat{f}_{n_s}})$ unresolved. To further explore this, applying Theorem A.5 yields Lemma A.6, indicating that investigating the sample complexity of $\mathbb{E}_{\widetilde{D}_s}\{\mathcal{L}(\hat{f}_{n_s})\}$ is a correct direction towards our goal.

However, there are two main challenges that hinder this exploration. The first is $\mathcal{L}(f^*)$, which represents the gap between $\mathcal{L}(\hat{f}_{n_s})$ and the excess risk $\mathcal{E}(\hat{f}_{n_s})$ defined in the Definition A.13. defined in Definition A.13. Since the excess risk can be addressed through typical error decomposition techniques and tools from nonparametric statistics, we aim to construct a measurable function under Assumption 3.3 that makes $\mathcal{L}(f^*)$ vanish, vanish, rather than directly assuming this term can be well-implemented by a specific neural network in HaoChen & Ma (2023).

The second issue stems from bias. To tackle this problem, we develop a novel risk decomposition in Section A.3.3. Utilizing this technique, $\mathbb{E}_{\widetilde{D}_s}\{\mathcal{E}(\hat{f}_{n_s})\}$ can be decomposed into three parts: statistical error: $\mathcal{E}_{\mathrm{sta}}$, approximation error brought by $\mathcal{F}$: $\mathcal{E}_{\mathcal{F}}$ and the error induced by using $\widehat{\mathcal{G}}(f)$ to approximate $\mathcal{G}(f)$: $\mathcal{E}_{\mathcal{G}}$. We subsequently address each term in succession. For $\mathcal{E}_{\mathrm{sta}}$, we apply standard empirical process techniques and leverage results from Golowich et al. (2018) in A.3.4 to bound it by $\frac{\mathcal{K}\sqrt{L}}{\sqrt{n_s}}$. Regarding $\mathcal{E}_{\mathcal{F}}$, we first reformulate the problem as a function approximation issue and adopt existing conclusions from Jiao et al. (2023), yielding a bound of $\mathcal{E}_{\mathcal{F}}$ in Section A.3.5. By leveraging the property $\mathbb{E}_{\widetilde{D}_s}[\widehat{\mathcal{L}}(f,G)] = \mathcal{L}(f,G)$, , we transform the problem of bounding $\mathbb{E}_{\widetilde{D}_s}\{\mathcal{E}_{\mathcal{G}}\}$ into a common problem of mean convergence rate, further controlling it by $\frac{1}{n_s^{1/4}}$ in Section A.3.6.

After completing these preliminaries, we balance these errors to determine appropriate values for the width $W$, depth $D$ and the Lipschitz constant $\mathcal{K}$ of the neural network while establishing an end-to-end upper bound for $\mathbb{E}_{\widetilde{D}_s}\{\mathcal{E}(\hat{f}_{n_s})\}$, More details are deferred to Seciton A.3.7. Finally, Lemma A.20 presents the formal version of Theorem 3.9, with the connection between Lemma A.20 and Theorem 3.9 detailed in A.3.8.

As stated above, we apply Theorem A.5 to the sample optimizer $\hat{f}_{n_s}$ to yield Lemma A.6.

**Lemma A.6.** *Given a $(\sigma_s, \sigma_t, \delta_s, \delta_t)$-augmentation, for any $\varepsilon > 0$, if $\psi(\sigma_t, \delta_t, \varepsilon, \hat{f}_{n_s}) > 0$, then with probability at least*

$$1 - \frac{\sqrt{\frac{2}{\min_{i \neq j} p_s(i)p_s(j)}\left[\frac{1}{\lambda}\mathbb{E}_{\widetilde{D}_s}\{\mathcal{L}(\hat{f}_{n_s})\}+\phi(\sigma_s,\delta_s,\varepsilon,\hat{f}_{n_s})\right]+2\sqrt{d^*}B_2 M\mathcal{K}\epsilon_1}}{B_2^2\psi(\sigma_t,\delta_t,\varepsilon,\hat{f}_{n_s})}, \text{ we have}$$

$$\mathbb{E}_{\widetilde{D}_s}\{\mathrm{Err}(Q_{\hat{f}_{n_s}})\} \leq (1 - \sigma_t) + \frac{m^2}{\varepsilon}\sqrt{\mathbb{E}_{\widetilde{D}_s}\{\mathcal{L}(\hat{f}_{n_s})\} + 8B_2 d^* M\mathcal{K}\epsilon_1 + 4B_2^2 d^* K\epsilon_2},$$

*where*

$$\phi(\sigma_s, \delta_s, \varepsilon, \hat{f}_{n_s}) := B_2\left(\varepsilon^2 + 4B_2^2\frac{m^2}{\varepsilon}\sqrt{\mathbb{E}_{\widetilde{D}_s}\{\mathcal{L}(\hat{f}_{n_s})\}}\right)^{\frac{1}{2}} + 4B_2^2\left[\left(1 - \sigma_s + \frac{\mathcal{K}\delta_s + 2\varepsilon}{2B_2}\right)^2\right.$$

$$+ (1 - \sigma_s) + \frac{Km^2}{\varepsilon}\sqrt{\mathbb{E}_{\widetilde{D}_s}\{\mathcal{L}(\hat{f}_{n_s})\}}\left(3 - 2\sigma_s + \frac{\mathcal{K}\delta_s + 2\varepsilon}{B_2}\right)$$

$$+ \left.\frac{m^4}{\varepsilon^2}\mathbb{E}_{\widetilde{D}_s}\{\mathcal{L}(\hat{f}_{n_s})\}\left(\sum_{k=1}^{K}\frac{1}{p_s(k)}\right)\right].$$

*In addition, the following inequalities always hold*

$$\mathbb{E}_{\widetilde{D}_s}\{R_t^2(\varepsilon, \hat{f}_{n_s})\} \leq \frac{m^4}{\varepsilon^2}\left(\mathbb{E}_{\widetilde{D}_s}\{\mathcal{L}(\hat{f}_{n_s})\} + 8B_2 d^* M\mathcal{K}\epsilon_1 + 4B_2^2 d^* K\epsilon_2\right).$$

*Proof.* Applying Lemma A.4 to $\hat{f}_{n_s}$ yields

$$R_s^2(\varepsilon, \hat{f}_{n_s}) \leq \frac{m^4}{\varepsilon^2}\mathcal{L}(\hat{f}_{n_s}) \tag{27}$$

$$R_t^2(\varepsilon, \hat{f}_{n_s}) \leq \frac{m^4}{\varepsilon^2}\mathcal{L}(\hat{f}_{n_s}) + \frac{8m^4}{\varepsilon^2}B_2 d^* M\mathcal{K}\epsilon_1 + \frac{4m^4}{\varepsilon^2}B_2^2 d^* K\epsilon_2 \tag{28}$$

and

$$\max_{i \neq j} |\mu_t(i)^\top \mu_t(j)| \leq \sqrt{\frac{2}{\min_{i \neq j} p_s(i) p_s(j)} \Big( \frac{1}{\lambda} \mathcal{L}(\hat{f}_{n_s}) + \varphi(\sigma_s, \delta_s, \varepsilon, \hat{f}_{n_s}) \Big)} + 2\sqrt{d^*} B_2 M \mathcal{K} \epsilon_1 \qquad (29)$$

Take expectation regarding to $D_s$ on the both sides of (27), (28) and (29), along with the Jensen's inequality to obtain

$$\mathbb{E}_{\widetilde{D}_s}\{R_s^2(\varepsilon, \hat{f}_{n_s})\} \leq \frac{m^4}{\varepsilon^2} \mathbb{E}_{\widetilde{D}_s}\{\mathcal{L}(\hat{f}_{n_s})\}$$

$$\mathbb{E}_{\widetilde{D}_s}\{R_t^2(\varepsilon, \hat{f}_{n_s})\} \leq \frac{m^4}{\varepsilon^2} \mathbb{E}_{\widetilde{D}_s}\{\mathcal{L}(\hat{f}_{n_s})\} + \frac{8m^4}{\varepsilon^2} B_2 d^* M \mathcal{K} \epsilon_1 + \frac{4m^4}{\varepsilon^2} B_2^2 d^* K \epsilon_2$$

$$\mathbb{E}_{\widetilde{D}_s}[\max_{i \neq j} |\mu_t(i)^\top \mu_t(j)|] \leq \sqrt{\frac{2}{\min_{i \neq j} p_s(i) p_s(j)} \Big( \frac{1}{\lambda} \mathbb{E}_{\widetilde{D}_s}\{\mathcal{L}(\hat{f}_{n_s})\} + \mathbb{E}_{\widetilde{D}_s}[\varphi(\sigma_s, \delta_s, \varepsilon, \hat{f}_{n_s})] \Big)} + 2\sqrt{d^*} B_2 M \mathcal{K} \epsilon_1$$

where $\mathbb{E}_{\widetilde{D}_s}\{\varphi(\sigma_s, \delta_s, \varepsilon, \hat{f}_{n_s})\} = 4B_2^2 \Big[ \Big( 1 - \sigma_s + \frac{\mathcal{K}\delta_s + 2\varepsilon}{2B_2} \Big)^2 + (1 - \sigma_s) + K \mathbb{E}_{\widetilde{D}_s}\{R_s(\varepsilon, \hat{f}_{n_s})\} \Big( 3 - 2\sigma_s + \frac{\mathcal{K}\delta_s + 2\varepsilon}{B_2} \Big) + \mathbb{E}_{\widetilde{D}_s}\{R_s^2(\varepsilon, \hat{f}_{n_s})\} \Big( \sum_{k=1}^K \frac{1}{p_s(k)} \Big) \Big] + B_2 \mathbb{E}_{\widetilde{D}_s}[\{\varepsilon^2 + 4B_2^2 R_s(\varepsilon, \hat{f}_{n_s})\}^{\frac{1}{2}}].$

Therefore, by Jensen inequality, we have

$$\mathbb{E}_{\widetilde{D}_s}\{\varphi(\sigma_s, \delta_s, \varepsilon, R_s(\varepsilon, \hat{f}_{n_s}))\} \leq 4B_2^2 \Big[ \Big( 1 - \sigma_s + \frac{\mathcal{K}\delta_s + 2\varepsilon}{2B_2} \Big)^2 + (1 - \sigma_s) + K \mathbb{E}_{\widetilde{D}_s}[R_s(\varepsilon, \hat{f}_{n_s})] \Big( 3 - 2\sigma_s + \frac{\mathcal{K}\delta_s + 2\varepsilon}{B_2} \Big)$$

$$+ \mathbb{E}_{\widetilde{D}_s}[R_s^2(\varepsilon, \hat{f}_{n_s})] \Big( \sum_{k=1}^K \frac{1}{p_s(k)} \Big) \Big] + B_2 [\varepsilon^2 + 4B_2^2 \mathbb{E}_{\widetilde{D}_s}\{R_s(\varepsilon, \hat{f}_{n_s})\}]^{\frac{1}{2}}$$

$$\leq 4B_2^2 \Big[ \Big( 1 - \sigma_s + \frac{\mathcal{K}\delta_s + 2\varepsilon}{2B_2} \Big)^2 + \frac{Km^2}{\varepsilon} \sqrt{\mathbb{E}_{\widetilde{D}_s}\{\mathcal{L}(\hat{f}_{n_s})\}} \Big( 3 - 2\sigma_s + \frac{\mathcal{K}\delta_s + 2\varepsilon}{B_2} \Big) + \frac{m^4}{\varepsilon^2} \mathbb{E}_{\widetilde{D}_s}\{\mathcal{L}(\hat{f}_{n_s})\} \Big( \sum_{k=1}^K \frac{1}{p_s(k)} \Big) \Big]$$

$$+ (1 - \sigma_s) + B_2 \Big( \varepsilon^2 + \frac{4B_2^2 m^2}{\varepsilon} \sqrt{\mathbb{E}_{\widetilde{D}_s}\{\mathcal{L}(\hat{f}_{n_s})\}} \Big)^{\frac{1}{2}} := \phi(\sigma_s, \delta_s, \varepsilon, \hat{f}_{n_s}).$$

Since Lemma A.1 reveals that if $\max_{i \neq j} |(\mu_t(i))^\top \mu_t(j)| < B_2^2 \psi(\sigma_t, \delta_t, \varepsilon, \hat{f}_{n_s})$, then $\text{Err}(Q_{\hat{f}_{n_s}}) \leq (1 - \sigma_t) + R_t(\varepsilon, \hat{f}_{n_s})$. Thus, if $\psi(\sigma_t, \delta_t, \varepsilon, \hat{f}_{n_s}) > 0$, by Markov inequality, we know that with probability at least $1 - \frac{\sqrt{\frac{2}{\min_{i \neq j} p_s(i) p_s(j)} \Big( \frac{1}{\lambda} \mathbb{E}_{\widetilde{D}_s}\{\mathcal{L}(\hat{f}_{n_s})\} + \phi(\sigma_s, \delta_s, \varepsilon, \hat{f}_{n_s}) \Big)} + 2\sqrt{d^*} B_2 M \mathcal{K} \epsilon_1}{B_2^2 \psi(\sigma_t, \delta_t, \varepsilon, \hat{f}_{n_s})}$, $\max_{i \neq j} |\mu_t(i)^\top \mu_t(j)| < B_2^2 \psi(\sigma_t, \delta_t, \varepsilon, \hat{f}_{n_s})$, which implies that

$$\mathbb{E}_{\widetilde{D}_s}\{\text{Err}(Q_{\hat{f}_{n_s}})\} \leq (1 - \sigma_t) + R_t(\varepsilon, \hat{f}_{n_s}) \leq (1 - \sigma_t) + \frac{m^2}{\varepsilon} \sqrt{\mathbb{E}_{\widetilde{D}_s}\{\mathcal{L}(\hat{f}_{n_s})\} + 8B_2 d^* M \mathcal{K} \epsilon_1 + 4B_2^2 d^* K \epsilon_2},$$

where the last inequality stems from (28). $\qquad \square$

Therefore, to justify Theorem 3.9, we need to explore the sample complexity of $\mathbb{E}_{\widetilde{D}_s}\{\mathcal{L}(\hat{f}_{n_s})\}$. To this end, it is necessary to introduce some basic facts about ACT and learning theory.

### A.3.1. PRELIMINARIES FOR PROVING THEOREM 3.9

Recall that for any $\boldsymbol{x} \in \mathcal{X}_s, \boldsymbol{x}_1, \boldsymbol{x}_2 \overset{\text{i.i.d.}}{\sim} A(\boldsymbol{x}), \tilde{\boldsymbol{x}} = (\boldsymbol{x}_1, \boldsymbol{x}_2) \in \mathbb{R}^{2d^*}$. If we define $\ell(\tilde{\boldsymbol{x}}, G) := \|f(\boldsymbol{x}_1) - f(\boldsymbol{x}_2)\|_2^2 + \lambda \langle f(\boldsymbol{x}_1) f(\boldsymbol{x}_2)^\top - I_{d^*}, G \rangle_F$, then our loss function at sample level can be rewritten as

$$\widehat{\mathcal{L}}(f, G) := \frac{1}{n_s} \sum_{i=1}^{n_s} \Big\{ \|f(\boldsymbol{x}_1^{(i)}) - f(\boldsymbol{x}_2^{(i)})\|_2^2 + \lambda \langle f(\boldsymbol{x}_1^{(i)}) f(\boldsymbol{x}_2^{(i)})^\top - I_{d^*}, G \rangle_F \Big\} = \frac{1}{n_s} \sum_{i=1}^{n_s} \ell(\tilde{\boldsymbol{x}}^{(i)}, G),$$

moreover, let $\mathcal{G}_1 := \{G \in \mathbb{R}^{d^* \times d^*} : \|G\|_F \leq B_2^2 + \sqrt{d^*}\}$. It is obvious that both $\mathcal{G}(f)$ for any $f : \|f\|_2 \leq B_2$ and $\widehat{\mathcal{G}}(f)$ for any $f \in \mathcal{NN}_{d,d^*}(W, L, \mathcal{K}, B_1, B_2)$ are the subset of $\mathcal{G}_1$. In this regard, following Proposition A.7 reveals that $\ell(\boldsymbol{u}, G)$ is a Lipschitz function on the domain $\{\boldsymbol{u} \in \mathbb{R}^{2d^*} : \|\boldsymbol{u}\|_2 \leq \sqrt{2} B_2\} \times \mathcal{G}_1 \subseteq \mathbb{R}^{2d^* + (d^*)^2}$.

**Proposition A.7.** $\ell$ is a Lipschitz function on the domain $\{\boldsymbol{u} \in \mathbb{R}^{2d^*} : \|\boldsymbol{u}\|_2 \leq \sqrt{2}B_2\} \times \mathcal{G}_1$.

*Proof.* We begin by proving that $\|\ell(\cdot, G)\|_{\mathrm{Lip}} < \infty$ for any fixed $G \in \mathcal{G}_1$. To this end, let $\boldsymbol{u} = (\boldsymbol{u}_1, \boldsymbol{u}_2)$, where $\boldsymbol{u}_1, \boldsymbol{u}_2 \in \mathbb{R}^{d^*}$. We first demonstrate that $J(\boldsymbol{u}) = \|\boldsymbol{u}_1 - \boldsymbol{u}_2\|_2^2$ is a Lipschitz function. Define $g(\boldsymbol{u}) := \boldsymbol{u}_1 - \boldsymbol{u}_2$. We have:

$$\|g(\boldsymbol{u}) - g(\boldsymbol{v})\|_2^2 = \|\boldsymbol{u}_1 - \boldsymbol{u}_2 - \boldsymbol{v}_1 + \boldsymbol{v}_2\|_2^2 \leq (\|\boldsymbol{u}_1 - \boldsymbol{v}_1\|_2 + \|\boldsymbol{u}_2 - \boldsymbol{v}_2\|_2)^2$$
$$= \|\boldsymbol{u}_1 - \boldsymbol{v}_1\|_2^2 + \|\boldsymbol{u}_2 - \boldsymbol{v}_2\|_2^2 + 2\|\boldsymbol{u}_1 - \boldsymbol{v}_1\|_2\|\boldsymbol{u}_2 - \boldsymbol{v}_2\|_2$$
$$\leq 2\left(\|\boldsymbol{u}_1 - \boldsymbol{v}_1\|_2^2 + \|\boldsymbol{u}_2 - \boldsymbol{v}_2\|_2^2\right) = 2\|\boldsymbol{u} - \boldsymbol{v}\|_2^2,$$

which implies that $g(\boldsymbol{u}) \in \mathrm{Lip}(\sqrt{2})$. Furthermore, $g$ possesses the property that $\|g(\boldsymbol{u})\|_2 = \|\boldsymbol{u}_1 - \boldsymbol{u}_2\|_2 \leq \|\boldsymbol{u}_1\|_2 + \|\boldsymbol{u}_2\|_2 \leq 2\|\boldsymbol{u}\|_2 \leq 2\sqrt{2}B_2$. Next, let $h(\boldsymbol{v}) := \|\boldsymbol{v}\|_2^2$. We have:

$$\left\|\frac{\partial h}{\partial \boldsymbol{v}}(g(\boldsymbol{u}))\right\|_2 = 2\|g(\boldsymbol{u})\|_2 \leq 4\sqrt{2}B_2.$$

Thus, $J(\boldsymbol{u}) = h(g(\boldsymbol{u})) = \|\boldsymbol{u}_1 - \boldsymbol{u}_2\|_2^2 \in \mathrm{Lip}(8B_2)$. Now, we show that $Q(\boldsymbol{u}) = \langle \boldsymbol{u}_1\boldsymbol{u}_2^\top - I_{d^*}, G \rangle_F$ is also a Lipschitz function. Define $\tilde{g}(\boldsymbol{u}) := \boldsymbol{u}_1\boldsymbol{u}_2^\top$. We have:

$$\|\tilde{g}(\boldsymbol{u}) - \tilde{g}(\boldsymbol{v})\|_F = \|\boldsymbol{u}_1\boldsymbol{u}_2^\top - \boldsymbol{v}_1\boldsymbol{v}_2^\top\|_F = \|\boldsymbol{u}_1\boldsymbol{u}_2^\top - \boldsymbol{u}_1\boldsymbol{v}_2^\top + \boldsymbol{u}_1\boldsymbol{v}_2^\top - \boldsymbol{v}_1\boldsymbol{v}_2^\top\|_F$$
$$= \|\boldsymbol{u}_1(\boldsymbol{u}_2 - \boldsymbol{v}_2)^\top + (\boldsymbol{u}_1 - \boldsymbol{v}_1)\boldsymbol{v}_2^\top\|_F \leq \|\boldsymbol{u}_1\|_F\|\boldsymbol{u}_2 - \boldsymbol{v}_2\|_F + \|\boldsymbol{u}_1 - \boldsymbol{v}_1\|_F\|\boldsymbol{v}_2\|_F$$
$$\leq (\|\boldsymbol{u}_1\|_2 + \|\boldsymbol{v}_2\|_2)\|\boldsymbol{u} - \boldsymbol{v}\|_2 \leq 2\sqrt{2}B_2\|\boldsymbol{u} - \boldsymbol{v}\|_2.$$

Subsequently, denote $\tilde{h}(A) := \langle A - I_{d^*}, G \rangle_F$. Then, we find that $\|\nabla\tilde{h}(A)\|_F = \|G\|_F \leq B_2^2 + \sqrt{d^*}$. Therefore, we conclude that $Q(\boldsymbol{u}) = \tilde{h}(\tilde{g}(\boldsymbol{u})) \in \mathrm{Lip}(2\sqrt{2}B_2(B_2^2 + \sqrt{d^*}))$. Combining the above results, we establish that for any $G \in \mathcal{G}_1$, we have $\|\ell(\cdot, G)\|_{\mathrm{Lip}} < \infty$ on the domain $\{\boldsymbol{u} : \|\boldsymbol{u}\|_2 \leq \sqrt{2}B_2\}$. Next, for a fixed $\boldsymbol{u} \in \mathbb{R}^{2d^*}$ such that $\|\boldsymbol{u}\|_2 \leq \sqrt{2}B_2$, we obtain:

$$|\ell(\boldsymbol{u}, G_1) - \ell(\boldsymbol{u}, G_2)| = |\langle \boldsymbol{u}, G_1 - G_2 \rangle_F| \leq \|\boldsymbol{u}\|_2\|G_1 - G_2\|_F = \sqrt{2}B_2\|G_1 - G_2\|_F,$$

which implies that $\ell(\boldsymbol{u}, \cdot) \in \mathrm{Lip}(\sqrt{2}B_2)$. Finally, we note that:

$$|\ell(\boldsymbol{u}_1, G_1) - \ell(\boldsymbol{u}_2, G_2)|^2 \leq \{|\ell(\boldsymbol{u}_1, G_1) - \ell(\boldsymbol{u}_2, G_1)| + |\ell(\boldsymbol{u}_2, G_1) - \ell(\boldsymbol{u}_2, G_2)|\}^2$$
$$\leq \left[\{\sqrt{2} + 2\sqrt{2}B_2(B_2^2 + \sqrt{d^*})\}\|\boldsymbol{u}_1 - \boldsymbol{u}_2\|_2 + \sqrt{2}B_2\|G_1 - G_2\|_F\right]^2$$
$$\leq 2\{\sqrt{2} + 2\sqrt{2}B_2(B_2^2 + \sqrt{d^*})\}^2\|\boldsymbol{u}_1 - \boldsymbol{u}_2\|_2^2 + 4B_2^2\|G_1 - G_2\|_F^2$$
$$\leq C\|\mathrm{vec}(\boldsymbol{u}_1, G_1) - \mathrm{vec}(\boldsymbol{u}_2, G_2)\|_2^2,$$

where $C$ is a constant such that $C \geq \max\left\{2\{\sqrt{2} + 2\sqrt{2}B_2(B_2^2 + \sqrt{d^*})\}^2, 4B_2^2\right\}$, thus yielding the desired result. $\square$

We summary the Lipschitz constants of $\ell(\boldsymbol{u}, G)$ with respect to $\boldsymbol{u} \in \{\boldsymbol{u} \in \mathbb{R}^{2d^*} : \|\boldsymbol{u}\|_2 \leq \sqrt{2}B_2\}$ and $G \in \mathcal{G}_1$ in Table 3.

*Table 3.* Lipschitz constant of $\ell$ with respect to each component

| Function | Lipschitz Constant |
|----------|--------------------|
| $\ell(\boldsymbol{u}, \cdot)$ | $\sqrt{2}B_2$ |
| $\ell(\cdot, G)$ | $2\sqrt{2}B_2(B_2^2 + \sqrt{d^*})$ |
| $\ell(\cdot)$ | $\max\left\{\sqrt{2}B_2, 2\sqrt{2}B_2(B_2^2 + \sqrt{d^*})\right\}$ |

Following Definition A.8, A.10 and Lemma A.9, A.11 are all typical elements in the area of learning theory.

**Definition A.8** (Rademacher complexity). Given a set $S \subseteq \mathbb{R}^n$, the Rademacher complexity of $S$ is denoted by

$$\mathcal{R}_n(S) := \mathbb{E}_\xi \Big[ \sup_{(s_1,\dots,s_n) \in S} \frac{1}{n} \sum_{i=1}^n \xi_i s_i \Big],$$

where $\{\xi_i\}_{i \in [n]}$ is a sequence of i.i.d Radmacher random variables which take the values $1$ and $-1$ with equal probability $1/2$.

Following vector-contraction principle of Rademacher complexity will be used in later contents.

**Lemma A.9** (Vector-contraction principle). *Let $\mathcal{X}$ be any set, $(x_1, \dots, x_n) \in \mathcal{X}^n$, let $F$ be a class of functions $f : \mathcal{X} \to \ell_2$ and let $h_i : \ell_2 \to \mathbb{R}$ have Lipschitz norm $L$. Then*

$$\mathbb{E} \sup_{f \in F} \big| \sum_i \epsilon_i h_i(f(x_i)) \big| \leq 2\sqrt{2} L \mathbb{E} \sup_{f \in F} \big| \sum_{i,k} \varepsilon_{ik} f_k(x_i) \big|,$$

*where $\epsilon_{ik}$ is an independent doubly indexed Rademacher sequence and $f_k(x_i)$ is the k-th component of $f(x_i)$.*

*Proof.* Combining Maurer (2016) and Theorem 3.2.1 of Giné & Nickl (2016) obtains the desired result. □

**Definition A.10** (Covering number). $\forall n \in \mathbb{N}^+$, Fix $\mathcal{S} \subseteq \mathbb{R}^n$ and $\varrho > 0$, the set $\mathcal{N}$ is called an $\varrho$-net of $\mathcal{S}$ with respect to a norm $\|\cdot\|$ on $\mathbb{R}^n$, if $\mathcal{N} \subseteq \mathcal{S}$ and for any $\boldsymbol{u} \in \mathcal{S}$, there exists $\boldsymbol{v} \in \mathcal{N}$ such that $\|\boldsymbol{u} - \boldsymbol{v}\| \leq \varrho$. The covering number of $\mathcal{S}$ is defined as

$$\mathcal{N}(\mathcal{S}, \|\cdot\|, \varrho) := \min\{|\mathcal{Q}| : \mathcal{Q} \text{ is an } \varrho\text{-cover of } \mathcal{S}\}$$

where $|\mathcal{Q}|$ is the cardinality of the set $\mathcal{Q}$.

According to the Corollary 4.2.13 of Vershynin (2018), $|\mathcal{N}(\mathcal{B}_2, \|\cdot\|_2, \varrho)|$, which is the the covering number of 2-norm unit ball in $\mathbb{R}^{(d^*)^2}$, can be bounded by $(\frac{3}{\varrho})^{(d^*)^2}$, so that if we denote $\mathcal{N}_{\mathcal{G}_1}(\varrho)$ is a cover of $\mathcal{G}_1$ with radius $\varrho$ whose cardinality $|\mathcal{N}_{\mathcal{G}_1}(\varrho)|$ is equal to the covering number of $\mathcal{G}_1$, then $|\mathcal{N}_{\mathcal{G}_1}(\varrho)| \leq \big\{ \frac{3}{(B_2^2 + \sqrt{d^*})\varrho} \big\}^{(d^*)^2}$.

**Lemma A.11** (Finite maximum inequality). *For any $N \geq 1$, if $X_i, i \leq N$, are sub-Gaussian random variables admitting constants $\sigma_i$, then*

$$\mathbb{E} \max_{i \leq N} |X_i| \leq \sqrt{2 \log 2N} \max_{i \leq N} \sigma_i$$

The proof of this lemma can be found in Giné & Nickl (2016), Lemma 2.3.4.

Recall $\mathcal{NN}_{d_1,d_2}(W, L, \mathcal{K}) := \{f_\theta(\boldsymbol{x}) = A_L \sigma(A_{L-1} \sigma(\cdots \sigma(A_0 \boldsymbol{x})) : \kappa(\theta) \leq \mathcal{K}\}$, as defined in eq 10. The second lemma we will employ is related to the upper bound for the Rademacher complexity of the hypothesis space consisting of norm-constrained neural networks, which was provided by Golowich et al. (2018).

**Lemma A.12** (Theorem 3.2 of Golowich et al. (2018)). $\forall n \in \mathbb{N}^+, \forall \boldsymbol{x}_1, \dots, \boldsymbol{x}_n \in [-B, B]^d$ *with* $B \geq 1, S := \{(f(\boldsymbol{x}_1), \dots, f(\boldsymbol{x}_n)) : f \in \mathcal{NN}_{d,1}(W, L, \mathcal{K})\} \subseteq \mathbb{R}^n$, *then*

$$\mathcal{R}_n(S) \leq \frac{1}{n} \mathcal{K} \sqrt{2(L + 2 + \log(d+1))} \max_{1 \leq j \leq d+1} \sqrt{\sum_{i=1}^n x_{i,j}^2} \leq \frac{B\mathcal{K}\sqrt{2(L + 2 + \log(d+1))}}{\sqrt{n}},$$

*where $x_{i,j}$ is the j-th coordinate of the vector $(\boldsymbol{x_i}^\top, 1)^\top \in \mathbb{R}^{d+1}$.*

**Definition A.13** (Excess risk). The difference between $\mathcal{L}(\hat{f}_{n_s})$ and $\mathcal{L}(f^*)$ is called excess risk, i.e.,

$$\mathcal{E}(\hat{f}_{n_s}) = \mathcal{L}(\hat{f}_{n_s}) - \mathcal{L}(f^*).$$

A.3.2. DEAL WITH $\mathcal{L}(f^*)$

Since our objective is to explore the sample complexity of $\mathbb{E}_{\widetilde{D}_s}\{\mathcal{L}(\hat{f}_{n_s})\}$, it is essential to assert that $\mathcal{L}(f^*) = 0$. This ensures that the tools used to analyze $\mathbb{E}_{\widetilde{D}_s}\{\mathcal{E}(\hat{f}_{n_s})\}$ are also applicable for handling $\mathbb{E}_{\widetilde{D}_s}\{\mathcal{L}(\hat{f}_{n_s})\}$. The justification comprises a total of two steps. First, we assert that if there exists a measurable map $f$ such that $\Sigma = \mathbb{E}_{\boldsymbol{x}\sim\mathbb{P}_s}[f(\boldsymbol{x})f(\boldsymbol{x})^\top]$ be positive definite,then we can make minor modifications to obtain $\tilde{f}$ such that $\mathcal{L}(\tilde{f}) = 0$. In the second step, we will demonstrate that the required $f$ exists under Assumption 3.3, and that the modified $\tilde{f}$ also satisfies the condition $B_1 \leq \|\tilde{f}\|_2 \leq B_2$, which implies that $\mathcal{L}(f^*) = 0$, since the definition of $f^*$ indicates that $\mathcal{L}(f^*) \leq \mathcal{L}(\tilde{f})$.

Our final target is to result in a measurable map $f$, s.t $B_1 \leq \|f\|_2 \leq B_2$ and $\sup_{f\in\mathcal{G}(f)}\mathcal{L}(f) = 0$, it suffices to find a $f: B_1 \leq \|f\|_2 \leq B_2$ satisfying both $\mathcal{L}_{\text{align}}(f) = 0$ and $\left\|\mathbb{E}_{\boldsymbol{x}}\mathbb{E}_{\boldsymbol{x_1},\boldsymbol{x_2}\in\mathcal{A}(\boldsymbol{x})}[f(\boldsymbol{x_1})f(\boldsymbol{x_2})^\top] - I_{d^*}\right\|_F = 0$. Note that

$$\left\|\mathbb{E}_{\boldsymbol{x}}\mathbb{E}_{\boldsymbol{x_1},\boldsymbol{x_2}\in\mathcal{A}(\boldsymbol{x})}\{f(\boldsymbol{x_1})f(\boldsymbol{x_2})^\top\} - I_{d^*}\right\|_F$$

$$= \left\|\mathbb{E}_{\boldsymbol{x}}\mathbb{E}_{\boldsymbol{x_1},\boldsymbol{x_2}\in\mathcal{A}(\boldsymbol{x})}\{f(\boldsymbol{x_1})f(\boldsymbol{x_1})^\top\} + \mathbb{E}_{\boldsymbol{x}}\mathbb{E}_{\boldsymbol{x_1},\boldsymbol{x_2}\in\mathcal{A}(\boldsymbol{x})}\left[f(\boldsymbol{x_1})\{f(\boldsymbol{x_2}) - f(\boldsymbol{x_1})\}^\top\right] - I_{d^*}\right\|_F$$

$$\leq \left\|\mathbb{E}_{\boldsymbol{x}}\mathbb{E}_{\boldsymbol{x_1}\in\mathcal{A}(\boldsymbol{x})}\{f(\boldsymbol{x_1})f(\boldsymbol{x_1})^\top\} - I_{d^*}\right\|_F + \mathbb{E}_{\boldsymbol{x}}\mathbb{E}_{\boldsymbol{x_1},\boldsymbol{x_2}}\{\|f(\boldsymbol{x_1})\|_2\|f(\boldsymbol{x_1}) - f(\boldsymbol{x_2})\|_2\}$$

$$\leq \left\|\mathbb{E}_{\boldsymbol{x}}\mathbb{E}_{\boldsymbol{x'}\in\mathcal{A}(\boldsymbol{x})}\{f(\boldsymbol{x'})f(\boldsymbol{x'})^\top\} - I_{d^*}\right\|_F + B_2\mathbb{E}_{\boldsymbol{x}}\mathbb{E}_{\boldsymbol{x_1},\boldsymbol{x_2}}\|f(\boldsymbol{x_1}) - f(\boldsymbol{x_2})\|_2. \qquad (\|f\|_2 \leq B_2)$$

The above deduction indicates that finding a measurable map $f$ such that $B_1 \leq \|f\|_2 \leq B_2$ and ensuring both $\mathcal{L}_{\text{align}}(f)$ and $\left\|\mathbb{E}_{\boldsymbol{x}}\mathbb{E}_{\boldsymbol{x'}\in\mathcal{A}(\boldsymbol{x})}\{f(\boldsymbol{x'})f(\boldsymbol{x'})^\top\} - I_{d^*}\right\|_F$ vanish is sufficient to achieve our goal.

**Lemma A.14.** *If there exists a measurable map $f$ making $\Sigma = \mathbb{E}_{\boldsymbol{x}\sim\mathbb{P}_s}\{f(\boldsymbol{x})f(\boldsymbol{x})^\top\}$ positive definite, then there exists a measurable map $\tilde{f}$ such that*

$$\mathcal{L}_{\text{align}}(\tilde{f}) = 0, \quad \|\mathbb{E}_{\boldsymbol{x}}\mathbb{E}_{\boldsymbol{x'}\in\mathcal{A}(\boldsymbol{x})}\{\tilde{f}(\boldsymbol{x'})\tilde{f}(\boldsymbol{x'})^\top\} - I_{d^*}\|_F = 0.$$

*Proof.* We conduct modifications for given $f$ as follows: For any $\boldsymbol{x} \in \mathcal{X}$, define

$$\tilde{f}_{\boldsymbol{x}}(\boldsymbol{x'}) = \begin{cases} V^{-1}f(\boldsymbol{x}) & \text{if } \boldsymbol{x'} \in \mathcal{A}(\boldsymbol{x}) \\ f(\boldsymbol{x}) & \text{if } \boldsymbol{x'} \notin \mathcal{A}(\boldsymbol{x}) \end{cases}$$

where $\Sigma = VV^\top$, which is the Cholesky decomposition of $\Sigma$, which is evident well-defined as $\Sigma$ is positive definite. Iteratively repeat this argument for all $\boldsymbol{x} \in \mathcal{X}$ to yield $\tilde{f}$, then we have

$$\mathbb{E}_{\boldsymbol{x}}\mathbb{E}_{\boldsymbol{x'}\in\mathcal{A}(\boldsymbol{x})}\{\tilde{f}(\boldsymbol{x'})\tilde{f}(\boldsymbol{x'})^\top\} = V^{-1}\mathbb{E}_{\boldsymbol{x}}\{f(\boldsymbol{x})f(\boldsymbol{x})^\top\}V^{-T} = I_{d^*}$$

and

$$\forall \boldsymbol{x} \in \mathcal{X}, \boldsymbol{x_1}, \boldsymbol{x_2} \in \mathcal{A}(\boldsymbol{x}), \|\tilde{f}(\boldsymbol{x_1}) - \tilde{f}(\boldsymbol{x_2})\|_2 = \|\tilde{f}(\boldsymbol{x}) - \tilde{f}(\boldsymbol{x})\|_2 = 0.$$

That is precisely what we desire. $\qquad\square$

*Remark* A.15. If we have a measurable partition $\mathcal{X} = \cup_{i=1}^{d^*}\mathcal{P}_i$ stated in Assumption 3.3 such that $\mathcal{P}_i \cap \mathcal{P}_j = \emptyset$ and $\forall i \in [d^*], \frac{1}{B_2^2} \leq \mathbb{P}_s(\mathcal{P}_i) \leq \frac{1}{B_1^2}$, just set the $f(\boldsymbol{x}) = \boldsymbol{e}_i$ if $\boldsymbol{x} \in \mathcal{P}_i$, where $\boldsymbol{e}_i$ is the standard basis of $\mathbb{R}^{d^*}$, then $\Sigma = \text{diag}\{\mathbb{P}_s(\mathcal{P}_1), \ldots, \mathbb{P}_s(\mathcal{P}_i), \ldots, \mathbb{P}_s(\mathcal{P}_{d^*})\}, V^{-1} = \text{diag}\{\sqrt{\frac{1}{\mathbb{P}_s(\mathcal{P}_1)}}, \ldots, \sqrt{\frac{1}{\mathbb{P}_s(\mathcal{P}_i)}}, \ldots, \sqrt{\frac{1}{\mathbb{P}_s(\mathcal{P}_{d^*})}}\}, \tilde{f}(\boldsymbol{x}) = \sqrt{\frac{1}{\mathbb{P}_s(\mathcal{P}_i)}}\boldsymbol{e}_i$ if $\boldsymbol{x} \in \mathcal{P}_i$, it is obviously that $B_1 \leq \|\tilde{f}\|_2 \leq B_2$.

In this context, exploring the sample complexity of $\mathbb{E}_{\widetilde{D}_s}\{\mathcal{E}(\hat{f}_{n_s})\}$ is equivalent to investigating $\mathbb{E}_{\widetilde{D}_s}\{\mathcal{L}(\hat{f}_{n_s})\}$. However, the unbiasedness between $\widehat{\mathcal{L}}(f)$ and $\mathcal{L}(f)$ hinders our ability to analyze $\mathbb{E}_{\widetilde{D}_s}\{\mathcal{E}(\hat{f}_{n_s})\}$. To address this issue, we develop the following novel risk decomposition.decomposition.

A.3.3. RISK DECOMPOSITION

If denote $\widehat{G}(f) = \frac{1}{n_s} \sum_{i=1}^{n_s} f(\boldsymbol{x}_1^{(i)}) f(\boldsymbol{x}_2^{(i)})^\top - I_{d^*}$ and $G^*(f) = \mathbb{E}_{\boldsymbol{x}} \mathbb{E}_{\boldsymbol{x}_1, \boldsymbol{x}_2 \in \mathcal{A}(\boldsymbol{x})} [f(\boldsymbol{x}_1) f(\boldsymbol{x}_2)^\top] - I_{d^*}$, we can decompose $\mathcal{E}(\hat{f}_{n_s})$ into three terms shown as follow and then deal each term successively. To achieve conciseness in subsequent conclusions, we employ $X \lesssim Y$ or $Y \gtrsim X$ to indicate the statement that $X \leq CY$ form some $C > 0$ if $X$ and $Y$ are two quantities.

**Lemma A.16.** *The excess risk $\mathcal{E}(\hat{f}_{n_s})$ satisfies*

$$\mathbb{E}_{\widetilde{D}_s}\{\mathcal{E}(\hat{f}_{n_s})\} \lesssim \underbrace{2\mathbb{E}_{\widetilde{D}_s}\{\sup_{f \in \mathcal{F}, G \in \widehat{\mathcal{G}}(f)} |\mathcal{L}(f,G) - \widehat{\mathcal{L}}(f,G)|\}}_{\text{statistical error}\,:\,\mathcal{E}_{\text{sta}}} + \underbrace{\inf_{f \in \mathcal{F}}\{\mathcal{L}(f) - \mathcal{L}(f^*)\}}_{\text{approximation error of }\mathcal{F}\,:\,\mathcal{E}_{\mathcal{F}}} + \underbrace{\mathbb{E}_{\widetilde{D}_s}\big[\sup_{f \in \mathcal{F}}\{G^*(f) - \widehat{G}(f)\}\big]}_{\text{approximation error of }\mathcal{G}\,:\,\mathcal{E}_{\mathcal{G}}},$$

*That is,*

$$\mathcal{E}(\hat{f}_{n_s}) \leq \mathcal{E}_{\text{sta}} + \mathcal{E}_{\mathcal{F}} + \mathcal{E}_{\mathcal{G}}.$$

*Proof.* Recall $\mathcal{F} = \mathcal{N}\mathcal{N}_{d,d^*}(W, L, \mathcal{K}, B_1, B_2)$, for any $f \in \mathcal{F}$,

$$\mathcal{L}(\widehat{f}_{n_s}) - \mathcal{L}(f^*) = \sup_{G \in \mathcal{G}(\hat{f}_{n_s})} \mathcal{L}(\hat{f}_{n_s}, G) - \sup_{G \in \mathcal{G}(f^*)} \mathcal{L}(f^*, G)$$

$$= \Big[ \sup_{G \in \mathcal{G}(\hat{f}_{n_s})} \mathcal{L}(\hat{f}_{n_s}, G) - \sup_{G \in \widehat{\mathcal{G}}(\hat{f}_{n_s})} \mathcal{L}(\hat{f}_{n_s}, G) \Big] + \Big[ \sup_{G \in \widehat{\mathcal{G}}(\hat{f}_{n_s})} \mathcal{L}(\hat{f}_{n_s}, G) - \sup_{G \in \widehat{\mathcal{G}}(\hat{f}_{n_s})} \widehat{\mathcal{L}}(\hat{f}_{n_s}, G) \Big]$$

$$+ \Big[ \sup_{G \in \widehat{\mathcal{G}}(\hat{f}_{n_s})} \widehat{\mathcal{L}}(\hat{f}_{n_s}, G) - \sup_{G \in \widehat{\mathcal{G}}(f)} \widehat{\mathcal{L}}(f, G) \Big] + \Big[ \sup_{G \in \widehat{\mathcal{G}}(f)} \widehat{\mathcal{L}}(f, G) - \sup_{G \in \widehat{\mathcal{G}}(f)} \mathcal{L}(f, G) \Big]$$

$$+ \Big[ \sup_{G \in \widehat{\mathcal{G}}(f)} \mathcal{L}(f, G) - \sup_{G \in \mathcal{G}(f)} \mathcal{L}(f, G) \Big] + \Big[ \sup_{G \in \mathcal{G}(f)} \mathcal{L}(f, G) - \sup_{G \in \mathcal{G}(f^*)} \mathcal{L}(f^*, G) \Big],$$

where the second and fourth terms can be bounded by $\mathcal{E}_{\text{sta}}$. In fact, regarding to the fourth term, we have

$$\sup_{G \in \widehat{\mathcal{G}}(f)} \widehat{\mathcal{L}}(f, G) - \sup_{G \in \widehat{\mathcal{G}}(f)} \mathcal{L}(f, G) \leq \sup_{G \in \widehat{\mathcal{G}}(f)} \{\widehat{\mathcal{L}}(f, G) - \mathcal{L}(f, G)\} \leq \sup_{G \in \widehat{\mathcal{G}}(f)} |\widehat{\mathcal{L}}(f, G) - \mathcal{L}(f, G)|$$

$$\leq \sup_{f \in \mathcal{F}, G \in \widehat{\mathcal{G}}(f)} |\widehat{\mathcal{L}}(f, G) - \mathcal{L}(f, G)|,$$

and the same conclusion holds for the second term.

The summation of first term and fifth term can be bounded by $\mathcal{E}_{\mathcal{G}}$. Actually, for the first term

$$\sup_{G \in \mathcal{G}(\hat{f}_{n_s})} \mathcal{L}(\hat{f}_{n_s}, G) - \sup_{G \in \widehat{\mathcal{G}}(\hat{f}_{n_s})} \mathcal{L}(\hat{f}_{n_s}, G) \leq \sup_{f \in \mathcal{F}} \{ \sup_{G \in \mathcal{G}(f)} \mathcal{L}(f, G) - \sup_{G \in \widehat{\mathcal{G}}(f)} \mathcal{L}(f, G) \}$$

$$\leq \sup_{f \in \mathcal{F}} \{ \sup_{G \in \mathcal{G}(f)} \mathcal{L}(f, G) - \mathcal{L}(f, \widehat{G}(f)) \} = \sup_{f \in \mathcal{F}} \{ \mathcal{L}(f, G^*(f)) - \mathcal{L}(f, \widehat{G}(f)) \}$$

$$\leq \sqrt{2} B_2 \sup_{f \in \mathcal{F}} \|G^*(f) - \widehat{G}(f)\|_F \leq \sqrt{2} B_2 \sup_{f \in \mathcal{F}} \left\| \mathbb{E}_{\boldsymbol{x}} \mathbb{E}_{\boldsymbol{x}_1, \boldsymbol{x}_2 \in \mathcal{A}(\boldsymbol{x})} [f(\boldsymbol{x}_1) f(\boldsymbol{x}_2)^\top] - \frac{1}{n_s} \sum_{i=1}^{n_s} f(\boldsymbol{x}_1^{(i)}) f(\boldsymbol{x}_2^{(i)})^\top \right\|_F. \quad (30)$$

where the second inequity stems from $\widehat{G}(f) \in \widehat{\mathcal{G}}(f)$ and the third inequality is due to $\ell(\boldsymbol{u}, \cdot) \in \text{Lip}(\sqrt{2} B_2)$, as outlined in Table 3. and for the fifth term, we turn out

$$\sup_{G \in \widehat{\mathcal{G}}(f)} \mathcal{L}(f, G) - \sup_{G \in \mathcal{G}(f)} \mathcal{L}(f, G) = \sup_{G \in \widehat{\mathcal{G}}(f)} \mathbb{E}_{\widetilde{D}_s}\{\langle \widehat{G}(f), G \rangle_F\} - \sup_{G \in \mathcal{G}(f)} \langle G^*(f), G \rangle_F$$

$$\leq \mathbb{E}_{\widetilde{D}_s}\{ \sup_{G \in \widehat{\mathcal{G}}(f)} \langle \widehat{G}(f), G \rangle_F \} - \sup_{G \in \mathcal{G}(f)} \langle G^*(f), G \rangle_F = \mathbb{E}_{\widetilde{D}_s}\{\|\widehat{G}(f)\|_F^2\} - \|G^*(f)\|_F^2$$

$$\leq 2(B_2^2 + \sqrt{d^*})\big(\mathbb{E}_{\widetilde{D}_s}\{\|\widehat{G}(f)\|_F\} - \|G^*(f)\|_F\big) \leq 2(B_2^2 + \sqrt{d^*})\big( \sup_{f \in \mathcal{F}} [\mathbb{E}_{\widetilde{D}_s}\{\|\widehat{G}(f)\|_F\} - \|G^*(f)\|_F]\big)$$

$$\lesssim \sup_{f \in \mathcal{F}} \left\{ \mathbb{E}_{\widetilde{D}_s} \left[ \left\| \frac{1}{n_s} \sum_{i=1}^{n_s} f(\boldsymbol{x}_1^{(i)}) f(\boldsymbol{x}_2^{(i)})^\top - I_{d^*} \right\|_F - \left\| \mathbb{E}_{\boldsymbol{x}} \mathbb{E}_{\boldsymbol{x}_1, \boldsymbol{x}_2 \in \mathcal{A}(\boldsymbol{x})} \{ f(\boldsymbol{x}_1) f(\boldsymbol{x}_2)^\top \} - I_{d^*} \right\|_F \right] \right\}$$

$$\leq \sup_{f \in \mathcal{F}} \left\{ \mathbb{E}_{\widetilde{D}_s} \left[ \left\| \frac{1}{n_s} \sum_{i=1}^{n_s} f(\boldsymbol{x}_1^{(i)}) f(\boldsymbol{x}_2^{(i)})^\top - \mathbb{E}_{\boldsymbol{x}} \mathbb{E}_{\boldsymbol{x}_1, \boldsymbol{x}_2 \in \mathcal{A}(\boldsymbol{x})} \{ f(\boldsymbol{x}_1) f(\boldsymbol{x}_2)^\top \} \right\|_F \right] \right\}$$

$$\leq \mathbb{E}_{\widetilde{D}_s} \left[ \sup_{f \in \mathcal{F}} \left\{ \left\| \frac{1}{n_s} \sum_{i=1}^{n_s} f(\boldsymbol{x}_1^{(i)}) f(\boldsymbol{x}_2^{(i)})^\top - \mathbb{E}_{\boldsymbol{x}} \mathbb{E}_{\boldsymbol{x}_1, \boldsymbol{x}_2 \in \mathcal{A}(\boldsymbol{x})} \{ f(\boldsymbol{x}_1) f(\boldsymbol{x}_2)^\top \} \right\|_F \right\} \right] \tag{31}$$

where the first equality is due to $\langle G^*(f), G \rangle_F = \mathbb{E}_{\widetilde{D}_s} \{ \langle \widehat{G}(f), G \rangle_F \}$ and the second inequality is derived from the facts that $\|\widehat{G}(f)\|_F \leq B_2^2 + \sqrt{d^*}$ and $\|G^*(f)\|_F \leq B_2^2 + \sqrt{d^*}$. Combining (30) and (31) yields $\mathbb{E}_{\widetilde{D}_s} \{ \mathcal{E}_{\mathcal{G}} \}$.

Furthermore, the third term $\sup_{G \in \widehat{\mathcal{G}}(\hat{f}_{n_s})} \widehat{\mathcal{L}}(\hat{f}_{n_s}, G) - \sup_{G \in \widehat{\mathcal{G}}(f)} \widehat{\mathcal{L}}(f, G) \leq 0$ because of the definition of $\hat{f}_{n_s}$. Taking infimum over all $f \in \mathcal{NN}_{d,d^*}(W, L, \mathcal{K}, B_1, B_2)$ yields

$$\mathcal{E}(\hat{f}_{n_s}) \lesssim \mathcal{E}_{\text{sta}} + \mathcal{E}_{\mathcal{F}} + \mathcal{E}_{\mathcal{G}},$$

which completes the proof. $\qquad\square$

A.3.4. BOUND $\mathcal{E}_{\text{sta}}$

**Lemma A.17.** *Regarding to $\mathcal{E}_{\text{sta}}$, we have*

$$\mathbb{E}_{\widetilde{D}_s} [\mathcal{E}_{\text{sta}}] \lesssim \frac{\mathcal{K}\sqrt{L}}{\sqrt{n_s}}.$$

*Proof.* We are going to be introducing the relevant notations at first.

For any $f : \mathbb{R}^d \to \mathbb{R}^{d^*}$, let $\widetilde{f} : \mathbb{R}^{2d} \to \mathbb{R}^{2d^*}$ such that $\widetilde{f}(\widetilde{\boldsymbol{x}}) = (f(\boldsymbol{x}_1), f(\boldsymbol{x}_2))$, where $\widetilde{\boldsymbol{x}} = (\boldsymbol{x}_1, \boldsymbol{x}_2) \in \mathbb{R}^{2d}$. Furthermore, define $\widetilde{\mathcal{F}} := \{ \widetilde{f} : f \in \mathcal{NN}_{d,d^*}(W, L, \mathcal{K}) \}$ and denote $D'_s = \{ \widetilde{\boldsymbol{x}}'^{(i)} \}_{i=1}^{n_s}$ as an independent identically distributed samples to $D_s$, which is called as ghost samples of $D_s$.

Next, we are attempt to establish the relationship between $\mathbb{E}_{\widetilde{D}_s}[\mathcal{E}_{\text{sta}}]$ and the Rademacher complexity of $\mathcal{NN}_{d,d^*}(W, L, \mathcal{K})$. By the definition of $\mathcal{E}_{\text{sta}}$, we have

$$\mathbb{E}_{\widetilde{D}_s} [\mathcal{E}_{\text{sta}}] = \mathbb{E}_{\widetilde{D}_s} \left[ \sup_{f \in \mathcal{NN}_{d,d^*}(W,L,\mathcal{K},B_1,B_2), G \in \widehat{\mathcal{G}}(f)} |\mathcal{L}(f, G) - \widehat{\mathcal{L}}(f, G)| \right]$$

$$\leq \mathbb{E}_{\widetilde{D}_s} \left[ \sup_{(f,G) \in \mathcal{NN}_{d,d^*}(W,L,\mathcal{K},B_1,B_2) \times \mathcal{G}_1} |\mathcal{L}(f, G) - \widehat{\mathcal{L}}(f, G)| \right]$$

$$\qquad\qquad (\text{As } \widehat{\mathcal{G}}(f) \subseteq \mathcal{G}_1 \text{ for any } f \in \mathcal{NN}_{d,d^*}(W, L, \mathcal{K}, B_1, B_2))$$

$$\leq \mathbb{E}_{\widetilde{D}_s} \left[ \sup_{(f,G) \in \mathcal{NN}_{d,d^*}(W,L,\mathcal{K}) \times \mathcal{G}_1} |\mathcal{L}(f, G) - \widehat{\mathcal{L}}(f, G)| \right]$$

$$\qquad\qquad (\text{As } \mathcal{NN}_{d,d^*}(W, L, \mathcal{K}, B_1, B_2) \subseteq \mathcal{NN}_{d,d^*}(W, L, \mathcal{K}))$$

$$= \mathbb{E}_{\widetilde{D}_s} \left[ \sup_{(\widetilde{f},G) \in \widetilde{\mathcal{F}} \times \mathcal{G}_1} \left| \frac{1}{n_s} \sum_{i=1}^{n_s} \mathbb{E}_{D'_s} [\ell(\widetilde{f}(\widetilde{\boldsymbol{x}}'^{(i)}), G)] - \frac{1}{n_s} \sum_{i=1}^{n_s} \ell(\widetilde{f}(\widetilde{\boldsymbol{x}}^{(i)}), G) \right| \right]$$

$$\leq \mathbb{E}_{D_s, D'_s} \left[ \sup_{(\widetilde{f},G) \in \widetilde{\mathcal{F}} \times \mathcal{G}_1} \left| \frac{1}{n_s} \sum_{i=1}^{n_s} \ell(\widetilde{f}(\widetilde{\boldsymbol{x}}'^{(i)}), G) - \frac{1}{n_s} \sum_{i=1}^{n_s} \ell(\widetilde{f}(\widetilde{\boldsymbol{x}}^{(i)}), G) \right| \right]$$

$$= \mathbb{E}_{D_s, D'_s, \xi} \left[ \sup_{(\widetilde{f},G) \in \widetilde{\mathcal{F}} \times \mathcal{G}_1} \left| \frac{1}{n_s} \sum_{i=1}^{n_s} \xi_i \left( \ell(\widetilde{f}(\widetilde{\boldsymbol{x}}'^{(i)}), G) - \ell(\widetilde{f}(\widetilde{\boldsymbol{x}}^{(i)}), G) \right) \right| \right] \tag{32}$$

$$\leq 2 \mathbb{E}_{D_s, \xi} \left[ \sup_{(\widetilde{f},G) \in \widetilde{\mathcal{F}} \times \mathcal{G}_1} \left| \frac{1}{n_s} \sum_{i=1}^{n_s} \xi_i \ell(\widetilde{f}(\widetilde{\boldsymbol{x}}^{(i)}), G) \right| \right]$$

$$\leq 4\sqrt{2}\|\ell\|_{\mathrm{Lip}}\Big(\mathbb{E}_{D_s,\xi}\big[\sup_{f\in\mathcal{NN}_{d,d^*}(W,L,\mathcal{K})}\Big|\frac{1}{n_s}\sum_{i=1}^{n_s}\sum_{j=1}^{d^*}\xi_{i,j,1}f_j(\boldsymbol{x}_1^{(i)})+\xi_{i,j,2}f_j(\boldsymbol{x}_2^{(i)})\Big|\big]$$

$$+\mathbb{E}_\xi\big[\sup_{G\in\mathcal{G}_1}\Big|\frac{1}{n_s}\sum_{i=1}^{n_s}\sum_{j=1}^{d^*}\sum_{k=1}^{d^*}\xi_{i,j,k}G_{jk}\Big|\big]\Big) \tag{33}$$

$$\leq 8\sqrt{2}\|\ell\|_{\mathrm{Lip}}\mathbb{E}_{D_s,\xi}\big[\sup_{f\in\mathcal{NN}_{d,d^*}(W,L,\mathcal{K})}\Big|\frac{1}{n_s}\sum_{i=1}^{n_s}\sum_{j=1}^{d^*}\xi_{i,j,1}f_j(\boldsymbol{x}_1^{(i)})\Big|\big]+4\sqrt{2}d^*\|\ell\|_{\mathrm{Lip}}\varrho$$

$$+4\sqrt{2}\|\ell\|_{\mathrm{Lip}}\mathbb{E}_\xi\big[\max_{G\in\mathcal{N}_{\mathcal{G}_1}(\varrho)}\Big|\frac{1}{n_s}\sum_{i=1}^{n_s}\sum_{j=1}^{d^*}\sum_{k=1}^{d^*}\xi_{i,j,k}G_{jk}\Big|\big] \tag{34}$$

$$\leq 8\sqrt{2}\|\ell\|_{\mathrm{Lip}}\mathbb{E}_{D_s,\xi}\big[\sup_{f\in\mathcal{NN}_{d,d^*}(W,L,\mathcal{K})}\Big|\frac{1}{n_s}\sum_{i=1}^{n_s}\sum_{j=1}^{d^*}\xi_{i,j}f_j(\boldsymbol{x}_1^{(i)})\Big|\big]+4\sqrt{2}d^*\|\ell\|_{\mathrm{Lip}}\varrho$$

$$+4\sqrt{2}(B_2^2+\sqrt{d^*})\|\ell\|_{\mathrm{Lip}}\sqrt{\frac{2\log\big(2|\mathcal{N}_{\mathcal{G}_1}(\varrho)|\big)}{n_s}} \tag{35}$$

$$\leq 8\sqrt{2}d^*\|\ell\|_{\mathrm{Lip}}\mathbb{E}_{D_s,\xi}\big[\sup_{f\in\mathcal{NN}_{d,1}(W,L,\mathcal{K})}\Big|\frac{1}{n_s}\sum_{i=1}^{n_s}\xi_i f(\boldsymbol{x}_1^{(i)})\Big|\big]+4\sqrt{2}d^*\|\ell\|_{\mathrm{Lip}}\varrho$$

$$+4\sqrt{2}(B_2^2+\sqrt{d^*})\|\ell\|_{\mathrm{Lip}}\sqrt{\frac{2\log\big(2(\frac{3}{(B_2^2+\sqrt{d^*})\varrho})^{(d^*)^2}\big)}{n_2}} \qquad (|\mathcal{N}_{\mathcal{G}_1}(\varrho)|\leq(\frac{3}{(B_2^2+\sqrt{d^*})\varrho})^{(d^*)^2})$$

$$\lesssim \frac{\mathcal{K}\sqrt{L}}{\sqrt{n_s}}+\sqrt{\frac{\log n_s}{n_s}} \qquad (\text{Lemma A.12 and set }\varrho=\mathcal{O}(1/\sqrt{n_s}))$$

$$\lesssim \frac{\mathcal{K}\sqrt{L}}{\sqrt{n_s}} \qquad (\text{If }\mathcal{K}\gtrsim\sqrt{\log n_s})$$

Where (32) stems from the fact that $\xi_i\big(\ell(\widetilde{f}(\tilde{\boldsymbol{x}}'^{(i)}),G)-\ell(\widetilde{f}(\tilde{\boldsymbol{x}}^{(i)}),G)\big)$ has identical distribution with $\ell(\widetilde{f}(\tilde{\boldsymbol{x}}'^{(i)}),G)-\ell(\widetilde{f}(\tilde{\boldsymbol{x}}^{(i)}),G)$. As we have shown that $\|\ell\|_{\mathrm{Lip}}<\infty$, just apply Lemma A.9 to obtain (33). Regarding 34, as $\mathcal{N}_{\mathcal{G}_1}(\epsilon_1)$ is a $\epsilon_1$-covering, for any fixed $G\in\mathcal{G}_1$, we can find a $H_G\in\mathcal{N}_{\mathcal{G}_1}(\epsilon_1)$ satisfying $\|G-H_G\|_F\leq\epsilon_1$, therefore we have

$$\mathbb{E}_\xi\big[\max_{G\in\mathcal{G}_1}\Big|\frac{1}{n_s}\sum_{i=1}^{n_s}\sum_{j=1}^{d^*}\sum_{k=1}^{d^*}\xi_{i,j,k}\big\{(H_G)_{jk}+G_{jk}-(H_G)_{jk}\big\}\Big|\big]$$

$$\leq \mathbb{E}_\xi\big\{\max_{G\in\mathcal{G}_1}\Big|\frac{1}{n_s}\sum_{i=1}^{n_s}\sum_{j=1}^{d^*}\sum_{k=1}^{d^*}\xi_{i,j,k}(H_G)_{jk}\Big|\big\}+\mathbb{E}_\xi\big\{\max_{G\in\mathcal{G}_1}\Big|\frac{1}{n_s}\sum_{i=1}^{n_s}\sum_{j=1}^{d^*}\sum_{k=1}^{d^*}\xi_{i,j,k}\big\{G_{jk}-(H_G)_{jk}\big\}\Big|\big\}$$

$$\leq \mathbb{E}_\xi\big\{\max_{G\in\mathcal{N}_{\mathcal{G}_1}(\epsilon_1)}\Big|\frac{1}{n_s}\sum_{i=1}^{n_s}\sum_{j=1}^{d^*}\sum_{k=1}^{d^*}\xi_{i,j,k}G_{jk}\Big|\big\}+\frac{1}{n_s}\sqrt{(d^*)^2 n_s}\sqrt{n_s\sum_{j=1}^{d^*}\sum_{k=1}^{d^*}\big\{G_{jk}-(H_G)_{jk}\big\}^2}$$

$$(\text{Cauchy-Schwarz inequality})$$

$$\leq \mathbb{E}_\xi\big\{\max_{G\in\mathcal{N}_{\mathcal{G}_1}(\epsilon_1)}\Big|\frac{1}{n_s}\sum_{i=1}^{n_s}\sum_{j=1}^{d^*}\sum_{k=1}^{d^*}\xi_{i,j,k}G_{jk}\Big|\big\}+d^*\epsilon_1.$$

To turn out the last term of (35), notice that $\|G\|_F\leq B_2^2+\sqrt{d^*}$ implies that $\sum_{j=1}^{d^*}\sum_{k=1}^{d^*}\xi_{i,j,k}G_{jk}\sim\mathrm{subG}(B_2^2+\sqrt{d^*})$, therefore $\frac{1}{n_s}\sum_{i=1}^{n_s}\sum_{j=1}^{d^*}\sum_{k=1}^{d^*}\xi_{i,j,k}G_{jk}\sim\mathrm{subG}(B_2^2+\sqrt{d^*})$, just apply Lemma A.11 to finish the proof. $\square$

A.3.5. BOUND $\mathcal{E}_{\mathcal{F}}$

If we denote

$$\mathcal{E}(\mathcal{H}^\alpha, \mathcal{N}\mathcal{N}_{d,1}(W, L, \mathcal{K})) := \sup_{g \in \mathcal{H}^\alpha} \inf_{f \in \mathcal{N}\mathcal{N}_{d,1}(W,L,\mathcal{K})} \|f - g\|_{C([0,1]^d)},$$

where $C([0,1]^d)$ is the space of continuous functions on $[0,1]^d$ equipped with the sup-norm. Theorem 3.2 of Jiao et al. (2023) has already proven $\mathcal{E}(\mathcal{H}^\alpha, \mathcal{N}\mathcal{N}_{d,1}(W, L, \mathcal{K}))$ can be bound by a quantity related to $\mathcal{K}$ when setting appropriate architecture of network, that is

**Lemma A.18** (Theorem 3.2 of Jiao et al. (2023)). *Let $d \in \mathbb{N}$ and $\alpha = r + \beta > 0$, where $r \in \mathbb{N}_0$ and $\beta \in (0, 1]$. There exists $c > 0$ such that for any $\mathcal{K} \geq 1$, any $W \geq c\mathcal{K}^{(2d+\alpha)/(2d+2)}$ and $L \geq 2\lceil \log_2(d+r) \rceil + 2$,*

$$\mathcal{E}(\mathcal{H}^\alpha, \mathcal{N}\mathcal{N}_{d,1}(W, L, \mathcal{K})) \lesssim \mathcal{K}^{-\alpha/(d+1)}.$$

For utilizing this conclusion, first notice that

$$\inf_{f \in \mathcal{N}\mathcal{N}_{d,d^*}(W,L,\mathcal{K})} \|f(\boldsymbol{u}) - f^*(\boldsymbol{u})\|_2 = \inf_{f \in \mathcal{N}\mathcal{N}_{d,d^*}(W,L,\mathcal{K})} \sqrt{\sum_{i=1}^{d^*} \{f_i(\boldsymbol{u}) - f_i^*(\boldsymbol{u})\}^2}$$

$$\leq \inf_{f \in \mathcal{N}\mathcal{N}_{d,d^*}(W,L,\mathcal{K})} \sqrt{\sum_{i=1}^{d^*} \|f_i - f_i^*\|_{C([0,1]^d)}^2} \leq \sup_{g \in \mathcal{H}^\alpha} \inf_{f \in \mathcal{N}\mathcal{N}_{d,d^*}(W,L,\mathcal{K})} \sqrt{\sum_{i=1}^{d^*} \|f_i - g\|_{C([0,1]^d)}^2}$$

$$\leq \sup_{g \in \mathcal{H}^\alpha} \sqrt{\sum_{i=1}^{d^*} \inf_{f \in \mathcal{N}\mathcal{N}_{d,1}(\lfloor W/d^* \rfloor, L, \mathcal{K})} \|f - g\|_{C([0,1]^d)}^2} \leq \sqrt{d^*}\mathcal{E}(\mathcal{H}^\alpha, \mathcal{N}\mathcal{N}_{d,1}(\lfloor W/d^* \rfloor, L, \mathcal{K})) \lesssim \mathcal{K}^{-\alpha/(d+1)},$$

where the third to last line inequality is from following reason: if $f_i \in \mathcal{N}\mathcal{N}_{d,1}(\lfloor W/d^* \rfloor, L, \mathcal{K})$, where $i \in [d^*]$, whose parameter are independent with each other, then their concatenation $f = (f_1, f_2, \cdots, f_{d^*})^\top$ can be regarded as an elements of $\mathcal{N}\mathcal{N}_{d,d^*}(W, D, \mathcal{K})$ with specific parameters, by following Proposition A.19, we have $f \in \mathcal{N}\mathcal{N}_{d,d^*}(W, L, \mathcal{K})$.

**Proposition A.19** ((iii) of Proposition 2.5 in Jiao et al. (2023)). *Let $f_1 \in \mathcal{N}\mathcal{N}_{d,d_1^*}(w_1, L_1, \mathcal{K}_1)$ and $f_2 \in \mathcal{N}\mathcal{N}_{d,d_2^*}(W_2, L_2, \mathcal{K}_2)$, define $f(\boldsymbol{x}) := (f_1(\boldsymbol{x}), f_2(\boldsymbol{x}))$, then $f \in \mathcal{N}\mathcal{N}_{d,d_1^*+d_2^*}(W_1 + W_2, \max\{L_1, L_2\}, \max\{\mathcal{K}_1, \mathcal{K}_2\})$.*

Above conclusion implies optimal approximation element of $f^*$ in $\mathcal{N}\mathcal{N}_{d,d^*}(W, L, \mathcal{K})$ can be arbitrarily close to $f^*$ under the setting that $\mathcal{K}$ is large enough. Hence we can conclude optimal approximation element of $f^*$ is also contained in $\mathcal{F} = \mathcal{N}\mathcal{N}_{d,d^*}(W, L, \mathcal{K}, B_1, B_2)$ as the setting that $B_1 \leq \|f^*\|_2 \leq B_2$.

Therefore, if we denote

$$\mathcal{T}(f) := \mathbb{E}_{\boldsymbol{x}}\mathbb{E}_{\boldsymbol{x}_1,\boldsymbol{x}_2 \in \mathcal{A}(\boldsymbol{x})}\{\|f(\boldsymbol{x}_1) - f(\boldsymbol{x}_2)\|_2^2\} + \lambda\|\mathbb{E}_{\boldsymbol{x}}\mathbb{E}_{\boldsymbol{x}_1,\boldsymbol{x}_2 \in \mathcal{A}(\boldsymbol{x})}\{f(\boldsymbol{x}_1)f(\boldsymbol{x}_2)^\top\} - I_{d^*}\|_F^2,$$

we can yield the upper bound of $\mathcal{E}_{\mathcal{F}}$ by following deduction

$$\mathcal{E}_{\mathcal{F}} = \inf_{f \in \mathcal{F}}\{\sup_{G \in \mathcal{G}(f)} \mathcal{L}(f, G) - \sup_{G \in \mathcal{G}(f^*)} \mathcal{L}(f^*, G)\} = \inf_{f \in \mathcal{F}}\{\mathcal{T}(f) - \mathcal{T}(f^*)\} = \inf_{f \in \mathcal{N}\mathcal{N}_{d,d^*}(W,L,\mathcal{K})}\{\mathcal{T}(f) - \mathcal{T}(f^*)\}$$

$$\leq \|\ell\|_{\mathrm{Lip}} \inf_{f \in \mathcal{N}\mathcal{N}_{d,d^*}(W,L,\mathcal{K})} \mathbb{E}_{\boldsymbol{x}}\mathbb{E}_{\tilde{\boldsymbol{x}}}\|\tilde{f}(\tilde{\boldsymbol{x}}) - \tilde{f}^*(\tilde{\boldsymbol{x}})\|_2 \leq \|\ell\|_{\mathrm{Lip}} \inf_{f \in \mathcal{N}\mathcal{N}_{d,d^*}(W,L,\mathcal{K})} \mathbb{E}_{\boldsymbol{x}}\mathbb{E}_{\boldsymbol{x}' \in \mathcal{A}(\boldsymbol{x})} \sqrt{2\sum_{i=1}^{d^*}\{f_i(\boldsymbol{x}') - f_i^*(\boldsymbol{x}')\}^2}$$

$$\leq \sqrt{2d^*}\|\ell\|_{\mathrm{Lip}} \sup_{g \in \mathcal{H}^\alpha} \inf_{f \in \mathcal{N}\mathcal{N}_{d,1}(\lfloor W/d^* \rfloor, L, \mathcal{K}/\sqrt{d^*})} \|f - g\|_{C([0,1]^d)} \leq \sqrt{2d^*}\|\ell\|_{\mathrm{Lip}}\mathcal{E}(\mathcal{H}^\alpha, \mathcal{N}\mathcal{N}_{d,1}(\lfloor W/d^* \rfloor, L, \mathcal{K}/\sqrt{d^*}))$$

$$\lesssim \mathcal{K}^{-\alpha/(d+1)}.$$

where the first inequality is because of Proposition A.7.

### A.3.6. BOUND $\mathcal{E}_\mathcal{G}$

Let $\mathcal{M}(\boldsymbol{u}) = \boldsymbol{u}_1\boldsymbol{u}_2^\top$, $\boldsymbol{u}_1, \boldsymbol{u}_2 \in \mathbb{R}^{d^*}$, which is a Lipchitz map on $\{\boldsymbol{u} \in \mathbb{R}^{2d^*} : \boldsymbol{u} \leq \sqrt{2}B_2\}$, as presented in Proposition A.7. Then

$$\mathbb{E}_{\widetilde{D}_s}\{\mathcal{E}_\mathcal{G}\} \lesssim \mathbb{E}_{\widetilde{D}_s}\left[\sup_{f \in \mathcal{F}} \left\|\mathbb{E}_{\boldsymbol{x}}\mathbb{E}_{\boldsymbol{x}_1, \boldsymbol{x}_2 \in \mathcal{A}(\boldsymbol{x})}\left[\frac{1}{n_s}\sum_{i=1}^{n_s}\{\mathcal{M}(\widetilde{f}(\tilde{\boldsymbol{x}})) - \mathcal{M}(\widetilde{f}(\tilde{\boldsymbol{x}}^{(i)}))\}\right]\right\|_F\right]$$

$$\leq \|\mathcal{M}\|_{\mathrm{Lip}}\mathbb{E}_{\widetilde{D}_s}\left[\left\|\mathbb{E}_{\boldsymbol{x}}\mathbb{E}_{\boldsymbol{x}_1, \boldsymbol{x}_2 \in \mathcal{A}(\boldsymbol{x})}\{\widetilde{f}(\tilde{\boldsymbol{x}})\} - \frac{1}{n_s}\sum_{i=1}^{n_s}\widetilde{f}(\tilde{\boldsymbol{x}}^{(i)})\right\|_2\right]$$

Furthermore, according to the multidimensional Chebyshev's inequality, we turn out that $\mathbb{P}_s\left(\left\|\frac{1}{n_s}\sum_{i=1}^{n_s}\widetilde{f}(\tilde{\boldsymbol{x}}^{(i)}) - \mathbb{E}_{\boldsymbol{x}}\mathbb{E}_{\boldsymbol{x}_1, \boldsymbol{x}_2 \in \mathcal{A}(\boldsymbol{x})}\{\widetilde{f}(\tilde{\boldsymbol{x}})\}\right\|_2 \geq \frac{1}{n_s^{1/4}}\right) \leq \frac{\mathbb{E}\|\widetilde{f}(\tilde{\boldsymbol{x}}) - \mathbb{E}\{\widetilde{f}(\tilde{\boldsymbol{x}})\}\|_2^2}{\sqrt{n_s}} \leq \frac{8B_2^2}{\sqrt{n_s}}$ as $\|\widetilde{f}(\tilde{\boldsymbol{x}})\|_2 \leq \sqrt{2}B_2$. Thus we have

$$\mathbb{E}_{\widetilde{D}_s}\{\mathcal{E}_\mathcal{G}\} \lesssim \frac{1}{n_s^{1/4}} \cdot \mathbb{P}_s\left(\left\|\frac{1}{n_s}\sum_{i=1}^{n_s}\widetilde{f}(\tilde{\boldsymbol{x}}^{(i)}) - \mathbb{E}_{\boldsymbol{x}}\mathbb{E}_{\boldsymbol{x}_1, \boldsymbol{x}_2 \in \mathcal{A}(\boldsymbol{x})}\{\widetilde{f}(\tilde{\boldsymbol{x}})\}\right\|_2 \geq \frac{1}{n_s^{1/4}}\right) + 2\sqrt{2}B_2 \cdot \frac{8B_2^2}{\sqrt{n_s}}$$

$$\leq \frac{1}{n_s^{1/4}} + 16\sqrt{2}B_2^3\frac{1}{\sqrt{n_s}} \lesssim \frac{1}{n_s^{1/4}}.$$

where the first inequity is due to $\|\widetilde{f}(\tilde{\boldsymbol{x}})\|_2 \leq \sqrt{2}B_2$.

### A.3.7. SUBSECTION: TRADE OFF BETWEEN STATISTICAL ERROR AND APPROXIMATION ERROR

Let $W \geq c\mathcal{K}^{(2d+\alpha)/(2d+2)}$ and $L \geq 2\lceil\log_2(d+r)\rceil + 2$, combine the bound results of statistical error and approximation error to yield

$$\mathbb{E}_{\widetilde{D}_s}\{\mathcal{E}(\hat{f}_{n_s})\} \lesssim \mathbb{E}_{\widetilde{D}_s}[\mathcal{E}_{\mathrm{sta}}] + \mathcal{E}_\mathcal{F} + \mathbb{E}_{\widetilde{D}_s}\{\mathcal{E}_\mathcal{G}\} \lesssim \frac{\mathcal{K}}{\sqrt{n_s}} + \mathcal{K}^{-\alpha/(d+1)}.$$

Taking $\mathcal{K} = n_s^{\frac{d+1}{2(\alpha+d+1)}}$ to yield $\mathbb{E}_{\widetilde{D}_s}\{\mathcal{E}(\hat{f}_{n_s})\} \lesssim n_s^{-\frac{\alpha}{2(\alpha+d+1)}}$. As we have shown that $\mathcal{L}(f^*) = 0$, above inequality implies $\mathbb{E}_{\widetilde{D}_s}\{\mathcal{L}(\hat{f}_{n_s})\} \lesssim n_s^{-\frac{\alpha}{2(\alpha+d+1)}}$. To ensure above deduction holds, We need to set $W \geq cn_s^{\frac{2d+\alpha}{4(\alpha+d+1)}}$ and $L \geq 2\lceil\log_2(d+r)\rceil + 2$.

### A.3.8. THE PROOF OF MAIN THEOREM

Next, we are going to prove our main theorem 3.9. We will state its formal version at first and then conclude Theorem 3.9 as a corollary.

To notation conciseness, let $p = \dfrac{\sqrt{\frac{2}{\min_{i \neq j} p_s(i)p_s(j)}\left(\frac{C}{\bar{\lambda}}n_s^{-\frac{\alpha}{2(\alpha+d+1)}} + \phi(n_s)\right)} + 2\sqrt{d^*}B_2Mn_s^{-\frac{\nu}{2(\alpha+d+1)}}}{B_2^2\psi(\sigma_s^{(n_s)}, \delta_s^{(n_s)}, \varepsilon_{n_s}, \hat{f}_{n_s})}$, where $C$ is a constant, $0 \leq \phi(n_s) \lesssim (1 - \sigma_s^{(n_s)} + n_s^{-\frac{\min\{\alpha, \nu, \varsigma, \tau\}}{4(\alpha+d+1)}})^2 + (1 - \sigma_s^{(n_s)}) + n_s^{-\frac{\min\{\alpha, \nu, \varsigma, \tau\}}{8(\alpha+d+1)}}$, then the formal version of our main theoretical result can be stated as follow.

**Lemma A.20.** *When Assumptions 3.5, 3.3, 3.2, 3.7 and 3.8 all hold, set $\varepsilon_{n_s} = m^2n_s^{-\frac{\min\{\alpha, \nu, \varsigma, \tau\}}{8(\alpha+d+1)}}, W \geq cn_s^{\frac{2d+\alpha}{4(\alpha+d+1)}}$, $L \geq 2\lceil\log_2(d+r)\rceil + 2, \mathcal{K} = n_s^{\frac{d+1}{2(\alpha+d+1)}}$ and $\mathcal{A} = \mathcal{A}_{n_s}$ in Assumption 3.7, then we have*

$$\mathbb{E}_{\widetilde{D}_s}[R_t^2(\varepsilon_{n_s}, \hat{f}_{n_s})] \lesssim n_s^{-\frac{\min\{\alpha, \nu, \varsigma\}}{4(\alpha+d+1)}} \tag{36}$$

*and*

$$\mathbb{E}_{\widetilde{D}_s}\{\max_{i \neq j}|\mu_t(i)^\top\mu_t(j)|\} \lesssim (1 - \sigma_s^{(n_s)}) + n_s^{-\frac{\min\{\alpha, 2\tau\}}{4(\alpha+d+1)}}. \tag{37}$$

*Furthermore, If $\psi(\sigma_s^{(n_s)}, \delta_s^{(n_s)}, \varepsilon_{n_s}, \hat{f}_{n_s}) > 0$, then with probability at least $1 - p$, we have*

$$\mathbb{E}_{\widetilde{D}_s}\{\mathrm{Err}(Q_{\hat{f}_{n_s}})\} \leq (1 - \sigma_t^{(n_s)}) + \mathcal{O}(n_s^{-\frac{\min\{\alpha, \nu, \varsigma\}}{8(\alpha+d+1)}}).$$

*Proof.* First recall the conclusion we've got in Lemma A.6

$$\mathbb{E}_{\widetilde{D}_s}\{R_t^2(\varepsilon, \hat{f}_{n_s})\} \leq \frac{m^4}{\varepsilon^2}\big(\mathbb{E}_{\widetilde{D}_s}\{\mathcal{L}(\hat{f}_{n_s})\} + 8B_2 d^* M \mathcal{K}\epsilon_1 + 4B_2^2 d^* K \epsilon_2\big),$$

$$\mathbb{E}_{\widetilde{D}_s}[\max_{i \neq j}|\mu_t(i)^\top \mu_t(j)|] \leq \sqrt{\frac{2}{\min\limits_{i \neq j} p_s(i)p_s(j)}\big(\frac{1}{\lambda}\mathbb{E}_{\widetilde{D}_s}\{\mathcal{L}(\hat{f}_{n_s})\} + \mathbb{E}_{\widetilde{D}_s}\{\phi(\sigma_s, \delta_s, \varepsilon, \hat{f}_{n_s})\}\big)} + 2\sqrt{d^*}B_2 M\mathcal{K}\epsilon_1,$$

and with probability at least $1 - \dfrac{\sqrt{\frac{2}{\min_{i \neq j} p_s(i)p_s(j)}\left(\frac{1}{\lambda}\mathbb{E}_{\widetilde{D}_s}\{\mathcal{L}(\hat{f}_{n_s})\} + \phi(\sigma_s, \delta_s, \varepsilon, \hat{f}_{n_s})\right)} + 2\sqrt{d^*}B_2 M\mathcal{K}\epsilon_1}{B_2^2 \psi(\sigma_t, \delta_t, \varepsilon, \hat{f}_{n_s})}$, we have

$$\mathbb{E}_{\widetilde{D}_s}\{\mathrm{Err}(Q_{\hat{f}_{n_s}})\} \leq (1 - \sigma_t) + \frac{m^2}{\varepsilon}\sqrt{\mathbb{E}_{\widetilde{D}_s}\{\mathcal{L}(\hat{f}_{n_s})\} + 8B_2 d^* M\mathcal{K}\epsilon_1 + 4B_2^2 d^* K \epsilon_2},$$

where $\phi(\sigma_s, \delta_s, \varepsilon, \hat{f}_{n_s}) = 4B_2^2\Big[\big(1 - \sigma_s + \frac{\mathcal{K}\delta_s + 2\varepsilon}{2B_2}\big)^2 + (1 - \sigma_s) + \frac{Km^2}{\varepsilon}\sqrt{\mathbb{E}_{\widetilde{D}_s}\{\mathcal{L}(\hat{f}_{n_s})\}}\big(3 - 2\sigma_s + \frac{\mathcal{K}\delta_s + 2\varepsilon}{B_2}\big) + \frac{m^4}{\varepsilon^2}\mathbb{E}_{\widetilde{D}_s}\{\mathcal{L}(\hat{f}_{n_s})\}\big(\sum_{k=1}^{K}\frac{1}{p_s(k)}\big)\Big] + B_2\big(\varepsilon^2 + \frac{4B_2^2 m^2}{\varepsilon}\sqrt{\mathbb{E}_{\widetilde{D}_s}\{\mathcal{L}(\hat{f}_{n_s})\}}\big)^{\frac{1}{2}}$.

To obtain the conclusion shown in this theorem from above formulations, first we plug $\epsilon_1 \leq n_s^{-\frac{\nu + d + 1}{2(\alpha + d + 1)}}$ and $\epsilon_2 \leq n_s^{-\frac{\varsigma}{2(\alpha + d + 1)}}$ into it. apart from that, we have shown $\mathbb{E}_{\widetilde{D}_s}\{\mathcal{L}(\hat{f}_{n_s})\} \lesssim n_s^{-\frac{\alpha}{2(\alpha + d + 1)}}$ in A.3.7 and known $\delta_s^{(n_s)} \leq n_s^{-\frac{\tau + d + 1}{2(\alpha + d + 1)}}$, combining with the setting $\varepsilon_{n_s} = m^2 n_s^{-\frac{\min\{\alpha, \nu, \varsigma, \tau\}}{8(\alpha + d + 1)}}, \mathcal{K} = n_s^{\frac{d+1}{2(\alpha + d + 1)}}$ implies that $\mathcal{K}\epsilon_1/\varepsilon_{n_s}^2 \leq n_s^{-\frac{\tau}{2(\alpha + d + 1)}}, \epsilon_2/\varepsilon_{n_s}^2 \leq n_s^{-\frac{\tau}{2(\alpha + d + 1)}}, \mathcal{K}\delta_s^{(n_s)} \leq n_s^{-\frac{\tau}{2(\alpha + d + 1)}}$ and $\mathbb{E}_{\widetilde{D}_s}\{\mathcal{L}(\hat{f}_{n_s})\}/\varepsilon_{n_s}^2 \leq n_s^{-\frac{\alpha}{4(\alpha + d + 1)}}$.

Plugin these facts into the corresponding term of above formulations to get what we desired. $\qquad\square$

Let us first state the formal version of Theorem 3.9 and then prove it.

**Theorem A.21** (Formal version of Theorem 3.9). *If Assumptions 3.5, 3.3, 3.2, 3.7 and 3.8 all hold, set $W \geq cn_s^{\frac{2d + \alpha}{4(\alpha + d + 1)}}, L \geq 2\lceil \log_2(d + r)\rceil + 2, \mathcal{K} = n_s^{\frac{d+1}{2(\alpha + d + 1)}}$ and $\mathcal{A} = \mathcal{A}_{n_s}$ in Assumption 3.7, then, provided that $n_s$ is sufficiently large, with probability at least $\sigma_s^{(n_s)} - \mathcal{O}\big(n_s^{-\frac{\min\{\alpha, \nu, \varsigma, \tau\}}{16(\alpha + d + 1)}}\big) - \mathcal{O}\big(\frac{1}{\sqrt{\min_k n_t(k)}}\big)$, we have*

$$\mathbb{E}_{\widetilde{D}_s}\{\mathrm{Err}(Q_{\hat{f}_{n_s}})\} \leq (1 - \sigma_t^{(n_s)}) + \mathcal{O}(n_s^{-\frac{\min\{\alpha, \nu, \varsigma\}}{8(\alpha + d + 1)}}).$$

*Proof of Theorem 3.9.* Note that the main difference between Theorem A.20 and Theorem 3.9 is the condition $\psi(\sigma_s^{(n_s)}, \delta_s^{(n_s)}, \varepsilon_{n_s}, \hat{f}_{n_s}) > 0$, so we are going to focus on whether this condition holds under the condition of Theorem 3.9.

To show this, first recall $\psi(\sigma_t^{(n_s)}, \delta_t^{(n_s)}, \varepsilon_{n_s}, \hat{f}_{n_s}) = \Gamma_{\min}(\sigma_t^{(n_s)}, \delta_t^{(n_s)}, \varepsilon_{n_s}, \hat{f}_{n_s}) - \sqrt{2 - 2\Gamma_{\min}(\sigma_t^{(n_s)}, \delta_t^{(n_s)}, \varepsilon_{n_s}, \hat{f}_{n_s})} - \frac{\Delta_{\hat{\mu}_t}}{2} - \frac{2\max_{k \in [K]}\|\hat{\mu}_t(k) - \mu_t(k)\|_2}{B_2}$. Note (28) and dominated convergence theorem imply $R_t(\varepsilon_{n_s}, \hat{f}_{n_s}) \to 0$ a.s., thus

$$\Gamma_{\min}(\sigma_t^{(n_s)}, \delta_t^{(n_s)}, \varepsilon_{n_s}, \hat{f}_{n_s}) = \Big(\sigma_t^{(n_s)} - \frac{R_t(\varepsilon_{n_s}, \hat{f}_{n_s})}{\min_i p_t(i)}\Big)\Big(1 + \big(\frac{B_1}{B_2}\big)^2 - \frac{\mathcal{K}\delta_t^{(n_s)}}{B_2} - \frac{2\varepsilon_{n_s}}{B_2}\Big) - 1 \to \big(\frac{B_1}{B_2}\big)^2$$

Combining with the fact that $\frac{\Delta_{\hat{\mu}_t}}{2} = \frac{1 - \min_{k \in [K]}\|\hat{\mu}_t(k)\|^2/B_2^2}{2} < \frac{1}{2}$ can yield

$$\Gamma_{\min}(\sigma_t^{(n_s)}, \delta_t^{(n_s)}, \varepsilon_{n_s}, \hat{f}_{n_s}) - \sqrt{2 - 2\Gamma_{\min}(\sigma_t^{(n_s)}, \delta_t^{(n_s)}, \varepsilon_{n_s}, \hat{f}_{n_s})} - \frac{\Delta_{\hat{\mu}_t}}{2} > 1/2$$

if we select proper $B_1$ and $B_2$.

Besides that, by Multidimensional Chebyshev's inequality, we know that

$$\mathbb{P}_t\big(\|\hat{\mu}_t(k) - \mu_t(k)\|_2 \geq \frac{B_2}{8}\big) \leq \frac{64\sqrt{\mathbb{E}_{\boldsymbol{z} \in \widetilde{C}_t(k)}\mathbb{E}_{\boldsymbol{z}' \in \mathcal{A}(\boldsymbol{z})}\|f(\boldsymbol{z}') - \mu_t(k)\|_2^2}}{B_2^2\sqrt{2n_t(k)}} \leq \frac{128}{B_2\sqrt{n_t(k)}},$$

so that $\psi(\sigma_t^{(n_s)}, \delta_t^{(n_s)}, \varepsilon_{n_s}, \hat{f}_{n_s}) \geq \frac{1}{4}$ with probability at least $1 - \frac{128K}{B_2\sqrt{\min_k n_t(k)}}$ if $n_s$ is large enough, of course the condition $\psi(\sigma_t^{(n_s)}, \delta_t^{(n_s)}, \varepsilon_{n_s}, \hat{f}_{n_s}) > 0$ in Theorem A.20 can be satisfied.

Therefore, with probability at least

$$1 - p - \frac{128K}{B_2\sqrt{\min_k n_t(k)}} \gtrsim 1 - (1 - \sigma_s^{(n_s)}) - \mathcal{O}\big(n_s^{-\frac{\min\{\alpha,\nu,\varsigma,\tau\}}{16(\alpha+d+1)}}\big) - \mathcal{O}\big(\frac{1}{\sqrt{\min_k n_t(k)}}\big)$$

$$= \sigma_s^{(n_s)} - \mathcal{O}\big(n_s^{-\frac{\min\{\alpha,\nu,\varsigma,\tau\}}{16(\alpha+d+1)}}\big) - \mathcal{O}\big(\frac{1}{\sqrt{\min_k n_t(k)}}\big).$$

we have the conclusions shown in Theorem 3.9, which completes the proof. $\qquad\square$

