# OpenReview forum: "Unsupervised Transfer Learning via Adversarial Contrastive Training"
_ICML.cc/2025/Conference — Submitted to ICML 2025_

### Official Review · Reviewer_L3h2 · 2025-03-05

**Overall Recommendation:** 3

**Summary:**

This paper presents a novel unbiased self-supervised approach, Adversarial Contrastive Training (ACT), aimed at mitigating biased sample risk. The proposed ACT method is both simple and effective, utilizing matrix G to enhance self-supervised transformation learning. Notably, the k-NN evaluation demonstrates state-of-the-art performance on mini-sized images using ResNet18. Furthermore, the authors provide theoretical insights suggesting that the ACT loss function facilitates clustering in the representation space, thereby improving the learned feature distribution.

**Claims And Evidence:**

There appears to be a discrepancy between the claims and the presented results. In the analysis of the loss function, certain issues highlighted in the introduction seem to be addressed through assumptions and conditions rather than direct evidence. It is unclear which specific aspect the authors aim to guarantee. Strengthening the connection between the key claims and the supporting evidence would enhance the clarity of the proposal. Furthermore, the theoretical link between the proposed guarantee and the core component, matrix G, remains ambiguous, making it challenging to fully assess its impact.

**Essential References Not Discussed:**

The authors emphasize theoretical confidence in population risk, yet ACT exhibits lower performance than BYOL in linear evaluation. Providing a detailed analysis or justification for this performance gap would strengthen the discussion and help contextualize the effectiveness of the proposed method. Additionally, the update mechanism of matrix G bears similarities to an exponential moving average, which is known to enhance performance. However, its specific contribution to the success is not explicitly analyzed. Furthermore, the impact of mini-batch handling does not appear to be critical to the effectiveness of the model, as BYOL has been shown to function without batch statistics [Richemond et al., 2020]. Addressing this aspect could provide a clearer understanding of how ACT compares to existing approaches.

- BYOL works even without batch statistics [Richemond et al., 2020]

**Experimental Designs Or Analyses:**

The connection between the problem, proposed solution, and experimental design is not entirely clear, which makes it challenging to fully assess the validity of the approach. In particular, the link between the theoretical guarantees and the actual method remains ambiguous. Furthermore, the experiments appear insufficient to convincingly demonstrate the effectiveness of the proposed approach, and the experimental design does not fully support the claims made. To strengthen the paper, the authors could either clarify how the theoretical guarantees directly relate to the proposed method or provide additional experiments to further substantiate its validity.

**Methods And Evaluation Criteria:**

The use of matrix G is a simple yet effective approach. However, the lack of experiments exploring its contributions and limitations makes it difficult to fully assess its impact. It is initially unclear why the study focuses solely on ReLU networks, as similar boundary conditions might also apply to GeLU, particularly in transformer-based architectures. Additionally, while matrix G appears effective for small-sized inputs, it incurs higher computational costs for larger inputs, which is not thoroughly analyzed. Finally, ACT includes a hyperparameter, $\gamma$, but no experiments evaluate its impact across different values. Exploring this aspect would strengthen the empirical support for the method.

**Other Comments Or Suggestions:**

Even if a method is not entirely novel, a theoretical perspective alone can still provide a meaningful contribution. However, for such a contribution to be effective, it must be clearly communicated to the research community. While the authors appear to have adopted notation from previous works, this has made it difficult to discern which aspects they aim to emphasize mathematically. Due to this lack of clarity, it is challenging to fully appreciate the significance of their contribution. Improving the presentation and explanation of key theoretical insights would enhance the paper’s impact.

**Other Strengths And Weaknesses:**

### Strength:
ACT demonstrates notable performance improvements through a simple yet effective approach leveraging matrix G.
### Weakness:
The clarity of the presentation of the paper could be improved. Certain notations and theoretical explanations lack specificity, making it challenging to fully grasp key concepts and their implications.

**Questions For Authors:**

What is the precise relationship between the theoretical guarantees and matrix G in the proposed method?

**Relation To Broader Scientific Literature:**

The ACT method shares conceptual similarities with Barlow Twins (BT) in its approach. If the authors are confident in their method, they could consider aligning their experiments more closely with those in the BT paper to facilitate a more direct and meaningful comparison. Additionally, while the authors adopt notations from cited works, which helps maintain consistency with prior research, some notations remain unclear, making certain equations difficult to interpret. Providing explicit definitions and clarifications would significantly improve readability and accessibility for a broader audience.

**Theoretical Claims:**

The equations are challenging to interpret due to the lack of specificity in the notations. Providing clearer definitions and more consistent notation would enhance readability and make the theoretical analysis more accessible. The conclusions drawn from the equations do not appear to contradict previously established findings. However, due to the complexity of the equations and the lack of clarity in notation, I cannot confidently assess their correctness. A more detailed explanation would help in evaluating the theoretical claims more rigorously.

---

> ### Author Rebuttal · Authors · 2025-04-01
>
> We thank you for your thorough review of our manuscript and for your constructive suggestions. Our point-by-point responses to your comments are given below.
>
> > **C1** Improve the presentation.
>
> Thank you for your constructive suggestion. Please see the response to **C4** of Reviewer f4DL.
>
> > **C2** The role of the matrix $G$ in the proposed method and its theoretical analysis.
>
> Thank you for your valuable feedback. The matrix $G$ plays a pivotal role in both the theoretical analysis and the algorithmic design. Specifically:
> *  **Theoretical Perspective**: The matrix $G$ ensures that the sample-level regularization term in line 180 remains **unbiased**, whereas the sample-level regularization term in eq. (2) is biased. As discussed in Section 3.2, the biased sample-level loss introduces challenges for the error analysis. In contrast, the unbiased sample-level loss as eq. (8) simplifies the analysis, facilitating a more tractable approach to proving the theoretical guarantees.
> * **Practical Perspective**: Experimental results in Table 1 highlight that ACT significantly improves downstream classification accuracy compared to two biased self-supervised learning methods: Barlow Twins and Hao Chen et al. (2022). This demonstrates the **practical benefits** of incorporating matrix $G$, especially in terms of enhancing the performance of the model.
>
> > **C3** The computational costs of updating $G$.
>
> As outlined in step 8 of Algorithm 1, the update of matrix $G$ involves only two encoder evaluations and a summation over the samples in a mini-batch. The computational cost of this step is relatively **minimal** compared to the cost of training the encoder, which not only requires encoder evaluations but also involves backpropagation and gradient descent updates.
>
> > **C4** The activation function and architectures of the networks.
>
> To maintain a fair comparison, we use the ReLU activation function and ResNet18 as the backbone, as done in BT and Hao Chen (2022). However, we appreciate your suggestion and will consider exploring the use of GeLU and transformer-based architectures, in future work involving more complex tasks.
>
> > **C5** Hyperparameter $\gamma$ of ACT.
>
> We would like to clarify that **ACT as eq. (4) does not include a hyperparameter $\gamma$**. We presume you mean $\lambda$. If it is, we conduct ablation studies across a range of values for the regularization parameter $\lambda$. The experimental results, as shown in https://anonymous.4open.science/r/RE1-FFCE, demonstrate that ACT is **robust** to the selection of $\lambda$
>
> > **C6** The experiments appear insufficient.
>
> We appreciate your feedback.
> * **Comparison with BT**: Experimental results in Table 1 demonstrate that ACT significantly outperforms Barlow Twins in downstream classification accuracy. Additionally, the results in Table 2 show that ACT achieves SOTA performance when compared to mainstream baselines.
> * **Transfer Learning**: We include experiments for transfer learning from CIFAR100 to CIFAR10, which are presented in https://anonymous.4open.science/r/RE4-8B12.
> * **Ablation Studies**: We conduct ablation studies across a variety of augmentation methods and a range of regularization parameter $\lambda$. The results are provided as https://anonymous.4open.science/r/RE1-FFCE and https://anonymous.4open.science/r/RE5-98C8.
>
> > **C7** ACT exhibits lower performance than BYOL in linear evaluation.
>
> Thank you for your insightful comment. (1) While ACT demonstrates lower performance than BYOL on the Tiny ImageNet dataset in linear evaluation, it still outperforms other methods. Additionally, as shown in Table 2, **ACT consistently outperforms BYOL on the CIFAR-10 and CIFAR-100**, which highlights its effectiveness. (2) The observed performance gap on Tiny ImageNet could be influenced by the unique **characteristics of the dataset**. Our theoretical analysis indicates that ACT's performance is dependent on properties of dataset, and this variability may explain the differing performance on Tiny ImageNet.
>
> > **C8** ACT bears similarities to an exponential moving average.
>
> Thank you for your comment. However, to the best of our knowledge, **the update of $G$ has limited relevance to EMA** used in BYOL. Specifically, EMA in BYOL involves online and target neural networks. In contrast, ACT employs a single network.
>
> > **C9** BYOL works even without batch statistics.
>
> Thank you for your comment. We would like to clarify that the paper you referenced, *BYOL works even without batch statistics*, has limited relevance to ACT. In BYOL, batch normalization (BN) plays a critical role in preventing collapse. The paper you mentioned shows that using group normalization can achieve competitive performance compared to BYOL with BN. However, rather than relying on BN, ACT prevents collapse through an explicit regularization term. This distinction means that **the impact of the paper you referenced in ACT is not as critical as it is in BYOL**.

---

> > ### Comment · Reviewer_L3h2 · 2025-04-04
> >
> > Thank you. I’ve raised my score.

---

> > > ### Author Response · Authors · 2025-04-04
> > >
> > > Thank you for your thorough review of our manuscript and for raising your score following our rebuttal. We greatly appreciate the constructive suggestions you provided, which have significantly improved the quality and clarity of our work. Your insights have been invaluable in helping us strengthen our paper, and we have carefully incorporated your feedback into the revised version.

---

### Official Review · Reviewer_a51U · 2025-03-05

**Overall Recommendation:** 4

**Summary:**

This work studies the theoretical aspects of contrastive learning. Specifically, the authors focus on the regularization framework for model collapse in contrastive learning, where the main issue in current works is the population-level bias and sample-level bias cannot be simultaneously mitigated. To deal with this, the authors reformulate the risk function, which admits unbias property w.r.t. expectation; based on this reformulation, rigorous theoretical analysis is provided, e.g., upper-bound of error, consistency/convergence.

---

**Rebuttal Update**:
I would like to thank the authors for providing detailed responses, which generally addressed my concerns. I think this work provides an effective estimator with intuitive reasoning and theoretical guarantees for popular self-supervised transfer learning objective. Thus, I keep my original positive recommendation.

**Claims And Evidence:**

The claims are generally consistent with theoretical results, while the empirical evidence from numerical experiments, i.e., performance improvement over SOTA baselines, needs further improvement and justification (details are provided in *Experimental Designs Or Analyses*).

**Essential References Not Discussed:**

The references are appropriate.

**Experimental Designs Or Analyses:**

The experiment and analyses are generally valid, while there are some minor concerns:

C1. In lines 255-257, the representation dimensions of baseline methods are 512, while the proposed method employs different dimensions in different datasets. I know that the dimensions of the proposed method are much smaller, while such differences in network architectures could have impacts on the fairness of comparison. Some justifications are highly appreciated.

C2. Following Q1, are there results of implementing the proposed ACT with a dimension of 512? These additional results could provide a more comprehensive understanding of the effectiveness of ACT.

C3. The different dimension settings naturally raise a question, i.e., how to choose a proper dimension for ACT? Will the ACT model be sensitive to the choices? And, how to choose the dimension empirically?

**Methods And Evaluation Criteria:**

The methodology and evaluation criteria are appropriate.

**Other Comments Or Suggestions:**

1. The $\kappa (\theta)$ in line 138 is used before it is undefined.

2. Line 189-191, repeated sentence ‘the collection of used data augmentations’.

3. Line 199, repeated notation $A_{i,1}$.

4. What is ${\hat{f}}_{n_s}$ (line 255)? Specifically, the meanings of subscripts of $f$ should be clarified.

**Other Strengths And Weaknesses:**

**Pros:**

1. The key idea, i.e., correcting the sample-level bias and population-level bias, is novel.

2. The proposed method with the rewriting risk function is reasonable.

3. The theoretical results rigorously support the proposed method.

**Cons:**

1. The organization should be improved, where the technical parts are quite dense and the notations are complex.

**Questions For Authors:**

Q1. The notation $\mathbb{P}_s(k)(\cdot)$ is confusing. Specifically, what is the domain of this function? (probably $\mathcal{X}$?)

Q2. How to obtain the last term of inequality in line 305? i.e., how to replace the $\hat{f}$ with $\bar{f}$ on the empirical loss $\hat{\mathcal{L}}$?

Q3. An interesting point claimed in this work is that the bias is induced by non-commutativity between the sum operator (expectation) and the Frobenius norm. Therefore, the modified risk in Eq. (3) ensures that the risk is linear w.r.t. the covariance difference term (also $G$), which raises two questions:

(3.1) Assume that the true risk (on all possible data) can be split into multiple batch-wise estimated risks, then true $G^*$ should be the same on each batch due to the linearity of $G$ w.r.t. $\mathcal{R}(f,G)$. Thus, the on-the-fly estimation provided in step 4/8 of Alg. 1 seems to be inconsistent with this conclusion. If this conclusion is incorrect, some explanations are highly appreciated.

(3.2) Another intuitive idea for learning $G$ is parameterizing it, e.g., a single-layer network (single weight matrix). The essential intuition for such a design is the linearity, which seems to suggest that $G^*$ (and also the approximation $G$) could or even should be unchanged in each batch.

**Relation To Broader Scientific Literature:**

The key contribution, i.e., unbiased estimations for empirical regularizer and model risk, seems to be novel.

**Theoretical Claims:**

The theoretical results and proofs look correct.

---

> ### Author Rebuttal · Authors · 2025-04-01
>
> We thank you for your thorough review of our manuscript and for your constructive suggestions. Our point-by-point responses to your comments are given below. Thank you for pointing out these typos. We correct them in the revised manuscript.
>
> > **C1** The dimension of ACT.
>
> * We appreciate the suggestion and agree that evaluating ACT with a 512-dimensional representation would ensure fairness in comparison. We implement the proposed ACT method using **the same 512-dimensional representation as the baseline methods**, which outperforms other methods. Further, we conducted **ablation studies** across a range of dimensions of ACT. Our experiments suggest that ACT is **robust** to dimension choices within a reasonable range. The experimental results are shown as https://anonymous.4open.science/r/RE3-1437.
> * Choosing the appropriate dimension for representation learning may vary depending on the dataset, task complexity, and computational and memory constraints. We recommend performing cross-validation (CV) over a range of dimensions. Additionally, dimension selection may be influenced by resource constraints, as lower-dimensional representations can offer faster inference and reduced memory usage.
>
> > **C2** Improve the presentation of the technical parts.
>
> Thank you for your constructive suggestion. Please see the response to **C4** of Reviewer f4DL.
>
> > **C3** Notations $\mathbb{P}\_{s}(\cdot)$ and $\hat{f}\_{n_{s}}$.
>
> * $\mathbb{P}\_{s}(\cdot)$ denotes the probability distribution of the source data, while $\mathbb{P}\_{s}(k)(\cdot)$ denotes **the probability distribution of the source data that categorized into the $k$-th latent class $C\_{s}(k)$**. Both of them are defined on the domain $\mathcal{X}$.
> * $\hat{f}\_{n_{s}}$ denotes the ACT estimator defined as eq. (4). The subscript $n_{s}$ is the number of unlabeled training samples in the source domain. We use this subscript to emphasize that **this estimator depends on the training dataset**, and we show that the error of this estimator converges as $n_{s}$ increases.
>
> > **C4** The derivation of line 305.
>
> Thank you for your suggestion. The last term of inequality in line 305 used the fact that $\hat{f}$ is the minimizer of the empirical risk over the hypothesis class. For the sake of clarity of the presentation, we provide the complete derivation:
>
> We define the minimizer of the biased sample-level loss as
> $$
> \hat{f}\_{n_{s}}^{\mathrm{bias}}\in\mathop{\arg\min}\_{f\in\mathcal{F}}\widehat{\mathcal{L}}(f):=\widehat{\mathcal{L}}\_{\mathrm{align}}(f)+\lambda\widehat{\mathcal{R}}(f),
> $$
> where $\widehat{\mathcal{R}}(\cdot)$ is defined as (2). We then consider the **expected population risk** of this estimator. For each $\bar{f}\in\mathcal{F}$, it holds that
> \begin{align*}
> \mathcal{L}(\hat{f}\_{n_{s}}^{\mathrm{bias}})
> &=\\{\mathcal{L}(\hat{f}\_{n_{s}}^{\mathrm{bias}})-\widehat{\mathcal{L}}(\hat{f}\_{n_{s}}^{\mathrm{bias}})\\}+\\{\widehat{\mathcal{L}}(\hat{f}\_{n_{s}}^{\mathrm{bias}}) -\mathcal{L}(\bar{f})\\}+\\{\mathcal{L}(\bar{f})-\mathcal{L}(f^{\*})\\}+\mathcal{L}(f^{\*}) \\\\
> &\leq\\{\mathcal{L}(\hat{f}\_{n_{s}}^{\mathrm{bias}})-\widehat{\mathcal{L}}(\hat{f}\_{n_{s}}^{\mathrm{bias}})\\}+\\{\widehat{\mathcal{L}}(\bar{f})-\mathcal{L}(\bar{f})\\}+\\{\mathcal{L}(\bar{f})-\mathcal{L}(f^{\*})\\}+\mathcal{L}(f^{\*}) \\\\
> &\leq 2\sup\_{f\in\mathcal{F}}|\mathcal{L}(f)-\widehat{\mathcal{L}}(f)|+\\{\mathcal{L}(\bar{f})-\mathcal{L}(f^{\*})\\}+\mathcal{L}(f^{\*}),
> \end{align*}
> where the first inequality follows from the fact that **$\hat{f}\_{n_{s}}^{\mathrm{bias}}$ minimizes the empirical risk $\widehat{\mathcal{L}}(\cdot)$ over the hypothesis class $\mathcal{F}$**, and thus $\widehat{\mathcal{L}}(\hat{f}\_{n_{s}}^{\mathrm{bias}})\leq\widehat{\mathcal{L}}(\bar{f})$.
>
> Standard techniques in empirical process theory can be applied to estimate the first term in **unbiased** situations. However, the **biased** nature of $\widehat{\mathcal{L}}(\cdot)$ complicates the process of bounding the first term.
>
> > **C5** (1) The on-the-fly estimation is not the same on each batch. (2) Parameterize $G$ using a network.
>
> Thank you for your constructive suggestion.
> * We agree that the true $G^{\*}$ should be consistent across batches, as it is determined by the population risk rather than by the batch-specific empirical risk. However, as indicated by our theoretical analysis, **the on-the-fly estimation of $G^{\*}$ differ only slightly between batches, provided that the batch size is sufficiently large**.
> * You raise an interesting point about parameterizing $G$ using a network. Since the solution to the inner maximization problem in Eq. (4) has a **closed-form** solution, as shown in step 4/8 of Alg. 1, we prefer to rely on the closed-form solution. However, we appreciate your suggestion and will consider **exploring this idea in more complex tasks in future work**.

---

### Official Review · Reviewer_etBC · 2025-03-08

**Overall Recommendation:** 3

**Summary:**

This paper focuses on the task of transfer learning and proposes a loss function based on adversarial contrastive learning. More concretely, based on this adversarial conservative learning framework, this paper learns a representation map from source data which can be transferred to the target distribution for downstream classification tasks.

**Claims And Evidence:**

Yes.

**Essential References Not Discussed:**

N/A

**Experimental Designs Or Analyses:**

Yes.

**Methods And Evaluation Criteria:**

Yes.

**Other Comments Or Suggestions:**

N/A

**Other Strengths And Weaknesses:**

Strength
1. This paper focuses on an important problem in learning theory: the benefit of contrastive representation learning for transfer learning.
2. The introduction of adversarial contrastive training with the notation of debiasing the sample-level spectral loss is interesting.
3. Conditions for theoretical results are clearly stated and well-organized.

Weakness
1. The presentation of theoretical results is not clear and concise enough. For example, the derivation of objective functions on page 6 can be deferred to the appendix.
2. The presentation of Section 2 is a little confusing to me. As this section is motivated by the bias in the sample-level spectral contrastive loss, I wonder if the proposed objective function $\hat \mathcal{L}(f,G)$ has some kind of unbiasedness at the sample level. If so, it would be beneficial to state it clearly; otherwise, the comparison is not fair enough.
3. The technical assumptions are presented in a dense way and there are not enough intuitive interpretations for some of the assumptions. For example, Assumption 3.8 is common in the theory of transfer learning, but it would be much better if concrete examples (e.g. Gaussian family with linear functions) can be presented for more straightforward understanding. It also applies to Assumption 3.7.

**Questions For Authors:**

Please also see the previous "Weakness" section.

1. I wonder what is the suggested choice of $\lambda$ implied by theory.
2. In Assumption 3.7, I wonder why is $\sigma_s$ related to augmentations.
3. In Theorem 3.9, it would be beneficial to interpret the role of $\sigma_s$ in the performance guarantee, and how it is related to Assumption 3.7.
4. If we use supervised transfer learning as a benchmark (with $n_s$ observations from the source distribution), how is the comparison with the rate in that scenario with the Holder class? What is the fundamental gap between (contrastive) unsupervised and supervised transfer learning and are there any scenarios such that the gap can be closed?

I would be happy to raise my score if the aforementioned questions are addressed.

**Relation To Broader Scientific Literature:**

As distribution shifts are common with real data, this transfer learning technique can be useful for scientific research.

**Theoretical Claims:**

Yes.

---

> ### Author Rebuttal · Authors · 2025-04-01
>
> We thank you for your thorough review of our manuscript and for your constructive suggestions.
>
> > **C1** Examples of Assumption 3.8.
>
> We exemplify the source/target distributions as **Gaussian mixtures** of the same componential variances. Then $\epsilon_{1}$ is the maximum distance between the means of the source and target distributions for each latent class. $\epsilon_{2}$ is the maximum distance between the mixture weights of the source and target distributions. Thus, Assumption 3.8 not only requires that the source and target distributions for each latent class are close in terms of their means, but also that their mixture weights are similar.
>
> > **C2** Explanations of Assumption 3.7.
>
> The concept of $(\sigma_{s},\delta_{s})$-augmentation is introduced to quantify the concentration of augmented data. We now provide a step-by-step explanation:
> * Augmentation distance: for a given augmentation set $\mathcal{A}$, the augmentation distance between two samples $x_{1}$ and $x_{2}$ are defined as: $\\|x_{1}-x_{2}\\|\_{\mathcal{A}}:=\min\_{x_{1}^{\prime}\in\mathcal{A}(x_{1}),x_{2}^{\prime}\in\mathcal{A}(x_{2})}\|x_{1}^{\prime}-x_{2}^{\prime}\|$. Since augmentations can capture semantic meanings of the original sample through various views, this distance reflects the maximal semantic similarity between the two samples.
> * $\sigma_{s}$-main-part of the latent class: for a latent class $C_{s}(k)$, the $\sigma_{s}$-main-part is defined as $\widetilde{C}\_{s}(k)\subseteq C\_{s}(k)$ satisfying $\mathbb{P}\_{s}\\{x\in\widetilde{C}\_{s}(k)\\}\geq\sigma_{s}\mathbb{P}\_{s}\\{x\in C_{s}(k)\\}$. The parameter $\sigma_{s}$ quantifies the concentration of the distribution $\mathbb{P}\_{s}(k)$ of this latent class. Specifically, for fixed $\widetilde{C}\_{s}(k)$ and $C\_{s}(k)$, a larger value of $\sigma_{s}$ indicates a higher concentration of $\mathbb{P}\_{s}(k)$.
> * Augmentation diameter of the $\sigma_{s}$-main-part: the parameter $\delta_{s}$ is defined as the diameter of the $\sigma_{s}$-main-part in augmentation distance, that is, $\sup\_{x_{1},x_{2}\in\widetilde{C}\_{s}(k)}\\|x_{1}-x_{2}\\|\_{\mathcal{A}}$. For a fixed distribution $\mathbb{P}\_{s}$ and a fixed parameter $\sigma_{s}$, the smaller value of the diameter $\delta_{s}$ means a higher concentration of the augmented distribution, as well as greater similarity between augmented data samples.
> * **Summary**: The concentration of the augmented distribution, as measured by parameters $(\sigma_{s},\delta_{s})$, depends on both $\mathbb{P}\_{s}(k)$ and $\mathcal{A}$. Specifically, for a fixed $\mathcal{A}$, a smaller value of $\sigma_{s}$ and a higher concentration of $\mathbb{P}\_{s}(k)$ result in a smaller $\widetilde{C}\_{s}(k)$, leading to a smaller value of $\delta_{s}$. Additionally, for a fixed distribution $\mathbb{P}\_{s}(k)$, a smaller value of $\sigma_{s}$ and a larger $\mathcal{A}$ lead to smaller $\\|x_{1}-x_{2}\\|\_{\mathcal{A}}$ for each pair $(x_{1},x_{2})$, resulting in a smaller value of $\delta_{s}$.
>
> **Example**: Suppose the samples in the $k$-th latent class follows the uniform distribution on $[0,R]$, i.e., $C_{s}(k)=[0,R]$ and $\mathbb{P}\_{s}(k)=\mathsf{unif}(0,R)$. For each $\sigma_{s}\in(0,1]$, we can find a $\sigma_{s}$-main-part of $C\_{s}(k)$ as $\widetilde{C}\_{s}(k)=[0,\sigma_{s}R]$. Further, we define $\mathcal{A}(x)=\{x^{\prime}\in\mathbb{R}:|x-x^{\prime}|\leq r\}$ for each $x\in\mathcal{X}$. Then the augmentation diameter $\delta_{s}$ of the $\sigma_{s}$-main-part is given as
> $$\sup\_{x_{1},x_{2}\in\widetilde{C}\_{s}(k)}\\|x_{1}^{\prime}-x_{2}^{\prime}\\|\_{\mathcal{A}}=\max\\{\sigma_{s}R-2r,0\\}=:\delta_{s}.$$
> **The parameters $\sigma_{s}$, $\delta_{s}$, $r$ and $R$ are interrelated by this equality**. Note that the parameter $R$ reflects the concentration of the distribution $\mathbb{P}\_{s}(k)$ within the latent class. A smaller value of $R$ indicates a higher concentration of $\mathbb{P}\_{s}(k)$, which in turn leads to a smaller value of the augmentation diameter $\delta_{s}$. Additionally, a larger augmentation set, i.e., a larger value of $r$, results in a smaller value of the augmentation diameter $\delta_{s}$.
>
> > **C3** The role of $\sigma_{s}$ in the performance guarantee.
>
> Thanks for your suggestion, we have added the interpretation for $\sigma_s$ in revised manuscript.
> * The probability of the inequality Line 343 is directly determined by $\sigma_s$. The closer  $\sigma_s$ is to 1, the larger the probability of this event.
> * The technical effect of $\sigma_s$ in Assumption 3.7. is to convert the condition $\psi > 0$ (Line 1646) into a probabilistic form, ensuring that this condition can be definitively satisfied.
>
> **Due to space constraints**, we are unable to address the questions regarding unbiasedness and supervised transfer learning in this rebuttal. We kindly ask that you evaluate our existing responses for now, and **we will be glad to provide the remaining explanations once we receive your feedback.**

---

> > ### Comment · Reviewer_etBC · 2025-04-06
> >
> > Thanks so much for your clarification. I’ve raised my score.

---

> > > ### Author Response · Authors · 2025-04-07
> > >
> > > We appreciate your thoughtful review and the raised score. We are committed to incorporating the valuable feedback received during this process to further strengthen our work. Our additional responses and explanations are provided below.
> > >
> > > > **C4** The derivation of objective functions.
> > >
> > > Thank you for your valuable suggestion. The derivation of the objective functions Section 3.2 is detailed in the appendix to streamline the main text.
> > >
> > > > **C5** The unbiasedness of $\widehat{\mathcal{L}}(f,G)$.
> > >
> > > Thank you for your thoughtful comment. The proposed objective function $\widehat{\mathcal{L}}(f,G)$ is an unbiased estimate to the population risk $\mathcal{L}(f,G)$ as pointed out in Line 185. We now provide a detailed derivation. It is sufficient to consider the regularization term, since the alignment term $\widehat{\mathcal{L}}\_{\mathrm{align}}(\cdot)$ is obviously unbiased. Specifically, for each fixed $f$ and $G$, one has
> > > \begin{align*}
> > > \mathbb{E}\_{\widetilde{D}\_{s}}[\widehat{\mathcal{R}}(f,G)]
> > > &=\mathbb{E}\_{\widetilde{D}\_{s}}\Big[\Big\langle\frac{1}{n_{s}}\sum\_{i=1}^{n_{s}}f(x_{1}^{(i)})f(x_{2}^{(i)})^{\top}-I_{d^{\*}},G\Big\rangle\_{F}\Big] \\\\
> > > &=\Big\langle\mathbb{E}\_{\widetilde{D}\_{s}}\Big[\frac{1}{n_{s}}\sum\_{i=1}^{n_{s}}f(x_{1}^{(i)})f(x_{2}^{(i)})^{\top}\Big]-I_{d^{\*}},G\Big\rangle\_{F}=\mathcal{R}(f,G),
> > > \end{align*}
> > > where the second equality invokes the linearity of the inner product. This equality implies the unbiasedness of the proposed regularization function $\widehat{\mathcal{R}}(f,G)$. Combining this with the unbiasedness of the sample-level alignment term $\widehat{\mathcal{L}}\_{\mathrm{align}}(\cdot)$ yields the unbiasedness of the proposed objective function $\widehat{\mathcal{L}}\_{\mathrm{align}}(\cdot)$. We revise Section 2 to more clearly outline that the proposed sample-level loss in eq. (4) is unbiased.
> > >
> > > > **C6** The choice of $\lambda$ implied by theory.
> > >
> > > Thank you for your insightful feedback.
> > >
> > > The regularization parameter $\lambda$ in ACT balances the alignment term $\mathcal{L}_{\mathrm{align}}(\cdot)$ and the regularization term $\mathcal{R}(\cdot)$.
> > >
> > > From a **theoretical perspective**, our theoretical analysis suggests that $\lambda=\mathcal{O}(1)$. Specifically,
> > > * In Lemma A.4, we demonstrate that the alignment factor $R\_{t}(\varepsilon,f)$ can be bounded by the alignment term $\mathcal{L}\_{\mathrm{align}}(\cdot)$, while the divergence factor $\max_{i\neq j}|\mu_{t}(t)^{\top}\mu_{t}(j)|$ is bounded by the regularization term $\mathcal{R}(\cdot)$.
> > > * Based on the definition of the population risk $\mathcal{L}(f)=\mathcal{L}\_{\mathrm{align}}(f)+\lambda\mathcal{R}(f)$, we find
> > > \begin{equation*}
> > > \mathcal{L}\_{\mathrm{align}}(f)\leq\mathcal{L}(f) \quad\text{and}\quad \mathcal{R}(f)\leq\lambda^{-1}\mathcal{L}(f)\lesssim\mathcal{L}(f),
> > > \end{equation*}
> > > where we used $\lambda=\mathcal{O}(1)$. This allows us to bound both the alignment factor $R_{t}(\varepsilon,f)$ and the divergence factor $\max_{i\neq j}|\mu_{t}(t)^{\top}\mu_{t}(j)|$ in terms of the population risk $\mathcal{L}(f)$, which leads to the conclusion in Theorem A.5.
> > >
> > > From a **practical perspective**, we include ablation studies across a range of regularization parameter. The experimental results are shown as https://anonymous.4open.science/r/RE1-FFCE. The experimental results show that ACT is robust to the selection of $\lambda$. For more complex tasks, we recommend performing cross-validation (CV) over a range of regularization parameter.
> > >
> > > > **C7** The comparison with supervised transfer learning.
> > >
> > > * **Supervised transfer learning**: the convergence rate of the nonparametric transfer learning derived in [1] is given as $\mathcal{O}(\max\\{n_{s},n_{t}\\}^{-\frac{2\alpha}{2\alpha+d}}+(\epsilon\vee n_{t}^{-\frac{\alpha}{2\alpha+d}})n_{t}^{-\frac{\alpha}{2\alpha+d}})$. The rate given in [2] is $\mathcal{O}(n_{s}^{-\frac{\alpha}{2d+3\alpha}}+n_{t}^{-\frac{\alpha}{2(d^{*}+1+2\alpha)}})$ for $\alpha>2$.
> > > * **Similarities**: Both supervised transfer learning and unsupervised transfer learning exhibit convergence as $n_{s}$ and $n_{t}$ increase.
> > > * **Differences**: While labels from the source domain are available in supervised transfer learning, unsupervised transfer learning relies on pseudo-labels generated through augmentation. As a result, unsupervised transfer learning depends on the specific parameters of the augmentation methods.
> > > * **Whether the gap can be closed?** If unsupervised learning techniques can be improved via more informative augmentations, the performance gap can potentially be narrowed. We will continue to explore ways to bridge this gap in future work.
> > >
> > > [1] T. Tony Cai and Hongming Pu. Transfer Learning for Nonparametric Regression: Non-asymptotic Minimax Analysis and Adaptive Procedure. (2024)
> > >
> > > [2] Yuling Jiao, Huazhen Lin, Yuchen Luo, and Jerry Zhijian Yang. Deep Transfer Learning: Model Framework and Error Analysis. (2024)

---

### Official Review · Reviewer_f4DL · 2025-03-14

**Overall Recommendation:** 4

**Summary:**

This paper introduces Adversarial Contrastive Training (ACT), a novel approach to unsupervised transfer learning that addresses bias issues in existing contrastive learning methods. The authors provide both theoretical guarantees and empirical evidence demonstrating the effectiveness of their approach. The theoretical guarantees connecting upstream unlabeled data to downstream performance are particularly valuable, offering insights into why these methods work well in few-shot learning scenarios.

**Claims And Evidence:**

Yes.

**Essential References Not Discussed:**

- Section 4 could be strengthened by including references of existing methods for clarity.

**Experimental Designs Or Analyses:**

- The experiments demonstrate consistent improvements over baseline methods across multiple datasets (CIFAR-10, CIFAR-100, and Tiny ImageNet), validating the practical relevance of addressing the bias issue.
- The paper mentions that $\lambda$ is an important hyperparameter but does not provide a comprehensive ablation study showing how different val- ues affect performance across datasets. This analysis would provide valuable insights into the robustness of the method.

**Methods And Evaluation Criteria:**

- The paper identifies a critical bias issue in existing contrastive learning methods and presents a clever solution through adversarial train- ing. The min-max formulation effectively tackles the bias between population-level and sample-level estimators.
- The paper presents Algorithm 1 in a general format but lacks a practical implementation. Including this would make the method more accessible to practitioners and facilitate reproduction.
- How sensitive is ACT to the choice of augmentation strategy? Is there a principled way to select this hyper-parameter?

**Other Comments Or Suggestions:**

No.

**Other Strengths And Weaknesses:**

- The paper lacks a dedicated notation list or table that would help readers track the numerous mathematical symbols used throughout the theoretical sections. This would significantly improve readability, especially for the complex mathematical derivations.

**Questions For Authors:**

- In the related works, the authors mention that the Rademacher complexity can be significantly reduced by controlling the scale of the network class, which causes the upper bound to be ineffective if the approximation error is ignored. Could the authors please elaborate on this claim?

**Relation To Broader Scientific Literature:**

The key contribution of this paper is identifying bias issues in existing contrastive learning methods, which have not been thoroughly discussed in prior work. Additionally, the proposed solution, ACT, is supported by both theoretical guarantees and empirical evidence, demonstrating its effectiveness.

**Theoretical Claims:**

- The authors develop a comprehensive end-to-end theoretical analysis for their method, showing how ACT can lead to downstream data being clustered in representation space. This theoretical work bridges an important gap in the literature.
- The paper provides valuable theoretical insights for few-shot learning, explaining why ACT can achieve good performance even with limited downstream samples.

---

> ### Author Rebuttal · Authors · 2025-04-01
>
> We thank you for your thorough review of our manuscript and for your constructive suggestions. Our point-by-point responses to your comments are given below.
>
> > **C1** Lack of practical implementation.
>
> We add a detailed PyTorch-type pseudo-code in the appendix of the revised version.
>
> > **C2** The choice of the augmentation strategy.
>
> Thank you for your comment.
> * We include **ablation studies** across a variety of data augmentation methods. The experimental results are shown as https://anonymous.4open.science/r/RE5-98C8. The experimental results show that ACT is generally **robust** to the choice of augmentation strategy.
> * Selecting the appropriate augmentation strategy for ACT may vary depending on the dataset and the complexity of the task. To empirically choose the best approach, we recommend using cross-validation (CV). However, this method can be computationally expensive. Fortunately, as our ablation studies demonstrate, ACT shows a relatively low sensitivity to the choice of augmentation method.
>
> > **C3** The choice of the regularization parameter.
>
> The regularization parameter $\lambda$ in ACT balances the alignment term $\mathcal{L}_{\mathrm{align}}(\cdot)$ and the regularization term $\mathcal{R}(\cdot)$.
>
> **From a theoretical perspective**, our theoretical analysis suggests that $\lambda=\mathcal{O}(1)$. Specifically,
> * In Lemma A.4, we demonstrate that the alignment factor $R\_{t}(\varepsilon,f)$ can be bounded by the alignment term $\mathcal{L}\_{\mathrm{align}}(\cdot)$, while the divergence factor $\max_{i\neq j}|\mu_{t}(t)^{\top}\mu_{t}(j)|$ is bounded by the regularization term $\mathcal{R}(\cdot)$.
> * Based on the definition of the population risk $\mathcal{L}(f)=\mathcal{L}\_{\mathrm{align}}(f)+\lambda\mathcal{R}(f)$, we find
> \begin{equation*}
> \mathcal{L}\_{\mathrm{align}}(f)\leq\mathcal{L}(f) \quad\text{and}\quad \mathcal{R}(f)\leq\lambda^{-1}\mathcal{L}(f)\lesssim\mathcal{L}(f),
> \end{equation*}
> where we used $\lambda=\mathcal{O}(1)$. This allows us to bound both the alignment factor $R_{t}(\varepsilon,f)$ and the divergence factor $\max_{i\neq j}|\mu_{t}(t)^{\top}\mu_{t}(j)|$ in terms of the population risk $\mathcal{L}(f)$, which leads to the conclusion in Theorem A.5.
>
> **From a practical perspective**, we include **ablation studies** across a range of regularization parameter $\lambda$. The experimental results are shown as https://anonymous.4open.science/r/RE1-FFCE. The experimental results show that ACT is **robust** to the selection of $\lambda$. For more complex tasks, we recommend performing cross-validation (CV) over a range of regularization parameter.
>
> > **C4** Presentation suggestions.
>
> Thank you for your constructive suggestion. To improve the clarity and readability of the technical sections, we add a comprehensive **table of notations** in the appendix. Additionally, we include a **proof sketch** in the appendix to offer an overview of the key steps in the proofs. We also provide more **detailed explanations** and broken down the steps more explicitly in the proofs of theoretical results to ensure better understanding.
>
> > **C5** The necessity of the approximation error analysis.
>
> Thank you for your insightful comment. As shown in eq. (8) of the manuscript, the excess risk can be decomposed into the **approximation error** and the **statistical error**, where the latter is bounded by the **Rademacher complexity**. The approximation error reflects the capacity of the deep neural network class to approximate the target function, and it generally decreases as the scale of the network class increases. On the other hand, the Rademacher complexity increases with the scale of the network class. This creates a **trade-off** between the approximation error and the statistical error, suggesting that the network class should be chosen with an appropriate scale that depends on both the number of samples and the complexity of the task. This theoretical insight is consistent with the findings in experimental practice.
>
> However, if the approximation error is ignored, the excess risk is only bounded by the Rademacher complexity, which would imply that the network class should be as small as possible. Such an theoretical result clearly contradicts practical applications, where smaller and simpler network classes often struggle to capture the underlying patterns in complex tasks, particularly when dealing with large datasets.

---

### Official Review · Reviewer_LFM2 · 2025-03-14

**Overall Recommendation:** 4

**Summary:**

The paper presents a novel self-supervised learning (SSL) transfer learning technique that is unbiased and with provable guarantees.
The method is part of the class of SSL decorrelation approaches that align the cross-correlation matrix of learned representations with the identity matrix.

The paper first highlights that current approaches rely on biased sample-level covariance/correlation matrix estimators.
Then, an unbiased adversarial learning objective is derived, which is formulated as a min-max problem.

Subsequently, the authors:
- highlight the limitations of biased estimators from a theoretical perspective
- and, based on a series of assumptions, derive bounds on (1) the angle between the classes in the target domain and (2) the misclassification error in the source domain (Theorem 3.9).

This theorem provides theoretical insights for few-shot learning and demonstrates that abundant unlabeled data benefits transfer learning (i.e., to a different target domain).

Empirically, the method improves (linear and kNN) classification accuracy compared to previous methods on three different datasets.

## Update after rebuttal

Since the authors addressed all my concerns, I raised my rating from "weak accept" to "accept".

**Claims And Evidence:**

### "We introduce Adversarial Contrastive Training (ACT), a novel self-supervised transfer learning method. This approach learns representations from unlabeled data by solving a min-max optimization problem that corrects the bias inherent in existing methods."

- The introduced method is **novel** and well explained.
- The paper **explains why current estimators are biased** and **derives and unbiased estimator**.

### "Through extensive experiments, we demonstrate that ACT significantly outperforms traditional biased iterative methods (Table 1). Our empirical evaluation shows that ACT achieves state-of-the-art classification performance across multiple benchmark datasets using both fine-tuned linear probes and k-nearest neighbor (k-nn) protocols (Table 2)."

- The method indeed **outperforms the biased approaches** (Table 1) in both linear probe and kNN evaluation settings.
- It **also outperforms other SSL methods** (Table 2) in this same setting. However, it **cannot be determined if the method is state-of-the-art** (SOTA) as it is not compared with SOTA methods, such as e.g. SwAV [1] or DINO [2].

### "We establish comprehensive end-to-end theoretical guarantees for ACT in transfer learning scenarios under misspecified and overparameterized settings (Theorem 3.9). (...) can lead to the downstream data distribution being clustered by category in the representation space, provided that the upstream unlabeled sample size is sufficient. Hence, even with a few downstream samples, ACT can achieve outstanding classification performance, offering valuable insights for few-shot learning."

- The paper **clearly explains the assumptions** and **provides theoretical guarantees** on the angle between the classes in the target domain and the misclassification error in the source domain (Theorem 3.9).
The former **provides bounds** for which target class centers would be close to orthogonal and thus separable, **consequently leading to high transfer learning accuracy**.

- However, a key limitation of the current manuscript is that the experimental section only evaluates the method on the source domains, and **does not include transfer learning experiments** (i.e., evaluations on a target domain from a different dataset).

[1] Caron, Mathilde, et al. "Unsupervised learning of visual features by contrasting cluster assignments." Advances in neural information processing systems 33 (2020): 9912-9924.

[2] Caron, Mathilde, et al. "Emerging properties in self-supervised vision transformers." Proceedings of the IEEE/CVF international conference on computer vision. 2021.

**Essential References Not Discussed:**

No essential references are missing to the best of the reviewer's knowledge.

**Experimental Designs Or Analyses:**

- The linear+kNN evaluation protocol, datasets, and hyperparameter choices are **common in the literature**.
- I verified that the **results from previous methods reported in Table 2 align** with the ones reported in the W-MSE paper [4].
- The **results for re-implementations** of previous techniques (Barlow Twins and "HaoChen 2022") **are lower than expected**. I, however, did not check all re-implementation details and hyper-parameter choices in the provided code.

[4] Ermolov, Aleksandr, et al. "Whitening for self-supervised representation learning." International conference on machine learning. PMLR, 2021.

**Methods And Evaluation Criteria:**

- The benchmarks make sense to show that the method is effective for learning representations on the in-domain/source-domain.
- However, the evaluations lack transfer learning benchmarks, similar to e.g. "Table 3. Transfer learning: image classification." in Barlow Twins [3].


[3] Zbontar, Jure, et al. "Barlow twins: Self-supervised learning via redundancy reduction." International conference on machine learning. PMLR, 2021.

**Other Comments Or Suggestions:**

- typo line 196, left: "Base on $\mathcal{A}$" -> "Based on $\mathcal{A}$"
- typo line 326, left: "(...) we can systematically analysis" -> "(...) we can systematically analyze"
- Are Assumption 3.5 on the Lipschitz constant, and the other assumptions, realistic for the class of augmentations used in the experiments (described in the paragraph "Image transformations details")?

**Other Strengths And Weaknesses:**

- In its training objective, the paper **introduces an additional *alignment* loss term to be invariant to data augmentations**. However, **the other (unbiased) objective already includes an alignment term** in the diagonal part of the cross-correlation matrix. Therefore, and given that Barlow Twins does not require it, is this additional alignment term necessary for the performance of this method?

**Questions For Authors:**

1. As both the text (e.g., in the abstract: "Adversarial Contrastive Learning (ACT), a novel unbiased self-supervised transfer learning approach") and the theoretical claims suggest the method is effective at transfer learning, could the authors **evaluate the method on a SSL transfer learning benchmark**?
2. Since Barlow Twins' formulation is very close and does not require an additional alignment objective, **is the additional alignment term of ACT necessary** for this method? Could the authors run a small ablation study on this question and/or discuss it?

The technical and theoretical contributions of the paper are valuable to the community, but the paper lacks an essential evaluation (see question 1).
Therefore, the reviewer suggests "weak accept", and is willing to raise the rating if the questions are addressed.

**Relation To Broader Scientific Literature:**

The paper provides an overall clear positioning of its research questions:

- It is part of the family of SSL decorrelation techniques (Barlow Twins, W-MSE, VICReg, ...), and its formulation is most similar to the one from Barlow Twins.
- The authors introduce related theoretical studies (on population risk in SSL and the generalization error in SSL) and detail the differences with their contributions.
- *(minor)* The paper could benefit from a short introduction and mention of related work in adversarial learning, as it is a component of the introduced method and algorithm.

**Theoretical Claims:**

I did not verify the correctness of the proofs provided in the supplementary material.

---

> ### Author Rebuttal · Authors · 2025-04-01
>
> We thank you for your thorough review of our manuscript and for your constructive suggestions. Our point-by-point responses to your comments are given below. Thank you for pointing out these typos. We correct them in the revised manuscript.
>
> > **C1** Additional experiments: (1) comparisons with SOTA methods, and (2) transfer learning.
>
> Thank you for your constructive suggestion.
> * We have re-implemented SwAV, and the corresponding experimental results are presented in https://anonymous.4open.science/r/RE6-DDAF. Due to time constraints, we were unable to include additional comparisons with DINO. Alternatively, we kindly ask you to refer to the benchmark results in CIFAR10 available at https://docs.lightly.ai/self-supervised-learning/getting_started/benchmarks.html. In summary, **ACT outperforms both SwAV and DINO**, demonstrating its strong competitive performance.
> * The transferability of ACT: Please see the response to **C6** of reviewer L3h2.
>
> > **C2** The results for re-implementations are lower than expected.
>
> Thank you for your comment.
> * Our re-implementation of Barlow Twins is based on the **official implementation** available at https://github.com/facebookresearch/barlowtwins. Unfortunately, the authors of ``HaoChen 2022'' do not provided their implementation, so we implement it according to their paper.
> * To ensure a fair comparison, the **hyperparameter choices** for ACT and all other methods, including Barlow Twins and ``HaoChen 2022'', is almost aligned with the settings used in [1].
> * The official Barlow Twins implementation's README provides a link to a Barlow Twins experiment on CIFAR-10 as https://github.com/IgorSusmelj/barlowtwins. **The experimental results reported in this linked implementation closely align with ours**, further supporting the fairness of our re-implementation.
>
> [1] Aleksandr Ermolov, Aliaksandr Siarohin, Enver Sangineto, and Nicu Sebe. Whitening for Self-Supervised Representation Learning. (2021)
>
> > **C3** Additional alignment objective.
>
> We agree that the the diagonal part of the cross-correlation matrix serves as a alignment term **under certain conditions**. As shown by Lemma 4.1 of [1], the alignment risk is then related to the diagonal part of the cross-correlation matrix as:
> $$
> \mathcal{L}\_{\mathrm{align}}^{2}(f)\leq 4d\sum_{i=1}^{d}\big\\{\mathbb{E}\_{x}\mathbb{E}\_{x_{1}\in\mathcal{A}(x)}[f_{i}(x_{1})^{2}]-\mathbb{E}\_{x}\mathbb{E}\_{x_{1},x_{2}\in\mathcal{A}(x)}[f_{i}(x_{1})f_{i}(x_{2})]\big\\}^{2}.
> $$
> It is crucial that the right-hand side of the inequality is consistent with the alignment term in the loss function of Barlow Twins, provided that $\mathbb{E}\_{x}\mathbb{E}\_{x_{1}\in\mathcal{A}(x)}[f_{i}(x_{1})^{2}]=1$ for each $i\in\\{1,\ldots,d\\}$. **However, this condition does not hold generally.**
> * From a theoretical perspective, the cross-correlation loss alone, as discussed previously, is insufficient for learning representations invariant to augmentations, as previously discussed. To address this, we introduce an additional explicit alignment term in the loss, as also suggested by [2,3].
> * From a practical perspective, we fully agree with the necessity of distinguishing the effect of alignment from the de-biased operation. We conduct an ablation study comparing ACT with and without the explicit alignment term as https://anonymous.4open.science/r/RE2-7B34. Our results indicate that **the explicit alignment term can slightly improves ACT's performance**.
>
>  [1] Weiran Huang, Mingyang Yi, Xuyang Zhao, and Zihao Jiang. Towards the Generalization of Contrastive Self-Supervised Learning. (2023)
>
> [2] Jeff Z. HaoChen, Colin Wei, Ananya Kumar, and Tengyu Ma. Beyond Separability: Analyzing the Linear Transferability of Contrastive Representations to Related Subpopulations. (2022)
>
> [3] Jeff Z. HaoChen, and Tengyu Ma. A Theoretical Study of Inductive Biases in Contrastive Learning. (2023)
>
> > **C4** Assumption 3.5.
>
> We appreciate your question. As outlined in the section ``Image transformations details'', the data augmentations used in our experiments -- crops, horizontal mirroring, brightness adjustment, grayscaling, and Gaussian blurring -- are linear operations, and thus Lipschitz continuous. Specifically, the Lipschitz constants of crops horizontal mirroring, and grayscaling are less than and equal to 1, and the Lipschitz constant of brightness adjustment depends on the adjustment factor. The Lipschitz constant of Gaussian blurring relies on the the kernel size and the variance of Gaussian. **Consequently, all Lipschitz constants can be explicitly calculated**, but due to character limitations, we had to place these details in the additional appendix of the revised manuscript.
>
> > **C5** Assumption 3.7.
>
> See the response to **C4** of Reviewer etBC for a detailed discussion on Assumption 3.7.

---

> > ### Comment · Reviewer_LFM2 · 2025-04-03
> >
> > Thank you for the detailed answers and for running the requested additional experiments in this short time frame.
> > Since the authors addressed all my concerns, I am raising my rating to "accept" as stated.

---

> > > ### Author Response · Authors · 2025-04-04
> > >
> > > We would like to thank you once again for your valuable contributions and insightful suggestions. Your specialized reviews have undoubtedly helped improve the quality of our paper, particularly regarding the ablation experiments on the alignment term, transfer learning, and the justification for the Lipschitz property of the used augmentations.

---

### Official Review · Reviewer_i8re · 2025-04-12

**Overall Recommendation:** 2

**Summary:**

The paper proposes a method for unsupervised transfer learning called Adversarial Contrastive Training (ACT). The key idea is to address the bias present in sample-level estimators of self-supervised contrastive learning by reformulating the regularization term into a minimax (adversarial) framework. In this formulation, a matrix variable G is introduced and alternated with the encoder, with the intent of minimizing the inherent bias from mini-batch estimation. The paper provides an end-to-end theoretical analysis on how the method benefits downstream tasks by proving convergence properties under several assumptions.

**Claims And Evidence:**

Claims: The paper claims that reformulating the self-supervised objective into a minimax problem involving an auxiliary variable G can obtain an unbiased estimator with better representations than existing biased methods.

Evidence: The theoretical part builds an error decomposition that formally connects the adversarial formulation to downstream clustering, and the experiments report modest improvements in accuracy. However, the experimental evidence is not sufficient for the limited datasets and marginal performance improvement.

**Essential References Not Discussed:**

N/A

**Experimental Designs Or Analyses:**

Design: The experiments are designed to compare ACT against existing methods on standard benchmarks.

Issues: The experimental section suffers from several shortcomings:

- There is a repetition of experimental results (as similar comparisons appear in both Sections 2.2 and 4) while seemingly addressing the same task.

- The datasets chosen for evaluation are small and do not cover more challenging datasets that would be more representative of real-world applications.

- The performance gains reported are marginal, and no ablation study specifically analyzes the effectiveness of introducing G.

**Methods And Evaluation Criteria:**

Methods: The paper’s main methodological innovation is the transformation of a regularization term into a maximization over an auxiliary matrix G. This leads to a minimax optimization framework where the encoder is updated to counteract the worst-case bias estimated by G. The alternating optimization algorithm (Algorithm 1) is designed to iteratively update the encoder and the adversarial variable.

Evaluation Criteria: The method is evaluated on common transfer learning benchmarks using linear probe and k-nearest neighbors classifiers. However, the evaluation is limited to small-scale datasets (CIFAR series and Tiny ImageNet)

**Other Comments Or Suggestions:**

See listed above.

**Other Strengths And Weaknesses:**

Strengths:

- Recasting the regularization term into a minimax framework is conceptually interesting and may open up new research directions in self-supervised learning.

- The paper provides a rigorous theoretical analysis that links the adversarial training process to concrete guarantees on downstream performance.

Weaknesses:

- Writing Clarity:

  - The explanation of the role of the auxiliary variable G is insufficient. The paper does not convey its meaning or the motivation behind its introduction. No dedicated section or ablation study specifically validates the contribution of G.

  - The submission suffers from structural issues: experimental results are repeated in different sections, making the narrative confusing, and the overall organization of the paper is somewhat chaotic.

  - The heavy reliance on dense mathematical derivations with minimal intuitive explanations makes it less accessible to readers who are not experts in theoretical machine learning.

- Experimental Insufficiency:

  - Evaluation is based on a small number of datasets with limited representativeness. The performance gains are marginal, raising questions about the proposed method's practical impact.

**Questions For Authors:**

See listed above.

**Relation To Broader Scientific Literature:**

The method is positioned within the rapidly growing area of self-supervised contrastive learning, where many recent works (e.g., Barlow Twins, SimCLR, BYOL) address the challenges associated with negative sampling and model collapse. By reformulating the self-supervised loss into an adversarial (minimax) problem, this work attempts to remove biases inherent in mini-batch estimators—a concern also touched upon in prior studies by HaoChen et al. and Zbontar et al.

**Theoretical Claims:**

The authors derive convergence results and error bounds that relate the downstream classification error to the minimax optimization formulation. They provide detailed proofs in the supplementary materials regarding error decomposition and convergence under several assumptions.

---

### Decision · Program_Chairs · 2025-05-01

**Decision:**

Reject

**Comment:**

This paper introduces adversarial contrastive learning, which incorporates min-max adversarial training into the self-supervised learning procedure.

This paper presents a theoretical guarantee, which gives an error bound on the performance of this adversarial contrastive learning procedure. This bound depends on various parameters of the augmentation and of the neural network parameterization.

The paper is reviewed by six reviewers and myself (two reviewers were added during an earlier stage of the review process). Since this paper’s contribution appears to be theoretical, several reviewers mention in their reviewers that they did not check the proofs in the appendix—Therefore, those reviews are down weighted.

Several reviewers did mention that “the presentation of theoretical results is not clear and concise enough.” (i8re and etBC).
- In the appendix, I find the proofs hard to follow:
    - In line 615, I find the “f cup” notation a little strange and it is not clearly stated in this proof.
    - Again in line 615, here it is implicit assumed that the product of the $\infty$ norm of all the layers are bounded by some $\kappa$. Is this a valid assumption to make? I did not see it clearly stated in the main theorem or in the introduction.
    - In line 661, what is $\tilde C_t(i)$?
    - Line 948, what is $d^*$?
- Some of the assumptions are not clearly motivated, lacking connections to contrastive learning practice:
    - For instance, Assumption 3.8, when would be likely to hold in practice? A discussion would be warranted or at least some experiments would help to better explain the cases where the theory will apply.
- At a higher level, my understanding is that this paper heavily draws on HaoChen & Ma (2023), and Huang, Yi, Zhao, Jiang (2023). I agree with several reviewers that the incremental contribution is not clearly stated. Perhaps adding some illustrative cases in a simplified setting would be helpful.

In summary, while most reviewers appreciate the strength of this work, including its novel formulation on theoretical contrastive learning, reviewers also find the proofs to be difficult to verify due to unclear presentation, and the contribution on top of prior work unclear. Additionally, there are assumptions required in the proof that are not explicitly stated in the theorem statement. Taken together, I thus recommend a reject for this submission.